

SciPost Phys. Lect. Notes 82 (2024)

# The quantum Ising chain for beginners

**Glen Bigan Mbeng[1], Angelo Russomanno[2] and Giuseppe E. Santoro[3,4,5]**

1 Universität Innsbruck, Technikerstraße 21 a, A-6020 Innsbruck, Austria
2 Dipartimento di Fisica "E. Pancini", Università di Napoli Federico II,
Complesso di Monte S. Angelo, via Cinthia, I-80126 Napoli, Italy
3 SISSA, Via Bonomea 265, I-34136 Trieste, Italy
4 International Centre for Theoretical Physics, P.O.Box 586, I-34014 Trieste, Italy
5 CNR-IOM Democritos, Via Bonomea 265, I-34136 Trieste, Italy

⋆ santoro@sissa.it

## Abstract

We present here various techniques to work with clean and disordered quantum Ising chains, for the benefit of students and non-experts. Starting from the Jordan-Wigner transformation, which maps spin-1/2 systems into fermionic ones, we review some of the basic approaches to deal with the superconducting correlations that naturally emerge in this context. In particular, we analyze the form of the ground state and excitations of the model, relating them to the symmetry-breaking physics, and illustrate aspects connected to calculating dynamical quantities, thermal averages, correlation functions, and entanglement entropy. A few problems provide simple applications of the techniques.

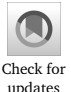

# 1 Introduction

The quantum many-body problem is notoriously difficult [1, 2]. Recent times have seen tremendous development in our experimental abilities in dealing with well-controlled quantum systems, in different platforms, from superconducting qubits used in quantum information processing [3–6] to quantum simulators, for instance with trapped ions [7] or ultracold atoms [8–10].

These experimental advances call for a parallel theoretical understanding of equilibrium, both at zero and finite temperatures, and out-of-equilibrium properties of systems of interacting spins. While few exactly solvable models are known [11], many numerical techniques have been developed in the last decades, ranging from quantum Monte Carlo [12] to density matrix renormalization group [13, 14], matrix product states [15] and tensor networks [16, 17].

An extremely rich class of models is given by Ising models with long-range interactions, directly relevant to many experimental platforms, like trapped ions [7], and Rydberg atoms [18]. For Rydberg atoms [18], the relevant Hamiltonian, when written in terms of spin-1/2 (Pauli) operators $\hat{\sigma}_j^\alpha$ (with $\alpha = x, y, z$ and $j$ a site-index) has the form:[1]

$$\widehat{H} = \sum_{i<j} J_{ij} \hat{\sigma}_i^z \hat{\sigma}_j^z + \sum_i \left( h_i^x \hat{\sigma}_i^x + h_i^z \hat{\sigma}_i^z \right), \tag{2}$$

where the couplings $J_{ij}$ — in principle long-ranged — can be tuned, for instance, by varying the distance between Rydberg atoms.

What we are going to study in these lecture notes is the much simpler case of an XY/Ising

---

[1]This is based on the identification, at each site $i$, of the atomic ground $|g_i\rangle$ and Rydberg excited state $|r_i\rangle$ with the spin-1/2 eigenstates of $\hat{\sigma}_i^z = |g_i\rangle\langle g_i| - |r_i\rangle\langle r_i|$, $|\uparrow\rangle$ and $|\downarrow\rangle$, respectively, with $\hat{\sigma}_i^x = |r_i\rangle\langle g_i| + |g_i\rangle\langle r_i|$. In terms of the projector on the Rydberg state $\hat{n}_i = |r_i\rangle\langle r_i|$, a standard expression for $\widehat{H}$ is:

$$\widehat{H} = \sum_i \frac{\hbar\Omega_i}{2} \hat{\sigma}_i^x - \sum_i \hbar\Delta_i \hat{n}_i + \sum_{i<j} V_{ij} \hat{n}_i \hat{n}_j, \tag{1}$$

where $\Omega_i$ is the Rabi frequency driving, $\Delta_i$ the detuning, and $V_{ij}$ the interaction between Rydberg states at different sites [18]. By writing $\hat{n}_i = (1 - \hat{\sigma}_i^z)/2$, the spin representation in Eq. (2) follows directly.

spin chain with nearest-neighbour interactions in a transverse field:

$$\widehat{H} = -\sum_{j=1}^{L}\left(J_j^x \hat{\sigma}_j^x \hat{\sigma}_{j+1}^x + J_j^y \hat{\sigma}_j^y \hat{\sigma}_{j+1}^y\right) - \sum_{j=1}^{L} h_j \hat{\sigma}_j^z, \tag{3}$$

where $L$ is the length of the chain, and we will allow for arbitrary nearest-neighbour couplings $J_j^{x/y}$ in the x and y direction in spin space, and arbitrary transverse fields $h_j$.

Interestingly, the quantum Ising chain with nearest-neighbour interactions is closely related to topological superconductivity, as the model is unitarily equivalent to the quadratic fermionic Hamiltonian of a p-wave superconducting chain (see Sec. 3), that displays a topological phase where zero-energy boundary Majorana modes appear [19].[2] The p-wave superconducting model, also known as the Kitaev chain, has been the center of a lot of research in topological superconductivity, see Refs. [20–22] for a review.

The quantum Ising chain problem is an ideal playground for testing many of the ideas of statistical mechanics, including recent non-equilibrium physics. As such, it is a standard test case in much of the recent literature. It has been used for studying the effect of quantum quenches in integrable systems [23–33], Kibble-Zurek scaling of excitations [34–37], dynamical quantum phase transitions [38, 39], dynamics of periodically driven systems [36, 40–44], entanglement transitions [45–55], work statistics [56–59], and time crystals [60, 61], just to give some examples. A recent book on the subject, Ref. [62], can be used as a source for some more literature.

These notes are intended for students willing to begin working with quantum Ising chains. They can be also useful as a practical guide to researchers entering the field. We, unfortunately, do no justice to the immense literature where concepts and techniques were first introduced or derived, and even less so to the many papers where physical applications are presented. We apologize in advance for that with authors whose work is not duly cited. However, most of the topics include detailed derivations which should make these notes reasonably self-standing.

The level of our presentation is roughly appropriate for graduate students, but master students should also be able to follow most of the developments, provided they acquire the necessary pre-requisites: second quantization [1] to deal with bosons and fermions, and basic knowledge of quantum mechanics of the spin-1/2 [63].

More in detail, we will discuss how to map, through the Jordan-Wigner transformation, the XY/Ising spin chain in Eq. 3 into a quadratic spinless fermion model:

$$\widehat{H} = -\sum_{j=1}^{L}\left((J_j^x + J_j^y)\hat{c}_j^\dagger \hat{c}_{j+1} + (J_j^x - J_j^y)\hat{c}_j^\dagger \hat{c}_{j+1}^\dagger + \text{H.c.}\right) + \sum_{j=1}^{L} h_j(2\hat{n}_j - 1), \tag{4}$$

where $\hat{c}_j^\dagger$ creates a spinless fermion at site $j$, and H.c. means Hermitian conjugate. This Hamiltonian, with a few details on the boundary conditions which we will discuss at length, coincides with the celebrated Kitaev chain model [19] for p-wave superconductors, supporting Majorana modes at the boundaries of an open chain.

Following that, we will show how essentially any static and dynamic property of the model can be determined. In particular, finding the $2^L$ eigenvalues (including multiplicities) and associated eigenvectors of $\widehat{H}$ amounts to diagonalising a $2L \times 2L$ matrix $\mathbb{H}$ containing the couplings, a massive simplification which allows dealing with very large chain lengths, $L \sim 1000$, with moderate numerical effort. With comparable efforts, one can calculate thermal properties, spin-spin correlation functions, and the entanglement entropy. Moreover, an explicit time-dependence of the Hamiltonian parameters can be dealt with quite easily by integrating over time a system of $2L$ differential equations, the so-called Bogoliubov-de Gennes equations.

---

[2]This phase corresponds to the symmetry-breaking phase of the quantum Ising chain, as we will better clarify in Sec. 4, and 7.2.

Here is an outline of the material presented. We start, in Sec. 2, from the Jordan-Wigner transformation, which allows mapping the spin-chain Hamiltonian in Eq. (3) into the spinless fermion Hamiltonian in Eq. (4), as detailed in Sec. 3. Next, Sec. 4 treats the case of an ordered Ising chain with periodic boundary conditions, where a simple analytical reduction to an assembly of $2 \times 2$ problems is possible. In Sec. 6 we discuss the Nambu formalism for dealing with the quadratic fermionic Hamiltonian in the general disordered case. Section 7 shows how to diagonalize the spinless fermion Hamiltonian in the general disordered case, while in Sec. 8 we derive the Bogoliubov-de Gennes equations which encode the unitary Schrödinger dynamics for the case of a time-dependent Hamiltonian. Sections 9 and 10 contain the technicalities related to the calculation of correlation functions involving Jordan-Wigner string operators and the entanglement entropy, while in Sec. 11 we show how to calculate thermal averages. Finally, in Appendix A we show how to calculate the overlap between different Fock states for two different Ising Hamiltonians.

A word on the notation. We will try to be consistent with a notation in which quantum mechanical operators acting in the full Hilbert space, $2^L$ dimensional for a system of $L$ spins, are denoted with a hat, such as $\widehat{H}$ for the Hamiltonian or $\widehat{U}$ for a unitary operator. Matrices (and vectors) are denoted in boldface, such as $\mathbf{U}$, if they refer to $L \times L$ (or $2 \times 2$) block matrices, or as $\mathbb{U}$ or $\mathbb{H}$, if they refer to $2L \times 2L$ matrices. Here is a table where the main symbols are explained.

| | |
|---|---|
| $\hat{\sigma}_j^\alpha$ | Pauli matrices ($\alpha = x, y, z$) at site $j$ |
| $\hat{c}_j^\dagger$ | Creation operator for a spinless fermion at site $j$ |
| $\hat{c}_k^\dagger$ | Creation operator for a spinless fermion with wave-vector $k$ |
| $\hat{\gamma}_\mu^\dagger$ | Creation operator for a Bogoliubov fermion |
| $\widehat{H}$ | The Hamiltonian operator |
| $\widehat{H}_{p=0,1}$ | The parity even/odd projection (for p $= 0/1$) of the fermionic Hamiltonian |
| $\widehat{\mathbb{H}}_{p=0,1}$ | The fermionic Hamiltonian with ABC/PBC (for p $= 0/1$) |
| $\widehat{\Psi}_k$ | The two-component Nambu fermion operator with wave-vector $k$ |
| $\mathbf{H}_k$ | The $2 \times 2$ Hamiltonian block with wave-vector $k$ |
| $\mathbf{R}_k$ | The 3-dimensional effective magnetic field with wave-vector $k$ |
| $\mathbf{U}_k$ | The $2 \times 2$ unitary matrix with eigenvectors of $\mathbf{H}_k$ with wave-vector $k$ |
| $\widehat{\Psi}$ | The $2L$-dimensional Nambu fermion operator composed of $\hat{c}_j$ and $\hat{c}_j^\dagger$ |
| $\widehat{\Phi}$ | The $2L$-dimensional Nambu fermion operator composed of $\hat{\gamma}_\mu$ and $\hat{\gamma}_\mu^\dagger$ |
| $\mathbb{H}$ | The $2L \times 2L$ Hamiltonian matrix in the Nambu formalism |
| $\mathbb{U}$ | The $2L \times 2L$ unitary matrix with eigenvectors of $\mathbb{H}$ |
| $\mathbf{U}, \mathbf{V}$ | $L \times L$ blocks of the unitary matrix $\mathbb{U}$ |

# 2 Jordan-Wigner transformation

For systems of bosons and fermions, a large assembly of many-body techniques has been developed [1]. In particular, while the full Hilbert space for particles on lattices with $L$ sites is exponentially large in $L$, models which are *quadratic* in the fermion or bosons creation/destruction operators are "solvable", as they reduce to solving a *single-particle problem* [1]. Spin systems, on the contrary, are neither bosons nor fermions. Their full Hilbert space is also exponentially large — $2^L$ for spin-1/2 on a lattice with $L$ sites — but the equivalent of a "solvable quadratic" problem is lacking: even the simplest spin-spin interactions make the Hamiltonian essentially unsolvable, in general.

Consider, to start with, a single spin-1/2, and the three components of the spin operators represented in terms of the usual Pauli matrices[3] $\hat{\sigma}^{\alpha}$ with $\alpha = x, y, z$. The Hilbert space of a single spin is two-dimensional: for instance, you can write a basis as $\{|\uparrow\rangle, |\downarrow\rangle\}$, in terms of the eigenstates of $\hat{\sigma}^z$, with $\hat{\sigma}^z|\uparrow\rangle = |\uparrow\rangle$ and $\hat{\sigma}^z|\downarrow\rangle = -|\downarrow\rangle$. Moreover, if $\hat{\sigma}_j^{\alpha}$ denote Pauli matrices at different lattice sites $j$, hence acting on "different" (distinguishable) two-dimensional Hilbert spaces, then

$$\left[\hat{\sigma}_j^{\alpha}, \hat{\sigma}_{j'}^{\alpha'}\right] = 0, \qquad \text{for} \quad j' \neq j. \tag{5}$$

But on the *same* site, the angular momentum commutation rules lead to

$$\left[\hat{\sigma}_j^x, \hat{\sigma}_j^y\right] = 2i\hat{\sigma}_j^z, \tag{6}$$

and cyclic permutations [63]. Interestingly, by defining the raising and lowering operators $\hat{\sigma}_j^{\pm} = (\hat{\sigma}_j^x \pm i\hat{\sigma}_j^y)/2$ which act on the basis states as $\hat{\sigma}^+|\downarrow\rangle = |\uparrow\rangle$ and $\hat{\sigma}^-|\uparrow\rangle = |\downarrow\rangle$, you can verify that

$$\left\{\hat{\sigma}_j^+, \hat{\sigma}_j^-\right\} = 1, \tag{7}$$

where $\left\{\hat{A}, \hat{B}\right\} = \hat{A}\hat{B} + \hat{B}\hat{A}$ denotes the *anti-commutator*, typical of the canonical anti-commutation rules for fermions [1].

Using bosons to describe spins would seem impossible. First of all, if we have a single boson $\hat{b}^{\dagger}$ with associated vacuum state $|0\rangle$, such that $\hat{b}|0\rangle = 0$, then, using the canonical bosonic commutation rules $\left[\hat{b}, \hat{b}^{\dagger}\right] = 1$ you can construct an infinite dimensional Hilbert space [1] with states

$$|n\rangle = \frac{1}{\sqrt{n!}}(\hat{b}^{\dagger})^n|0\rangle, \quad \text{where} \quad n = 0, 1, \cdots \infty.$$

However, if we decide to truncate such a Hilbert space to *only two states*, $\{|0\rangle, |1\rangle\}$, assuming $(\hat{b}^{\dagger})^2|0\rangle = 0$, then the Hilbert space of a single spin-1/2 can be easily mimicked. Such a truncation, which can be thought of as adding a large — ideally "infinite" — on-site repulsion term to the boson Hamiltonian, is known as *hard-core boson*. We transform the Pauli spin-1/2 operators $\hat{\sigma}_j^{\alpha}$ (with $\alpha = x, y, z$, and $j$ a generic site index) into *hard-core bosons* $\hat{b}_j^{\dagger}$, by identifying[4] at each site $|0\rangle \leftrightarrow |\uparrow\rangle$ and $|1\rangle = \hat{b}^{\dagger}|0\rangle \leftrightarrow |\downarrow\rangle$. Recalling that $\hat{\sigma}^{\pm} = (\hat{\sigma}^x \pm i\hat{\sigma}^y)/2$ act as $\hat{\sigma}^+|\downarrow\rangle = |\uparrow\rangle$, and $\hat{\sigma}^-|\uparrow\rangle = |\downarrow\rangle$, we must have:

$$\begin{cases} \hat{\sigma}_j^+ = \hat{b}_j, \\ \hat{\sigma}_j^- = \hat{b}_j^{\dagger}, \\ \hat{\sigma}_j^z = 1 - 2\hat{b}_j^{\dagger}\hat{b}_j \end{cases} \implies \begin{cases} \hat{\sigma}_j^x = \hat{b}_j^{\dagger} + \hat{b}_j, \\ \hat{\sigma}_j^y = i(\hat{b}_j^{\dagger} - \hat{b}_j), \\ \hat{\sigma}_j^z = 1 - 2\hat{b}_j^{\dagger}\hat{b}_j. \end{cases} \tag{8}$$

These operators $\hat{b}_j^{\dagger}$ commute at different sites — as the original $\hat{\sigma}_j^{\alpha}$ do — but are not ordinary bosonic operators. They anti-commute on the same site[5] $\left\{\hat{b}_j, \hat{b}_j^{\dagger}\right\} = 1$ and they verify the hard-core constraint $(\hat{b}_j^{\dagger})^2|0\rangle = 0$, i.e., *at most one boson* is allowed on each site.

---

[3]Recall that

$$\hat{\sigma}^x = \begin{pmatrix} 0 & 1 \\ 1 & 0 \end{pmatrix}, \qquad \hat{\sigma}^y = \begin{pmatrix} 0 & -i \\ i & 0 \end{pmatrix}, \qquad \hat{\sigma}^z = \begin{pmatrix} 1 & 0 \\ 0 & -1 \end{pmatrix},$$

which verify $[\hat{\sigma}^x, \hat{\sigma}^y] = 2i\hat{\sigma}^z$. The physical spin operators have an extra factor $\hbar/2$.

[4]This identification is not unique, as you can swap the two states.

[5]Since on the same site $\left\{\hat{\sigma}_j^+, \hat{\sigma}_j^-\right\} = 1$, this implies that $\left\{\hat{b}_j, \hat{b}_j^{\dagger}\right\} = 1$, while ordinary bosons would have the commutator $[\hat{b}_j, \hat{b}_j^{\dagger}] = 1$.

$$= \hat{b}_1^\dagger \, \hat{b}_2^\dagger \, \hat{b}_5^\dagger \, |0\rangle \; = \; \hat{c}_1^\dagger \, \hat{c}_2^\dagger \, \hat{c}_5^\dagger \, |0\rangle$$

Figure 1: Top: An $L = 6$ site spin configuration. Bottom: The corresponding particle configuration.

**ⓘ Info:** The hard-core boson mapping might be viewed as a way of rewriting spin-1/2 models in a rather general setting. For instance, if you have a Heisenberg model for spin-1/2 sitting on a lattice, whose sites are denoted by $j$ and with nearest-neighbour pairs denoted by $\langle j, j' \rangle$ we could write, defining $\hat{n}_j = \hat{b}_j^\dagger \hat{b}_j$:

$$\widehat{H}_{\text{Heis}} = \frac{J}{4} \sum_{\langle j,j' \rangle} \left( \hat{\sigma}_j^z \hat{\sigma}_{j'}^z + 2(\hat{\sigma}_j^+ \hat{\sigma}_{j'}^- + \hat{\sigma}_j^- \hat{\sigma}_{j'}^+) \right) \to J \sum_{\langle j,j' \rangle} \left( (\hat{n}_j - \tfrac{1}{2})(\hat{n}_{j'} - \tfrac{1}{2}) + \tfrac{1}{2}(\hat{b}_{j'}^\dagger \hat{b}_j + \hat{b}_j^\dagger \hat{b}_{j'}) \right).$$

The second expression shows that we are dealing with hard-core bosons hopping on the lattice and repelling each other at nearest-neighbors. Needless to say, this helps in no way in solving the problem.

The hard-core constraint seems to be ideally representable in terms of *spinless fermions* $\hat{c}_j^\dagger$, where the absence of double occupancy is automatically enforced by the *Pauli exclusion principle*, and the anti-commutation on the same site comes for free.

Unfortunately, whereas the mapping of $\hat{\sigma}_j^\alpha$ into hard-core bosons $\hat{b}_j^\dagger$ is true in any spatial dimension, writing $\hat{b}_j^\dagger$ in terms of spinless fermions $\hat{c}_j^\dagger$ is straightforwardly useful only in *one-dimension* (1D), where a natural ordering of sites is possible, $j = 1, 2, \cdots, L$. In other words, because fermion operators on different sites must anti-commute, the exact handling of the resulting minus signs — which are absent in the original spin problem — is very natural only in 1D.

Let $\hat{c}_j^\dagger$ and $\hat{c}_j$ denote the creation and annihilation operators for spinless fermions at site $j$, with canonical *anti-commutation* relations $\{\hat{c}_j, \hat{c}_{j'}^\dagger\} = \delta_{j,j'}$, $\{\hat{c}_j, \hat{c}_{j'}\} = \{\hat{c}_j^\dagger, \hat{c}_{j'}^\dagger\} = 0$, and $\hat{n}_j = \hat{c}_j^\dagger \hat{c}_j$ the corresponding number operator, whose eigenvalues can be only 0 or 1.

**ⓘ Key properties of $e^{i\pi\hat{n}_j}$.** Important in the whole discussion below are the following simple properties of $e^{i\pi\hat{n}_j}$:

$$
\begin{aligned}
&\textbf{K1)} \quad e^{i\pi\hat{n}_{j'}}\hat{c}_j = \hat{c}_j \, e^{i\pi\hat{n}_{j'}}, \qquad \text{for} \quad j' \neq j, \\[4pt]
&\textbf{K2)} \quad e^{i\pi\hat{n}_j}\hat{c}_j = -\hat{c}_j \, e^{i\pi\hat{n}_j}, \\[4pt]
&\textbf{K3)} \quad e^{i\pi\hat{n}_j} e^{i\pi\hat{n}_j} = 1, \\[4pt]
&\textbf{K4)} \quad e^{-i\pi\hat{n}_j} = e^{i\pi\hat{n}_j} = 1 - 2\hat{n}_j.
\end{aligned}
\tag{9}
$$

In words: $e^{i\pi\hat{n}_j}$ commutes with fermionic operators at different sites, while it anti-commutes on the same site. The anti-commutation in **K2** can be verified by using the fermionic anti-commutation rules, or by arguing that $\hat{n}_j = 0$ when it sits to the left of $\hat{c}_j$, while $\hat{n}_j = 1$ when

it sits to the right of $\hat{c}_j$. **K3**, equivalent to $e^{i2\pi\hat{n}_j} = 1$, implies that $e^{i\pi\hat{n}_j} = e^{-i\pi\hat{n}_j}$. By taking the Hermitian conjugate of **K1** and **K2** you obtain identical expressions for $\hat{c}_j^\dagger$. **K4** follows because the possible eigenvalues of $\hat{n}_j$ are 0 and 1.

The Jordan-Wigner (JW) transformation of hard-core bosons into spinless fermions reads:

$$\hat{b}_j = \hat{K}_j \hat{c}_j = \hat{c}_j \hat{K}_j\,, \qquad \text{with} \qquad \hat{K}_j = \prod_{j'=1}^{j-1} e^{i\pi\hat{n}_{j'}} = e^{i\pi\sum_{j'=1}^{j-1}\hat{n}_{j'}}\,, \qquad (10)$$

where the *non-local* string operator $\hat{K}_j$ is simply a *sign*, $\hat{K}_j = \pm 1$, counting the parity of the number of fermions *before* site $j$, between sites 1 and $j-1$, multiplying the fermionic operator $\hat{c}_j$, with which it commutes.

We will now show that the following two properties of the $\hat{b}_j$ follow:

$$\textbf{Prop.1}: \begin{pmatrix} \{\hat{b}_j, \hat{b}_j^\dagger\} = 1\,, \\ \{\hat{b}_j, \hat{b}_j\} = 0\,, \\ \{\hat{b}_j^\dagger, \hat{b}_j^\dagger\} = 0\,, \end{pmatrix} \qquad \textbf{Prop.2}: \begin{pmatrix} \left[\hat{b}_j, \hat{b}_{j'}^\dagger\right] = 0\,, \\ \left[\hat{b}_j, \hat{b}_{j'}\right] = 0\,, \qquad \text{if } j \neq j'\,, \\ \left[\hat{b}_j^\dagger, \hat{b}_{j'}^\dagger\right] = 0\,, \end{pmatrix} \qquad (11)$$

which is a formal way of writing that the $\hat{b}_j$ are *hard-core bosons*. **Prop.1** is straightforward because the string $\hat{K}_j$ cancels completely, for instance

$$\hat{b}_j^\dagger \hat{b}_j = \hat{c}_j^\dagger \hat{K}_j^\dagger \hat{K}_j \hat{c}_j = \hat{c}_j^\dagger \hat{c}_j\,,$$

and, similarly, $\hat{b}_j \hat{b}_j^\dagger = \hat{c}_j \hat{c}_j^\dagger$. In essence, on each site $\hat{b}_j$ inherits the anti-commutation property **Prop.1** from the fermion $\hat{c}_j$.

To prove **Prop.2**, let us consider $\left[\hat{b}_{j_1}, \hat{b}_{j_2}^\dagger\right]$, assuming $j_2 > j_1$. Using Eq. (10) and properties **K1**, **K3** from Eq. (9) it is simple to show that

$$\hat{b}_{j_1} \hat{b}_{j_2}^\dagger = \hat{c}_{j_1}\left(\prod_{j=j_1}^{j_2-1} e^{i\pi\hat{n}_j}\right)\hat{c}_{j_2}^\dagger\,, \qquad (12)$$

which means that only the piece of JW string from $j_1$ to $j_2 - 1$ survives. Similarly, you can show that

$$\hat{b}_{j_2}^\dagger \hat{b}_{j_1} = \left(\prod_{j=j_1}^{j_2-1} e^{i\pi\hat{n}_j}\right)\hat{c}_{j_2}^\dagger \hat{c}_{j_1} = -\left(\prod_{j=j_1}^{j_2-1} e^{i\pi\hat{n}_j}\right)\hat{c}_{j_1}\hat{c}_{j_2}^\dagger$$

$$= +\hat{c}_{j_1}\left(\prod_{j=j_1}^{j_2-1} e^{i\pi\hat{n}_j}\right)\hat{c}_{j_2}^\dagger\,, \qquad (13)$$

where the change of sign in the first line is due to the fermionic anti-commutation, $\hat{c}_{j_2}^\dagger \hat{c}_{j_1} = -\hat{c}_{j_1}\hat{c}_{j_2}^\dagger$, and the crucial final change of sign is due to **K2**, while all the other operators $e^{i\pi\hat{n}_j}$ for $j \neq j_1$ commute with $\hat{c}_{j_1}$, due to **K1**. Comparing Eq. (12) with Eq. (13) you deduce that $\left[\hat{b}_{j_1}, \hat{b}_{j_2}^\dagger\right] = 0$. All the other commutation relationships in **Prop.2** are proven similarly.

Here is a summary of a few useful expressions where the string operator $\hat{K}_j$ disappears exactly:

$$
\begin{aligned}
\hat{b}_j^\dagger \hat{b}_j &= \hat{c}_j^\dagger \hat{c}_j\,, \\
\hat{b}_j^\dagger \hat{b}_{j+1}^\dagger &= \hat{c}_j^\dagger (1-2\hat{n}_j)\hat{c}_{j+1}^\dagger = \hat{c}_j^\dagger \hat{c}_{j+1}^\dagger\,, \\
\hat{b}_j^\dagger \hat{b}_{j+1} &= \hat{c}_j^\dagger (1-2\hat{n}_j)\hat{c}_{j+1} = \hat{c}_j^\dagger \hat{c}_{j+1}\,, \\
\hat{b}_j \hat{b}_{j+1} &= \hat{c}_j (1-2\hat{n}_j)\hat{c}_{j+1} = \hat{c}_j (1-2(1-\hat{c}_j\hat{c}_j^\dagger))\hat{c}_{j+1} = -\hat{c}_j\hat{c}_{j+1}\,, \\
\hat{b}_j \hat{b}_{j+1}^\dagger &= \hat{c}_j (1-2\hat{n}_j)\hat{c}_{j+1}^\dagger = \hat{c}_j (1-2(1-\hat{c}_j\hat{c}_j^\dagger))\hat{c}_{j+1}^\dagger = -\hat{c}_j\hat{c}_{j+1}^\dagger\,.
\end{aligned}
\tag{14}
$$

Notice the minus signs on the right-hand side, which should not be forgotten. Notice also that we have used

$$
\prod_{j'=1}^{j-1}\left(e^{i\pi\hat{n}_{j'}}\right)\prod_{j'=1}^{j}\left(e^{i\pi\hat{n}_{j'}}\right) = e^{i\pi\hat{n}_j} = 1-2\hat{n}_j\,,
\tag{15}
$$

because all but the last $e^{i\pi\hat{n}_j}$-term cancel in the two strings.



**Jordan-Wigner transformation.** Summarising, spins are mapped into fermions using:

$$
\begin{cases}
\hat{\sigma}_j^x = \hat{K}_j\,(\hat{c}_j^\dagger + \hat{c}_j)\,, \\
\hat{\sigma}_j^y = \hat{K}_j\,i(\hat{c}_j^\dagger - \hat{c}_j)\,, \\
\hat{\sigma}_j^z = 1-2\hat{n}_j\,,
\end{cases}
\qquad \text{with} \qquad \hat{K}_j = \prod_{j'=1}^{j-1} e^{i\pi\hat{n}_{j'}}\,.
\tag{16}
$$

Armed with these expressions, it is simple to show that some nearest-neighbor spin-spin operators transform simply into quadratic fermionic operators

$$
\begin{aligned}
\hat{\sigma}_j^x \hat{\sigma}_{j+1}^x &= \left(\hat{c}_j^\dagger \hat{c}_{j+1} + \hat{c}_j^\dagger \hat{c}_{j+1}^\dagger + \text{H.c.}\right)\,, \\
\hat{\sigma}_j^y \hat{\sigma}_{j+1}^y &= \left(\hat{c}_j^\dagger \hat{c}_{j+1} - \hat{c}_j^\dagger \hat{c}_{j+1}^\dagger + \text{H.c.}\right)\,.
\end{aligned}
\tag{17}
$$

Unfortunately, a longitudinal field term involving a single $\hat{\sigma}_j^x$ or $\hat{\sigma}_j^y$ *cannot* be translated into a simple local fermionic operator.

One important point to note concerns boundary conditions. One often assumes periodic boundary conditions (PBC) for the spin operators, which means that the model is defined on a *ring geometry* with $L$ sites, $j = 1,\cdots,L$, and the understanding that $\hat{\sigma}_0^\alpha \equiv \hat{\sigma}_L^\alpha$ and $\hat{\sigma}_{L+1}^\alpha \equiv \hat{\sigma}_1^\alpha$. This immediately implies the same PBC conditions for the hard-core bosons. Hence, for instance: $\hat{b}_L^\dagger \hat{b}_{L+1} \equiv \hat{b}_L^\dagger \hat{b}_1$. But observe what happens when we rewrite a term of this form using spinless fermions:

$$
\hat{b}_L^\dagger \hat{b}_1 = \left(\prod_{j=1}^{L-1} e^{i\pi\hat{n}_j}\right)\hat{c}_L^\dagger \hat{c}_1 = -\left(\prod_{j=1}^{L} e^{i\pi\hat{n}_j}\right)\hat{c}_L^\dagger \hat{c}_1 = -e^{i\pi\widehat{N}}\hat{c}_L^\dagger \hat{c}_1\,,
\tag{18}
$$

where

$$
\widehat{N} = \sum_{j=1}^{L} \hat{c}_j^\dagger \hat{c}_j\,,
\tag{19}
$$

is the total number of fermions, and the second equality follows because to the *left* of $\hat{c}_L^\dagger$ we certainly have $\hat{n}_L = 1$, and therefore $e^{i\pi\hat{n}_L} \equiv -1$. Similarly, you can verify that:

$$
\hat{b}_L^\dagger \hat{b}_1^\dagger = \left(\prod_{j=1}^{L-1} e^{i\pi\hat{n}_j}\right)\hat{c}_L^\dagger \hat{c}_1^\dagger = -\left(\prod_{j=1}^{L} e^{i\pi\hat{n}_j}\right)\hat{c}_L^\dagger \hat{c}_1^\dagger = -e^{i\pi\widehat{N}}\hat{c}_L^\dagger \hat{c}_1^\dagger\,.
\tag{20}
$$

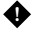 **Warning:** This shows that boundary conditions are affected by the *fermion parity* $e^{i\pi\widehat{N}} = (-1)^{\widehat{N}}$, and PBC become anti-periodic boundary condition (ABC) when $\widehat{N}$ is *even*. No problem whatsoever is present, instead, when the boundary conditions are *open* (OBC), because there is no link, in the Hamiltonian, between operators at site $L$ and operators at site $L + 1 \equiv 1$. More about this in a short while.

## 3 Transverse field Ising-XY models: Fermionic formulation

**Info:** There is a whole class of one-dimensional spin systems where a fermionic reformulation can be useful. Probably the most noteworthy is the XXZ Heisenberg chain, which would read:

$$\widehat{H}_{\text{XXZ}} = \sum_{j} \Big( J_j^{\perp} (\hat{\sigma}_j^x \hat{\sigma}_{j+1}^x + \hat{\sigma}_j^y \hat{\sigma}_{j+1}^y) + J_j^{zz} \hat{\sigma}_j^z \hat{\sigma}_{j+1}^z \Big) - \sum_j h_j \hat{\sigma}_j^z. \tag{21}$$

The corresponding fermionic formulation reads:

$$\widehat{H}_{\text{XXZ}} \to \sum_{j} \Big( 2J_j^{\perp} (\hat{c}_j^{\dagger} \hat{c}_{j+1} + \text{H.c.}) + J_j^{zz}(2\hat{n}_j - 1)(2\hat{n}_{j+1} - 1) \Big) + \sum_j h_j(2\hat{n}_j - 1), \tag{22}$$

which shows that the fermions interact at nearest-neighbours, due to the $J_j^{zz}$-term.

Let us now concentrate on a class of one-dimensional models where the resulting fermionic Hamiltonian can be exactly diagonalized, because it is quadratic in the fermions: such a class includes the XY model and the Ising model in a transverse field.

**Anisotropic XY model in a transverse field.** After a rotation in spin space, we can write the spin Hamiltonians leading to a quadratic fermion problem (allowing for non-uniform, possibly random, couplings) as follows:

$$\widehat{H} = -\sum_{j=1}^{L} \Big( J_j^x \hat{\sigma}_j^x \hat{\sigma}_{j+1}^x + J_j^y \hat{\sigma}_j^y \hat{\sigma}_{j+1}^y \Big) - \sum_{j=1}^{L} h_j \hat{\sigma}_j^z, \tag{23}$$

where $\hat{\sigma}_j^\alpha$ are Pauli matrices. The couplings $J_j^{x,y}$ and the transverse fields $h_j$ can be chosen, for instance, as independent random variables with uniform distribution. For a system of finite size $L$ with open boundary condition (OBC), the first sum runs over $j = 1, \cdots, L-1$, or, equivalently, we would set $J_L^{x,y} = 0$. If periodic boundary conditions (PBC) are chosen, the sum runs over $j = 1, \cdots, L$ and one assumes that $\hat{\sigma}_{L+1}^\alpha \equiv \hat{\sigma}_1^\alpha$. For $J_j^y = 0$ we have the Ising model in a transverse field, for $J_j^y = J_j^x$ the isotropic XY model in a transverse field.

In terms of hard-core bosons, the Hamiltonian becomes:

$$\widehat{H} = -\sum_{j=1}^{L} \Big( J_j^{+} \hat{b}_j^{\dagger} \hat{b}_{j+1} + J_j^{-} \hat{b}_j^{\dagger} \hat{b}_{j+1}^{\dagger} + \text{H.c.} \Big) + \sum_{j=1}^{L} h_j(2\hat{n}_j - 1), \tag{24}$$

where we have introduced a shorthand notation[6] $J_j^{\pm} = J_j^x \pm J_j^y$.

---

[6] This notation should not generate confusion with the angular momentum ladder operators. Here there is no imaginary unit $i$, and the couplings $J_j^{\pm} = J_j^x \pm J_j^y$ are just real numbers.

Next, we switch to spinless fermions, since all terms appearing in the previous expression do not involve explicitly the string operator $\hat{K}_j$. In terms of fermions, the Hamiltonian is essentially identical. We would remark that in the fermionic context, the pair creation and annihilation terms are characteristic of the BCS theory of superconductivity [64]. The only tricky point has to do with the boundary conditions. If one uses open boundary conditions, the first sum runs over $j = 1, \cdots, L-1$ and there is never a term involving site $L+1$, hence we have:

$$\widehat{H}_{\text{OBC}} = -\sum_{j=1}^{L-1}\left(J_j^+\hat{c}_j^\dagger\hat{c}_{j+1} + J_j^-\hat{c}_j^\dagger\hat{c}_{j+1}^\dagger + \text{H.c.}\right) + \sum_{j=1}^{L}h_j(2\hat{n}_j - 1). \tag{25}$$

In the PBC case, terms like $\hat{b}_L^\dagger\hat{b}_{L+1} \equiv \hat{b}_L^\dagger\hat{b}_1 = -\mathrm{e}^{i\pi\widehat{N}}\hat{c}_L^\dagger\hat{c}_1$ and $\hat{b}_L^\dagger\hat{b}_{L+1}^\dagger \equiv \hat{b}_L^\dagger\hat{b}_1^\dagger = -\mathrm{e}^{i\pi\widehat{N}}\hat{c}_L^\dagger\hat{c}_1^\dagger$ appear in the Hamiltonian, where $\widehat{N}$ is the number of fermions operator. Therefore:

$$\widehat{H}_{\text{PBC}} = \widehat{H}_{\text{OBC}} + \mathrm{e}^{i\pi\widehat{N}}\left(J_L^+\hat{c}_L^\dagger\hat{c}_1 + J_L^-\hat{c}_L^\dagger\hat{c}_1^\dagger + \text{H.c.}\right). \tag{26}$$

ⓘ **Info:** Notice that, although the number of fermions $\widehat{N}$ is *not conserved* by the Hamiltonian in Eq. (26), its parity $\mathrm{e}^{i\pi\widehat{N}} = (-1)^{\widehat{N}}$ is a "constant of motion" with value 1 or $-1$. So, from the fermionic perspective, it is *as if* we apply anti-periodic boundary conditions (ABC), hence $\hat{c}_{L+1} = -\hat{c}_1$, if there is an *even* number of fermions and periodic boundary condition (PBC), hence $\hat{c}_{L+1} = \hat{c}_1$, if there is an *odd* number of fermions. This symmetry can also be directly seen from the spin Hamiltonian in Eq. (23), where one should observe that the nearest-neighbour $\hat{\sigma}_j^x\hat{\sigma}_{j+1}^x$ and $\hat{\sigma}_j^y\hat{\sigma}_{j+1}^y$ can only flip *pairs* of spins, hence the parity of the overall magnetization along the $z$ direction is unchanged. Such a parity can be easily and equivalently expressed as:

$$\widehat{\mathcal{P}} = \prod_{j=1}^{L}\hat{\sigma}_j^z = \prod_{j=1}^{L}(1 - 2\hat{n}_j) = \prod_{j=1}^{L}\mathrm{e}^{i\pi\hat{n}_j} = \mathrm{e}^{i\pi\widehat{N}}. \tag{27}$$

We remark that $\widehat{\mathcal{P}}$ flips all the $\hat{\sigma}_j^x$ and $\hat{\sigma}_j^y$, i.e., $\widehat{\mathcal{P}}\hat{\sigma}_j^{x,y}\widehat{\mathcal{P}} = -\hat{\sigma}_j^{x,y}$, in the Hamiltonian in Eq. (23), leaving it invariant. This parity symmetry is the $\mathbb{Z}_2$-symmetry which the system breaks in the ordered ferromagnetic phase, as we will better discuss later on.

Let us define the projectors on the subspaces with even and odd number of particles:

$$\widehat{P}_{\text{even}} = \frac{1}{2}(\hat{1} + \mathrm{e}^{i\pi\widehat{N}}) = \widehat{P}_0, \qquad \text{and} \qquad \widehat{P}_{\text{odd}} = \frac{1}{2}(\hat{1} - \mathrm{e}^{i\pi\widehat{N}}) = \widehat{P}_1. \tag{28}$$

With these projectors, we can define two fermionic Hamiltonians acting on the $2^{L-1}$-dimensional even/odd parity subspaces of the full Hilbert space:

$$\widehat{H}_0 = \widehat{P}_0\widehat{H}_{\text{PBC}}\widehat{P}_0, \qquad \text{and} \qquad \widehat{H}_1 = \widehat{P}_1\widehat{H}_{\text{PBC}}\widehat{P}_1, \tag{29}$$

in terms of which we might express the full fermionic Hamiltonian in block form as:

$$\widehat{H}_{\text{PBC}} = \left(\begin{array}{c|c}\widehat{H}_0 & 0 \\ \hline 0 & \widehat{H}_1\end{array}\right). \tag{30}$$

Observe that if you write a fermionic Hamiltonian of the form:

$$\widehat{\mathbb{H}}_{\text{p}=0,1} = -\sum_{j=1}^{L-1}\left(J_j^+\hat{c}_j^\dagger\hat{c}_{j+1} + J_j^-\hat{c}_j^\dagger\hat{c}_{j+1}^\dagger + \text{H.c.}\right) + (-1)^{\text{p}}\left(J_L^+\hat{c}_L^\dagger\hat{c}_1 + J_L^-\hat{c}_L^\dagger\hat{c}_1^\dagger + \text{H.c.}\right)$$
$$+ \sum_{j=1}^{L}h_j(2\hat{n}_j - 1), \tag{31}$$

then you can regard $\widehat{\mathbb{H}}_1$ as a legitimate PBC-fermionic Hamiltonian since

$$\widehat{\mathbb{H}}_1 = -\sum_{j=1}^{L}\left(J_j^+ \hat{c}_j^\dagger \hat{c}_{j+1} + J_j^- \hat{c}_j^\dagger \hat{c}_{j+1}^\dagger + \text{H.c.}\right) + \sum_{j=1}^{L} h_j(2\hat{n}_j - 1), \tag{32}$$

with the interpretation $\hat{c}_{L+1} \equiv \hat{c}_1$. Similarly, $\widehat{\mathbb{H}}_0$ is a legitimate ABC-fermionic Hamiltonian where you should pose $\hat{c}_{L+1} \equiv -\hat{c}_1$. Neither of them, however, expresses the correct fermionic form of the PBC-spin Hamiltonian. However, they are useful in expressing the fermionic blocks:

$$\widehat{H}_0 = \widehat{P}_0 \widehat{\mathbb{H}}_0 \widehat{P}_0 = \widehat{\mathbb{H}}_0 \widehat{P}_0, \qquad \text{and} \qquad \widehat{H}_1 = \widehat{P}_1 \widehat{\mathbb{H}}_1 \widehat{P}_1 = \widehat{\mathbb{H}}_1 \widehat{P}_1, \tag{33}$$

since $\widehat{\mathbb{H}}_{p=0,1}$ conserve the fermionic parity, hence they commute with $\widehat{P}_{0,1}$.

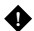

**Warning:** The distinction between $\widehat{H}_{0,1}$ and the corresponding $\widehat{\mathbb{H}}_{0,1}$ might appear pedantic, but is important, since the former acts non-trivially only on $2^{L-1}$-dimensional blocks, while the latter live in the full Hilbert space, hence are $2^L$-dimensional. This fact, for instance, complicates the calculation of thermal averages and is further discussed in Sec. 11.

**Info:** In the OBC case, since $J_L^\pm = 0$, the two fermionic Hamiltonians coincide and you can omit the label: $\widehat{\mathbb{H}}_0 = \widehat{\mathbb{H}}_1 \to \widehat{\mathbb{H}}$. Because of that, in the OBC case, you can simply set $\widehat{H}_{\text{OBC}} = \widehat{\mathbb{H}}$ and work with a single fermionic Hamiltonian.

## 4 Uniform XY-Ising model

As a warm-up, let us study the uniform case, where $J_j^x = J^x$, $J_j^y = J^y$, $h_j = h$, originally solved in Ref. [65]. It is customary to parameterise $J^x = J(1+\kappa)/2$ and $J^y = J(1-\kappa)/2$, so that $J^+ = J$ and $J^- = \kappa J$. The Hamiltonian is then:[7]

$$\widehat{H}_{\text{OBC}} = -J \sum_{j=1}^{L-1}\left(\hat{c}_j^\dagger \hat{c}_{j+1} + \kappa \hat{c}_j^\dagger \hat{c}_{j+1}^\dagger + \text{H.c.}\right) + h \sum_{j=1}^{L}(2\hat{c}_j^\dagger \hat{c}_j - 1), \tag{34}$$

for the OBC case, and:

$$\widehat{H}_{\text{PBC}} = \widehat{H}_{\text{OBC}} + e^{i\pi\widehat{N}} J\left(\hat{c}_L^\dagger \hat{c}_1 + \kappa \hat{c}_L^\dagger \hat{c}_1^\dagger + \text{H.c.}\right). \tag{35}$$

We assume from now on that the number of sites $L$ is *even*: This is not a big restriction and is useful.

In the spin-PBC case, if the number of fermions $\widehat{N}$ takes an odd value, then we effectively have $\hat{c}_{L+1} \equiv \hat{c}_1$; if, on the contrary, $\widehat{N}$ takes an even value, then the $L$-th bond has an opposite sign to the remaining ones, which can also be reformulated as $\hat{c}_{L+1} \equiv -\hat{c}_1$. Since the

---

[7]Notice that one can change the sign of the $h$-term by making a particle-hole transformation $\tilde{c}_j \to (-1)^j \hat{c}_j^\dagger$, which transforms $\tilde{n}_j \to 1 - \hat{n}_j$, and $1 - 2\tilde{n}_j \to 2\hat{n}_j - 1$, while leaving the hopping term untouched (same sign of $J$). With the current choice of the $h$-term, the $h \to +\infty$ ground state in the spin representation $|\uparrow\uparrow\cdots\uparrow\rangle$ is mapped into the fermionic vacuum, which will be useful in discussing the ground state. (Notice that the phase factor $(-1)^j$ exchange the roles of $k = 0$ and $k = \pi$ in the discussion of the ground state.) Similarly, the same particle-hole transformation but without phase factor $(-1)^j$ would also invert the sign of the $J$-term, from ferromagnetic to antiferromagnetic.

Hamiltonian conserves the fermion parity, both the even and the odd particle subsectors of the fermionic Hilbert space have to be considered when diagonalizing the model, precisely as in the general case of Eq. (30). Introducing the two fermionic Hamiltonians as in Eq. (31) we now have:

$$\widehat{\mathbb{H}}_{\mathrm{p}=0,1} = -J \sum_{j=1}^{L} \left( \hat{c}_j^\dagger \hat{c}_{j+1} + \kappa \hat{c}_j^\dagger \hat{c}_{j+1}^\dagger + \mathrm{H.c.} \right) + h \sum_{j=1}^{L} (2\hat{n}_j - 1), \qquad (36)$$

where we recall that $\mathrm{p} = 0, 1$ is associated with the fermionic parity — $\mathrm{p} = 0$ for even and $\mathrm{p} = 1$ for odd parity — and that this compact way of writing assumes that the boundary terms are treated with:

$$\hat{c}_{L+1} \equiv (-1)^{\mathrm{p}+1} \hat{c}_1. \qquad (37)$$

Let us now introduce the fermion operators in $k$-space, $\hat{c}_k$ and $\hat{c}_k^\dagger$, with $\{\hat{c}_k, \hat{c}_{k'}^\dagger\} = \delta_{k,k'}$. The direct and inverse transformations are defined as follows:

$$\begin{cases} \hat{c}_k = \dfrac{e^{-i\phi}}{\sqrt{L}} \sum_{j=1}^{L} e^{-ikj} \hat{c}_j, \\[2mm] \hat{c}_j = \dfrac{e^{i\phi}}{\sqrt{L}} \sum_{k} e^{+ikj} \hat{c}_k, \end{cases} \qquad (38)$$

where the overall phase $e^{i\phi}$ does not affect the canonical anti-commutation relations, but might be useful to change the phase of the anomalous BCS pair-creation terms (see below). Which values of $k$ should be used in the previous transformation *depends* on p. For $\mathrm{p} = 1$ we have $\hat{c}_{L+1} \equiv \hat{c}_1$, which in turn implies, from the expression for $\hat{c}_j$ in terms of $\hat{c}_k$, that $e^{ikL} = 1$, hence the standard PBC choice for the $k$'s:

$$\mathrm{p} = 1 \implies \mathcal{K}_{\mathrm{p}=1} = \left\{ k = \frac{2n\pi}{L}, \text{ with } n = -\frac{L}{2} + 1, \cdots, 0, \cdots, \frac{L}{2} \right\}. \qquad (39)$$

For $\mathrm{p} = 0$ we have $\hat{c}_{L+1} \equiv -\hat{c}_1$, which implies that $e^{ikL} = -1$, hence an anti-periodic boundary conditions (ABC) choice for the $k$'s:

$$\mathrm{p} = 0 \implies \mathcal{K}_{\mathrm{p}=0} = \left\{ k = \pm \frac{(2n-1)\pi}{L}, \text{ with } n = 1, \cdots, \frac{L}{2} \right\}. \qquad (40)$$

In terms of $\hat{c}_k$ and $\hat{c}_k^\dagger$, with the appropriate choice of the $k$-vectors, $\widehat{\mathbb{H}}_{\mathrm{p}}$ becomes:[8]

$$\widehat{\mathbb{H}}_{\mathrm{p}} = -J \sum_{k \in \mathcal{K}_{\mathrm{p}}} \left( 2\cos k \, \hat{c}_k^\dagger \hat{c}_k + \kappa \left( e^{-2i\phi} e^{ik} \hat{c}_k^\dagger \hat{c}_{-k}^\dagger + \mathrm{H.c.} \right) \right) + h \sum_{k \in \mathcal{K}_{\mathrm{p}}} (2\hat{c}_k^\dagger \hat{c}_k - 1). \qquad (41)$$

Notice the coupling of $-k$ with $k$ in the (anomalous) pair-creation term, with the exceptions of $k = 0$ and $k = \pi$ for the $\mathrm{p} = 1$ (PBC) case, which do not have a separate $-k$ partner. It is

---

[8]We use the standard fact that the sum over $j$ introduces a Krönecker delta for the wave-vectors:

$$\frac{1}{L} \sum_{j=1}^{L} e^{-i(k-k')j} = \delta_{k,k'}.$$

useful to manipulate the (normal) number-conserving terms[9] to rewrite the Hamiltonian as:

$$\widehat{\mathbb{H}}_p = \sum_{k \in \mathcal{K}_p} \left( (h - J\cos k)\left(\hat{c}_k^\dagger \hat{c}_k - \hat{c}_{-k} \hat{c}_{-k}^\dagger\right) - \kappa J\left(e^{-2i\phi} e^{ik} \hat{c}_k^\dagger \hat{c}_{-k}^\dagger + \text{H.c.}\right) \right). \tag{42}$$

The two terms with $k = 0$ and $k = \pi$, present for $p = 1$ (PBC), taken together can be written as:

$$\widehat{H}_{0 \& \pi} = -2J(\hat{n}_0 - \hat{n}_\pi) + 2h(\hat{n}_0 + \hat{n}_\pi - 1). \tag{43}$$

The remaining $p = 1$ terms, and all terms for $p = 0$, come into pairs $(k, -k)$. Let us define the *positive $k$* values as follows:

$$\begin{aligned}
\mathcal{K}_{p=1}^+ &= \left\{ k = \frac{2n\pi}{L}, \text{ with } n = 1, \cdots, \tfrac{L}{2} - 1 \right\}, \\
\mathcal{K}_{p=0}^+ &= \left\{ k = \frac{(2n-1)\pi}{L}, \text{ with } n = 1, \cdots, \tfrac{L}{2} \right\}.
\end{aligned} \tag{44}$$

Then we can write the Hamiltonians as:

$$\widehat{\mathbb{H}}_0 = \sum_{k \in \mathcal{K}_0^+} \widehat{H}_k, \qquad\qquad \widehat{\mathbb{H}}_1 = \widehat{H}_{0 \& \pi} + \sum_{k \in \mathcal{K}_1^+} \widehat{H}_k, \tag{45}$$

where we have grouped terms with $k$ and $-k$ into a single Hamiltonian $\widehat{H}_k$ of the form:

$$\widehat{H}_k = 2(h - J\cos k)\left(\hat{c}_k^\dagger \hat{c}_k - \hat{c}_{-k} \hat{c}_{-k}^\dagger\right) - 2\kappa J \sin k\left(i e^{-2i\phi} \hat{c}_k^\dagger \hat{c}_{-k}^\dagger - i e^{2i\phi} \hat{c}_{-k} \hat{c}_k\right). \tag{46}$$

Interestingly, the Hamiltonians $\widehat{H}_k$ commute for different $k$, $[\widehat{H}_k, \widehat{H}_{k'}] = 0$, and act non-trivially only in the 4-dimensional space generated by the states:

$$\left\{ \hat{c}_k^\dagger \hat{c}_{-k}^\dagger |0\rangle, |0\rangle, \hat{c}_k^\dagger |0\rangle, \hat{c}_{-k}^\dagger |0\rangle \right\}, \tag{47}$$

where they have a $4 \times 4$ matrix of the form:

$$\left( \begin{array}{cc|cc}
2(h - J\cos k) & -2i\kappa J e^{-2i\phi} \sin k & 0 & 0 \\
2i\kappa J e^{2i\phi} \sin k & -2(h - J\cos k) & 0 & 0 \\
\hline
0 & 0 & 0 & 0 \\
0 & 0 & 0 & 0
\end{array} \right). \tag{48}$$

ⓘ

**Check of dimensions.** Recall that both $\widehat{\mathbb{H}}_{p=0,1}$ have $2^L$ eigenvalues (including multiplicity). Indeed, there are $\frac{L}{2}$ such terms for $\widehat{\mathbb{H}}_0$, hence a dimension $4^{\frac{L}{2}} = 2^L$. Notice that $\widehat{H}_{k=0,\pi}$ also works in a 4-dimensional subspace,

$$\left\{ |0\rangle, \hat{c}_0^\dagger \hat{c}_\pi^\dagger |0\rangle, \hat{c}_0^\dagger |0\rangle, \hat{c}_\pi^\dagger |0\rangle \right\}, \tag{49}$$

and there are $\frac{L}{2} - 1$ wave-vectors in $\mathcal{K}_1^+$, hence again a total dimension for $\widehat{\mathbb{H}}_1$ of $4^{\frac{L}{2}-1} 4 = 2^L$. Recall, finally, that the correct eigenvalues are obtained from the block Hamiltonians $\widehat{\mathbb{H}}_{p=0,1}$ which have $2^{L-1}$ eigenvalues each (including multiplicity), those with even ($p = 0$) or odd ($p = 1$) fermion parity.

---

[9]We use that

$$\sum_k 2\cos k\, \hat{c}_k^\dagger \hat{c}_k = \sum_k \cos k \left(\hat{c}_k^\dagger \hat{c}_k - \hat{c}_{-k} \hat{c}_{-k}^\dagger\right),$$

where we used the anti-commutation relations, $\sum_k \cos k = 0$, and

$$\sum_k (2\hat{c}_k^\dagger \hat{c}_k - 1) = \sum_k \left(\hat{c}_k^\dagger \hat{c}_k - \hat{c}_{-k} \hat{c}_{-k}^\dagger\right).$$

To deal with the necessary combination of states $\{\hat{c}_k^\dagger \hat{c}_{-k}^\dagger |0\rangle, |0\rangle\}$ involved in the non-trivial $2 \times 2$ blocks of the Hamiltonian, requiring essentially a Bogoliubov transformation, we now define a fermionic two-component spinor

$$\widehat{\Psi}_k = \begin{pmatrix} \hat{c}_k \\ \hat{c}_{-k}^\dagger \end{pmatrix}, \qquad \widehat{\Psi}_k^\dagger = (\hat{c}_k^\dagger, \hat{c}_{-k}), \qquad (50)$$

with anti-commutation relations ($\alpha = 1, 2$ stands for the two components of $\widehat{\Psi}$)

$$\{\widehat{\Psi}_{k\alpha}, \widehat{\Psi}_{k'\alpha'}^\dagger\} = \delta_{\alpha,\alpha'}\delta_{k,k'}. \qquad (51)$$

We can then rewrite each $\widehat{H}_k$ as:

$$\widehat{H}_k = \sum_{\alpha,\alpha'} \widehat{\Psi}_{k\alpha}^\dagger (\mathbf{H}_k)_{\alpha\alpha'}\widehat{\Psi}_{k\alpha'} = (\hat{c}_k^\dagger, \hat{c}_{-k})\underbrace{\begin{pmatrix} 2(h - J\cos k) & -2\kappa J i e^{-2i\phi}\sin k \\ 2\kappa J i e^{2i\phi}\sin k & -2(h - J\cos k) \end{pmatrix}}_{\mathbf{H}_k}\begin{pmatrix} \hat{c}_k \\ \hat{c}_{-k}^\dagger \end{pmatrix}, \quad (52)$$

where we have highlighted a $2 \times 2$ Hermitian matrix $\mathbf{H}_k$ which can be expressed in terms of new pseudo-spin Pauli matrices $\hat{\tau}^{x,y,z}$ as:

$$\mathbf{H}_k = \mathbf{R}_k \cdot \hat{\boldsymbol{\tau}}. \qquad (53)$$

Here we recognise an "effective magnetic field" $\mathbf{R}_k$ given by:

$$\mathbf{R}_k = 2\big(-\kappa J \sin 2\phi \, \sin k, \, \kappa J \cos 2\phi \, \sin k, \, (h - J\cos k)\big)^{\mathrm{T}}. \qquad (54)$$

> **ⓘ**
> **Info:** Observe the role of the arbitrary phase $\phi$ introduced in the transformation from real space to momentum space, Eq. (38). For $\phi = 0$ the effective magnetic field lives in the $y - z$ plane in pseudo-spin space, while for $\phi = \frac{\pi}{4}$ it lives in the $x - z$ plane and the pseudo-spin Hamiltonian is real, as it involves $\hat{\tau}^x$ and $\hat{\tau}^z$.

By solving the $2 \times 2$ eigenvalue problem for the pseudo-spin Hamiltonian $\mathbf{H}_k$ we find the eigenvalues $\epsilon_{k\pm} = \pm\epsilon_k$ with:

$$\epsilon_k = |\mathbf{R}_k| = 2J\sqrt{\left(\frac{h}{J} - \cos k\right)^2 + \kappa^2 \sin^2 k} \geq 0, \qquad (55)$$

with corresponding eigenvectors $(v_{k\pm}, u_{k\pm})^{\mathrm{T}}$ which can be expressed in terms of spin eigenstates in the direction $\mathbf{R}_k/|\mathbf{R}_k|$. From now on we will fix $\phi = 0$, so that the pseudo-spin effective magnetic field lives in the $y - z$ plane. Define the shorthand $\mathbf{R}_k = (0, y_k, z_k)^{\mathrm{T}}$ with $z_k = 2(h - J\cos k)$ and $y_k = 2\kappa J \sin k$. For the negative energy eigenvector, we have:

$$\begin{pmatrix} v_{k-} \\ u_{k-} \end{pmatrix} \equiv \begin{pmatrix} v_k \\ u_k \end{pmatrix} = \frac{1}{\sqrt{2\epsilon_k(\epsilon_k + z_k)}}\begin{pmatrix} iy_k \\ \epsilon_k + z_k \end{pmatrix}, \qquad (56)$$

where we have introduced the shorthands $v_k = v_{k-}$ and $u_k = u_{k-}$. Note, in passing, that $u_{-k} = u_k$, while $v_{-k} = -v_k$, since $z_k$ is even in $k$, while $y_k$ is odd. The positive-energy eigenvector $(v_{k+}, u_{k+})^{\mathrm{T}}$ is related to the previous one by a simple transformation:[10]

$$\begin{pmatrix} v_{k+} \\ u_{k+} \end{pmatrix} = \begin{pmatrix} u_k^* \\ -v_k^* \end{pmatrix} = \frac{1}{\sqrt{2\epsilon_k(\epsilon_k + z_k)}}\begin{pmatrix} \epsilon_k + z_k \\ iy_k \end{pmatrix}. \qquad (59)$$

---

[10]Indeed, write the eigenvalue problem for $(v_k, u_k)^{\mathrm{T}}$, with energy $\epsilon_{k-} = -\epsilon_k$:

$$\begin{cases} z_k v_k - iy_k u_k = -\epsilon_k v_k, \\ iy_k v_k - z_k u_k = -\epsilon_k u_k. \end{cases} \qquad (57)$$

The unitary matrix $\mathbf{U}_k$ having the two previous eigenvectors as columns:

$$\mathbf{U}_k = \begin{pmatrix} u_k^* & v_k \\ -v_k^* & u_k \end{pmatrix}, \tag{60}$$

diagonalizes $\mathbf{H}_k$:

$$\mathbf{U}_k^\dagger \, \mathbf{H}_k \, \mathbf{U}_k = \begin{pmatrix} \epsilon_k & 0 \\ 0 & -\epsilon_k \end{pmatrix}. \tag{61}$$

So, define new fermion two-component operators $\widehat{\Phi}_k$ through

$$\begin{pmatrix} \hat{\gamma}_k \\ \hat{\gamma}_{-k}^\dagger \end{pmatrix} \overset{\text{def}}{=} \widehat{\Phi}_k = \mathbf{U}_k^\dagger \widehat{\Psi}_k = \begin{pmatrix} u_k \hat{c}_k - v_k \hat{c}_{-k}^\dagger \\ v_k^* \hat{c}_k + u_k^* \hat{c}_{-k}^\dagger \end{pmatrix}. \tag{62}$$

It is straightforward to verify that $\hat{\gamma}_k$ is indeed a fermion.[11] In terms of $\widehat{\Phi}_k = (\hat{\gamma}_k, \hat{\gamma}_{-k}^\dagger)^\mathsf{T}$ and $\widehat{\Phi}_k^\dagger = \widehat{\Psi}_k^\dagger \mathbf{U}_k = (\hat{\gamma}_k^\dagger, \hat{\gamma}_{-k})$, we have:

$$\widehat{H}_k = \widehat{\Psi}_k^\dagger \mathbf{U}_k \mathbf{U}_k^\dagger \mathbf{H}_k \mathbf{U}_k \mathbf{U}_k^\dagger \widehat{\Psi}_k = \widehat{\Phi}_k^\dagger \begin{pmatrix} \epsilon_k & 0 \\ 0 & -\epsilon_k \end{pmatrix} \widehat{\Phi}_k = \epsilon_k \left( \hat{\gamma}_k^\dagger \hat{\gamma}_k - \hat{\gamma}_{-k} \hat{\gamma}_{-k}^\dagger \right)$$

$$= \epsilon_k \left( \hat{\gamma}_k^\dagger \hat{\gamma}_k + \hat{\gamma}_{-k}^\dagger \hat{\gamma}_{-k} - 1 \right). \tag{64}$$

The form of the two bands $\pm \epsilon_k$, as a function of $k$ and for several values of $h$ is noteworthy. Figure 2 shows plots that illustrate them.

---

Now change the sign of the second equation, take the complex-conjugate of both, and rewrite them in inverted order, to get:

$$\begin{cases} z_k(u_k^*) - i y_k(-v_k^*) = \epsilon_k(u_k^*), \\ i y_k(u_k^*) - z_k(-v_k^*) = \epsilon_k(-v_k^*), \end{cases} \tag{58}$$

which is the eigenvalue equation for $(v_{k+}, u_{k+})^\mathsf{T}$.

[11]One can verify anti-commutation relationships very easily:

$$\{\hat{\gamma}_k, \hat{\gamma}_k^\dagger\} = \{u_k \hat{c}_k - v_k \hat{c}_{-k}^\dagger, u_k^* \hat{c}_k^\dagger - v_k^* \hat{c}_{-k}\}$$

$$= |u_k|^2 \{\hat{c}_k, \hat{c}_k^\dagger\} + |v_k|^2 \{\hat{c}_{-k}^\dagger, \hat{c}_{-k}\} = |u_k|^2 + |v_k|^2 = 1, \tag{63}$$

where the last equality follows from the normalization condition for the eigenvectors.

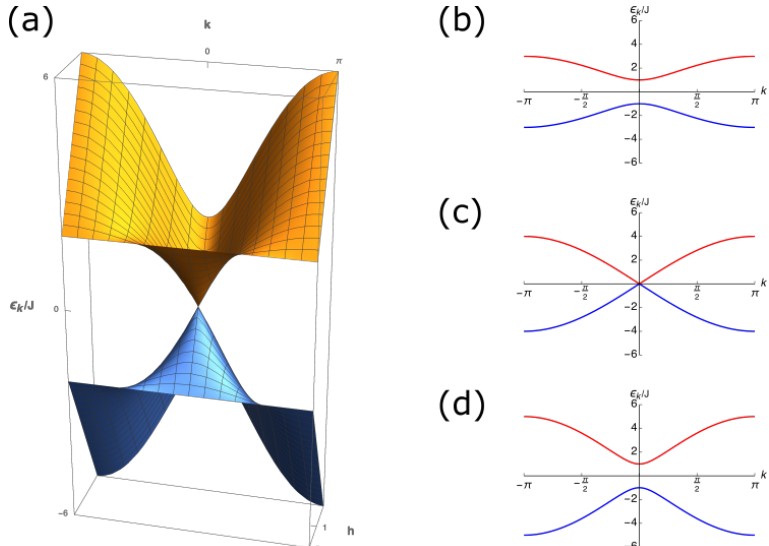

Figure 2: (a) The two bands $\pm\epsilon_k$ plotted by varying the transverse field $h$ in the range $[0, 2]$. (b-d) The bands $\pm\epsilon_k$ for three different transverse fields $h$: (b) $h/J = 0.5$ (inside the ferromagnetic region), (c) $h/J = 1$ (the critical point), (d) $h/J = 1.5$ (inside the paramagnetic phase). Notice the remarkable behaviour at $h = h_c = J$, clearly visible in panel (c): A gapless linear spectrum. Notice also how you can hardly distinguish the bands of the two gapped phases in (b) and (d). But their topology is distinctly different: see the discussion related to Fig. 3. Here $J = 1$ and $\kappa = 1$.

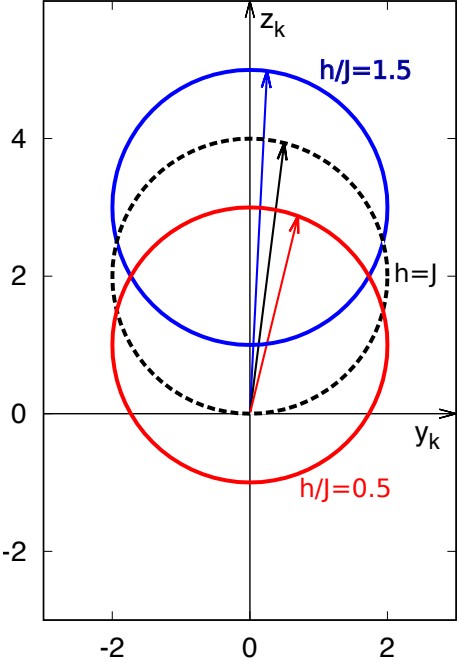

Figure 3: Curves drawn by the vector $\mathbf{R}_k$ as $k$ spans $[-\pi, \pi)$, for three values of $h$. Here $J = 1$, $\kappa = 1$.

**Winding and topology.** It is instructive to trace the behaviour of the "effective magnetic field" $\mathbf{R}_k$, of magnitude $|\mathbf{R}_k| = \epsilon_k$, that the system "sees" as the wave-vector $k$ spans the so-called Brillouin zone $[-\pi, \pi]$. Fixing $\phi = 0$ in Eq. (54), this effective magnetic field lies in the $y-z$ plane, $\mathbf{R}_k = (0, y_k, z_k)^{\mathrm{T}}$ with $y_k = 2\kappa J \sin k$ and $z_k = 2(h - J \cos k)$, where it draws the ellipse of equation[12]

$$\frac{y_k^2}{4\kappa^2 J^2} + \frac{(z_k - 2h)^2}{4J^2} = 1\,, \tag{65}$$

as $k$ spans the interval $[-\pi, \pi]$. We show in Fig. 3 three examples of this ellipse (circles, for $\kappa = 1$), one for $|h| < J$, one for $h > J$, and that for $h = J$. For $|h| < J$ we see that the vector $\mathbf{R}_k$ turns around and comes back to its original position, making one complete revolution around the origin, as $k$ varies in $[-\pi, \pi]$. We term the number of revolutions as the *index* [66] (or winding number) of the vector, and here it equals 1. As we change $h$ in the range $-J < h < J$, the index, for continuity reasons, keeps the constant value 1 (it can only assume discrete values). In the case $h > J$, the vector $\mathbf{R}_k$ makes no revolution around the origin and its index is 0: It keeps this value for any $h > J$, for the same continuity argument as before. The transition of the index between the two values 1 and 0 occurs at $h = J$. At that point, the continuity of the index as a function of the curve is broken, because the index is not defined for $h = J$, as the curve passes through the origin for $k = 0$. The index is a topological quantity, invariant under continuous transformations. Because it takes different values for $|h| < J$ and $h > J$ we say that these two phases have different topologies. We see that $\mathbf{R}_k = \mathbf{0}$ corresponds to a degeneracy point of the $2 \times 2$ Hamiltonian $\mathbf{H}_k$ — realised for $k = 0$ and $h = J$ (but also for $k = \pi$ and $h = -J$) — and the discontinuity of the index corresponds to the closing of the gap in the single-quasiparticle spectrum shown in Fig. 2(c).

## 4.1 Ground state and excited states of the uniform XY-Ising model

The expression $\mathbf{H}_k = \epsilon_k\big(\hat{\gamma}_k^\dagger \hat{\gamma}_k + \hat{\gamma}_{-k}^\dagger \hat{\gamma}_{-k} - 1\big)$ in Eq. (64), together with the expression for $\epsilon_k \geq 0$ in Eq. (55), allows to immediately conclude that the ground state of the Hamiltonian must be the state $|\emptyset_\gamma\rangle$ which annihilates the $\hat{\gamma}_k$ for all $k$, positive and negative, the so-called *Bogoliubov vacuum*:

$$\hat{\gamma}_k |\emptyset_\gamma\rangle = 0\,, \qquad \forall\, k\,. \tag{66}$$

In principle, one can define two such states, one in the p = 0 (even, ABC) sector, and one in the p = 1 (odd, PBC) sector. However, one finds that the winner between the two, i.e., the actual global ground state, is the one in the p = 0 (even) sector, with an energy

$$E_0^{\mathrm{ABC}} = -\sum_{k>0}^{\mathrm{ABC}} \epsilon_k\,. \tag{67}$$

The ground state can be written explicitly as:

$$|\emptyset_\gamma\rangle^{\mathrm{ABC}} \propto \prod_{k>0}^{\mathrm{ABC}} \hat{\gamma}_{-k}\hat{\gamma}_k |0\rangle\,, \tag{68}$$

---

[12] The ellipse degenerates into a segment for $\kappa = 0$, corresponding to the isotropic XY model. Hence, our argument requires $\kappa \neq 0$.

where $|0\rangle$ is the vacuum for the original fermions, $\hat{c}_k|0\rangle = 0$. The explicit calculation shows that:

$$\prod_{k>0}\hat{\gamma}_{-k}\hat{\gamma}_k|0\rangle = \prod_{k>0}\left(u_{-k}\hat{c}_{-k} - v_{-k}\hat{c}_k^\dagger\right)\left(u_k\hat{c}_k - v_k\hat{c}_{-k}^\dagger\right)|0\rangle$$
$$= \prod_{k>0}(-v_k)\left(u_k + v_k\hat{c}_k^\dagger\hat{c}_{-k}^\dagger\right)|0\rangle, \tag{69}$$

where we used that $u_{-k} = u_k$ and $v_{-k} = -v_k$. By normalising the state, we arrive at the BCS form:

$$|\emptyset_\gamma\rangle^{\text{ABC}} = \prod_{k>0}^{\text{ABC}}\left(u_k + v_k\hat{c}_k^\dagger\hat{c}_{-k}^\dagger\right)|0\rangle. \tag{70}$$

The PBC-sector ground state must contain an *odd* number of particles. Since a BCS-paired state is always fermion-even, the unpaired Hamiltonian terms $\widehat{H}_{0\&\pi}$ must contribute with exactly *one* fermion in the ground state. It is simple to verify that, with our choice of the sign of $h > 0$, the ground state has $\hat{n}_{k=0} \to 1$ and $\hat{n}_{k=\pi} \to 0$, resulting in an extra term of the form

$$\delta E_{0\&\pi} = \min(\widehat{H}_{0\&\pi}) = -2J. \tag{71}$$

The PBC-ground state is, therefore:

$$|\emptyset_\gamma\rangle^{\text{PBC}} = \hat{c}_{k=0}^\dagger \prod_{0<k<\pi}^{\text{PBC}}\left(u_k + v_k\hat{c}_k^\dagger\hat{c}_{-k}^\dagger\right)|0\rangle = \hat{\gamma}_0\prod_{0<k<\pi}^{\text{PBC}}\left(u_k + v_k\hat{c}_k^\dagger\hat{c}_{-k}^\dagger\right)|0\rangle, \tag{72}$$

where we defined $\hat{\gamma}_0 = \hat{c}_0^\dagger$ and $\hat{\gamma}_\pi = \hat{c}_\pi$ for the unpaired states. The corresponding energy is:

$$E_0^{\text{PBC}} = -2J - \sum_{0<k<\pi}^{\text{PBC}} \epsilon_k. \tag{73}$$

And here comes an amusing subtlety of the *thermodynamic limit* $L \to \infty$. When you consider the *energy-per-site* $e_0 = E_0/L$, then the ground state energy should simply tend to *an integral*:

$$e_0 = -\lim_{L\to\infty}\frac{1}{L}\sum_{k>0}^{\text{ABC}}\epsilon_k = -\int_0^\pi\frac{\mathrm{d}k}{2\pi}\,\epsilon_k. \tag{74}$$

The same integral appearing in Eq. (74) gives the ground state energy-per-site in the PBC sector, but an amusing role is played by the two boundary points at 0 and $\pi$, when one considers the energy splitting $\Delta E_0 = E_0^{\text{PBC}} - E_0^{\text{ABC}}$. Notice in particular that Eq. (67) for $E_0^{\text{ABC}}$ involves $L/2$ $k$-points in the interval $(0, \pi)$, while Eq. (73) for $E_0^{\text{PBC}}$ involves $L/2 - 1$ points in the interval $(0, \pi)$ and an extra term $-2J$. In the thermodynamic limit $L \to \infty$, you discover that the energy splitting $\Delta E_0 = E_0^{\text{PBC}} - E_0^{\text{ABC}}$ is, in the whole ferromagnetically ordered region $-J < h < J$, a quantity that goes to zero *exponentially fast* when $L \to \infty$: In other words, the two sectors, ABC and PBC, provide the required *double degeneracy* of the ferromagnetic phase: You can see that easily for $h = 0$. Less trivial, but true, for all $|h| < J$. On the contrary, $\Delta E_0$ is *finite*[13] in the quantum disordered regions $|h| > J$, $\Delta E_0 = 2(|h| - J)$, and goes to zero as a power-law, more precisely as $\pi/(2L)$, at the critical points $h_c = \pm J$. In Fig. 4 we illustrate these facts by numerically evaluating $\Delta E_0$.

Regarding the excited states, let us start from the $p = 0$ (even, ABC) sector. Consider, as a warm-up, the state $\hat{\gamma}_{k_1}^\dagger|\emptyset_\gamma\rangle^{\text{ABC}}$. A simple calculation shows that, *regardless of the sign* of $k_1$:

$$\hat{\gamma}_{k_1}^\dagger|\emptyset_\gamma\rangle^{\text{ABC}} = \hat{c}_{k_1}^\dagger\prod_{\substack{k>0 \\ k\neq|k_1|}}^{\text{ABC}}\left(u_k + v_k\hat{c}_k^\dagger\hat{c}_{-k}^\dagger\right)|0\rangle. \tag{75}$$

---

[13]Again, the convergence in $L$ to such finite value is exponentially fast in the whole quantum disordered region.

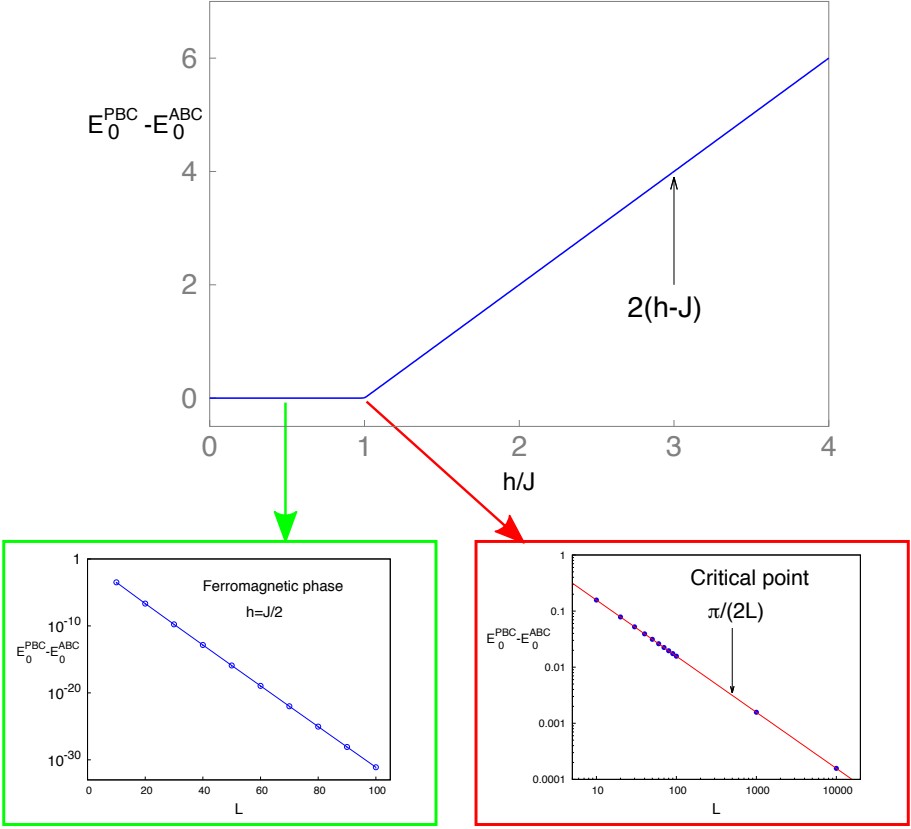

Figure 4: The gap between the ground state in the PBC and ABC sectors versus the transverse field $h/J$. The two lower insets illustrate the exponential drop to 0 of the gap in the ferromagnetic region (left), and the power-law behaviour at the critical point (right). Here $\kappa = 1$.

In essence, the application of $\hat{\gamma}_{k_1}^\dagger$ transforms the Cooper-pair at momentum $(|k_1|, -|k_1|)$ into an unpaired fermion in the state $\hat{c}_{k_1}^\dagger |0\rangle$. This would cost an extra energy $+\epsilon_{k_1}$ over the ground state: the gain $-\epsilon_{k_1}$ obtained from pairing is indeed transformed into a no-gain (energy 0) for the unpaired state $\hat{c}_{k_1}^\dagger |0\rangle$, consistently with the $4 \times 4$ structure of Eq. (48) predicting two eigenvalues 0 for the unpaired states. There is a problem with parity, however: A single unpaired fermion changes the overall fermion parity of the state. Hence, the lowest allowed states must involve *two* creation operators, $\hat{\gamma}_{k_1}^\dagger \hat{\gamma}_{k_2}^\dagger$ with $k_1 \neq |k_2|$:

$$\hat{\gamma}_{k_1}^\dagger \hat{\gamma}_{k_2}^\dagger |\emptyset_\gamma\rangle^{\text{ABC}} = \hat{c}_{k_1}^\dagger \hat{c}_{k_2}^\dagger \prod_{\substack{k>0 \\ k\neq|k_1|,|k_2|}}^{\text{ABC}} \left(u_k + v_k \hat{c}_k^\dagger \hat{c}_{-k}^\dagger\right)|0\rangle. \tag{76}$$

The energy of such excitation is $E_0^{\text{ABC}} + \epsilon_{k_1} + \epsilon_{k_2}$ because we loose two Cooper pairs. Quite amusingly, if you consider the special case $\hat{\gamma}_{k_1}^\dagger \hat{\gamma}_{-k_1}^\dagger$ you find that:

$$\hat{\gamma}_{k_1}^\dagger \hat{\gamma}_{-k_1}^\dagger |\emptyset_\gamma\rangle^{\text{ABC}} = \left(-v_{k_1}^* + u_{k_1}^* \hat{c}_{k_1}^\dagger \hat{c}_{-k_1}^\dagger\right) \prod_{\substack{k>0 \\ k\neq|k_1|}}^{\text{ABC}} \left(u_k + v_k \hat{c}_k^\dagger \hat{c}_{-k}^\dagger\right)|0\rangle. \tag{77}$$

This means that $\hat{\gamma}_{k_1}^\dagger \hat{\gamma}_{-k_1}^\dagger$ transforms the Cooper pair at momentum $(|k_1|, -|k_1|)$ into the corre-

sponding anti-bonding pair:

$$\left(u_{k_1} + v_{k_1}\hat{c}^\dagger_{k_1}\hat{c}^\dagger_{-k_1}\right)|0\rangle \xrightarrow{\hat{\gamma}^\dagger_{k_1}\hat{\gamma}^\dagger_{-k_1}} \left(-v^*_{k_1} + u^*_{k_1}\hat{c}^\dagger_{k_1}\hat{c}^\dagger_{-k_1}\right)|0\rangle.$$

This costs an energy $2\epsilon_{k_1}$, consistent with the previous expression $E^{\text{ABC}}_0 + \epsilon_{k_1} + \epsilon_{k_2}$, if you consider that $\epsilon_{-k_1} = \epsilon_{k_1}$.

Generalising, we can construct all the excited states in the even-fermion sector, by applying an *even* number of $\hat{\gamma}^\dagger_k$ to $|\emptyset_\gamma\rangle^{\text{ABC}}$, each $\hat{\gamma}^\dagger_k$ costing an energy $\epsilon_k$. In the occupation number (Fock) representation we have, therefore:

$$|\psi_{\{n_k\}}\rangle = \prod_k^{\text{ABC}}\left(\hat{\gamma}^\dagger_k\right)^{n_k}|\emptyset_\gamma\rangle^{\text{ABC}}, \quad \text{with} \quad n_k = 0, 1 \ \& \ \sum_k^{\text{ABC}} n_k = \text{even},$$

$$E_{\{n_k\}} = E^{\text{ABC}}_0 + \sum_k^{\text{ABC}} n_k\epsilon_k. \tag{78}$$

We see that there are $2^{L-1}$ such states, as required.

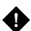

**Remark:** An important remark and check is here in order. First: The counting of the excitation number is correct if the $k$ in Eq. (78) are allowed to range among the *L positive and negative* wave-vectors allowed by ABC: $2^L$ Fock states if the parity check is not enforced, $2^{L-1}$ if we enforce parity. Second: recall that we can transform a Cooper pair of energy $-\epsilon_k$ into the corresponding anti-paired state, of energy $+\epsilon_k$. The state that realises that is $\hat{\gamma}^\dagger_k\hat{\gamma}^\dagger_{-k}|\emptyset\rangle^{\text{ABC}}$. Its energy is $2\epsilon_k$ above that of the ground state, consistently with the formula given in Eq. (78), since $\epsilon_{-k} + \epsilon_k = 2\epsilon_k$.

In the p = 1 (odd, PBC) sector, some care must be exercised. One should apply an even number of $\hat{\gamma}^\dagger_k$ to the ground state $|\emptyset_\gamma\rangle^{\text{PBC}}$, including in the choice the unpaired operators $\hat{\gamma}^\dagger_0$, amounting to removing the fermion from the $k = 0$ state, and $\hat{\gamma}^\dagger_\pi$, amounting to creating a fermion in the $k = \pi$ state.

### 4.1.1 The spectral gap

We now want to understand the spectral gap of our model, i.e., the difference between the first excited state $E_1$ and the ground state $E_0$, $\Delta_E = E_1 - E_0$. More generally, we would ask also for the spectral gap of excitations having a given momentum $k$, $\Delta_k$. Naively, one might think that, starting from the ABC ground state of energy $E^{\text{ABC}}_0$, the lowest excited state is obtained by considering states with two extra fermions, $\hat{\gamma}^\dagger_k\hat{\gamma}^\dagger_{-k}|\emptyset\rangle^{\text{ABC}}$, which have energy $E^{\text{ABC}}_0 + 2\epsilon_k$. However, we should consider excitations that, starting for instance from $|\emptyset\rangle^{\text{ABC}}$ and applying a *single* creation operator $\hat{\gamma}^\dagger_k$ lead to a state in the subspace with *opposite* fermion parity, $\hat{\gamma}^\dagger_k|\emptyset\rangle^{\text{ABC}}$, with an energy gap:

$$\Delta_k = \epsilon_k. \tag{79}$$

The difficulty with this way of reasoning is related to the choice of $k$ appropriate in that construction since the $k$ values corresponding to ABC boundary conditions do not coincide with those for PBC. However, it is clear that this will make no difference in the thermodynamics limit $L \to \infty$, so that Eq. (79) does indeed express the spectral gap for excitations at momentum $k$. The smallest such gap is obtained for $k = 0$ when $h > 0$ so that, from Eq. (55), we deduce that:

$$\Delta_E = 2|h - J| = 2|h - h_c|, \tag{80}$$

where $h_c = J$.[14] Two things are worth noticing: 1) $\Delta_E$ vanishes *linearly* with the deviation from the critical point $|h-h_c|$; 2) exactly at criticality, see Fig. 2(c), the spectral gap $\Delta_k$ vanishes linearly in the momentum $|k| \to 0$:

$$\Delta_k^{\text{crit}} = 2J\sqrt{(1-\cos k)^2 + \kappa^2 \sin^2 k} \approx 2J|\kappa k|. \tag{81}$$

### 4.1.2 The Green's functions

In calculating expectation values of operators, for instance, spin-spin correlation functions, it is useful to identify the elementary one-body expectation values, often referred to as *one-particle Green's functions*. Since the number of fermions is not conserved, there are ordinary and anomalous Green's functions [64, 67], which we define here as follows:[15]

$$\mathbf{G}_{jj'} \equiv \langle \psi_0 | \hat{c}_j \hat{c}_{j'}^\dagger | \psi_0 \rangle, \quad \text{and} \quad \mathbf{F}_{jj'} \equiv \langle \psi_0 | \hat{c}_j \hat{c}_{j'} | \psi_0 \rangle. \tag{82}$$

We assume that the initial state $|\psi_0\rangle$ is the Bogoliubov vacuum of the operators $\gamma$ which diagonalise the Hamiltonian, $|\psi_0\rangle = |\emptyset_\gamma\rangle^{\text{ABC}}$. By expressing the real-space fermionic operators in terms of their momentum space counterparts we immediately deduce, using momentum conservation, that:

$$\begin{cases} \mathbf{G}_{jj'} = \dfrac{1}{L}\sum_k e^{ik(j-j')}\langle\psi_0|\hat{c}_k\hat{c}_k^\dagger|\psi_0\rangle = \dfrac{1}{L}\sum_k e^{ik(j-j')}|u_k|^2\,, \\[12pt] \mathbf{F}_{jj'} = \dfrac{e^{2i\phi}}{L}\sum_k e^{ik(j-j')}\langle\psi_0|\hat{c}_k\hat{c}_{-k}|\psi_0\rangle = -\dfrac{e^{2i\phi}}{L}\sum_k e^{ik(j-j')}u_k^* v_k\,, \end{cases} \tag{83}$$

where the last step comes from using the relationship $\hat{c}_k = u_k^* \hat{\gamma}_k + v_k \hat{\gamma}_{-k}^\dagger$ and the fact that $\hat{\gamma}_k |\psi_0\rangle = 0$.

---

**Problem 1. The elementary Green's functions.**

Show that, in the thermodynamic limit $L \to \infty$, by taking $\phi = 0$ and using the properties of $u_k$ and $v_k$, one can write:

$$\begin{cases} \mathbf{G}_{jj'} = \displaystyle\int_0^\pi \frac{dk}{2\pi} \frac{z_k}{\epsilon_k} \cos(k(j-j')) + \frac{1}{2}\delta_{j,j'}\,, \\[12pt] \mathbf{F}_{jj'} = \displaystyle\int_0^\pi \frac{dk}{2\pi} \frac{y_k}{\epsilon_k} \sin(k(j-j'))\,, \end{cases} \tag{84}$$

where $z_k = 2(h - J\cos k)$ and $y_k = 2\kappa J \sin k$. Calculate numerically the Green's functions in the Ising case $\kappa = 1$, for three representative values of the transverse field: A) $h = J/2$, b) $h = J$, c) $h = 2J$. Observe that the Green's functions decay exponentially fast in the separation $|j - j'|$ in cases a) and c). In the Ising case $\kappa = 1$, and at the critical point $h = J$, show analytically that the Green's functions decay as a power law of the distance $j - j'$, obtaining

$$\mathbf{G}_{jj'} = \frac{1}{2}\delta_{j,j'} - \frac{1}{\pi}\frac{1}{4(j-j')^2 - 1}\,, \quad \text{and} \quad \mathbf{F}_{jj'} = \frac{2}{\pi}\frac{(j-j')}{4(j-j')^2 - 1}\,.$$

---

[14]For $h < 0$ the smallest gap is at $k = \pi$, and $\Delta_E = 2|h + J| = 2|h - h_c|$, with $h_c = -J$.

[15]There are many definitions of Green's functions. Here we consider equal-time operators: Apart from a factor $-i$ we have, in Kadanoff-Baym notation [67], would one would denote as $G^>$.

## 4.2 Relationship with the spin representation

It is instructive to comment on the relationship between the spectrum we have found in the fermionic representation and the corresponding physics in the original spin representation. In this section, we fix the anisotropy parameter to $\kappa = 1$, focusing on the Ising case.

Let us start with the classical Ising model ($h = 0$)

$$\widehat{H}_{\text{classical}} = -J \sum_{j=1}^{L} \hat{\sigma}_j^x \hat{\sigma}_{j+1}^x, \tag{85}$$

and consider the two degenerate ground states that you can easily construct in this case:

$$|+, +, \cdots, +\rangle, \qquad \text{and} \qquad |-, -, \cdots, -\rangle, \tag{86}$$

where $|\pm\rangle = \frac{1}{\sqrt{2}}(1, \pm 1)^{\mathsf{T}}$ denote the two eigenstates of $\hat{\sigma}^x$ with eigenvalues $\pm 1$. Recall that the parity operator reads, in terms of spins, as $\widehat{\mathcal{P}} = \prod_{j=1}^{L} \hat{\sigma}_j^z$, and that $\hat{\sigma}^z|\pm\rangle = |\mp\rangle$. Hence, you easily deduce that:

$$\widehat{\mathcal{P}}|+, +, \cdots, +\rangle = |-, -, \cdots, -\rangle, \qquad \text{and} \qquad \widehat{\mathcal{P}}|-, -, \cdots, -\rangle = |+, +, \cdots, +\rangle. \tag{87}$$

This implies that the two eigenstates of the parity operator must be:

$$|\psi_\pm\rangle = \frac{1}{\sqrt{2}}\Big(|+, +, \cdots, +\rangle \pm |-, -, \cdots, -\rangle\Big) \quad \Longrightarrow \quad \widehat{\mathcal{P}}|\psi_\pm\rangle = \mp|\psi_\pm\rangle. \tag{88}$$

These two opposite parity states must be represented by the two fermionic ground states belonging to the ABC and PBC sectors. They are exactly degenerate for $h = 0$. The states in this doublet are crucial for the symmetry-breaking in the thermodynamic limit.

Now consider the effect of a small $h$, taking for simplicity of argument the Ising Hamiltonian with OBC:

$$\widehat{H}_{\text{OBC}} = -J \sum_{j=1}^{L-1} \hat{\sigma}_j^x \hat{\sigma}_{j+1}^x - h \sum_{j=1}^{L} \hat{\sigma}_j^z. \tag{89}$$

Let us consider the limit $|h| \ll J$. The two lowest-energy states have exactly the form in Eq. (86), or Eq. (88), at lowest-perturbative order in $|h|/J$. To construct higher-energy excitations, consider domain-wall configurations of the form

$$|l\rangle = |\underbrace{-, -, \cdots, -}_{\text{sites } 1 \to l}, +, \cdots, +\rangle, \qquad \text{with} \quad l = 1 \cdots L - 1. \tag{90}$$

For $h = 0$, these $L - 1$ lowest-energy domain-wall excitations are degenerate and separated from the two ground states by a gap of $2J$. Therefore, we can study the effect of a small transverse-field term, for $|h| \ll J$, using standard textbook degenerate perturbation theory [63]. The Hamiltonian restricted to the $L - 1$-dimensional subspace of the domain-wall excitations has the form

$$\hat{H}_{\text{eff}} = 2J \sum_{l=1}^{L-1} |l\rangle \langle l| - h \sum_{l=1}^{L-2} \Big(|l\rangle \langle l + 1| + \text{H.c.}\Big). \tag{91}$$

This Hamiltonian is quite easy to diagonalize, as it resembles a standard tight-binding problem with open boundary conditions. As in the quantum mechanical example of an infinite square well [63], it is simple to verify that the appropriate sine combination of two opposite momenta plane waves of momentum $k$ satisfy the correct boundary conditions:

$$|\psi_k\rangle = \frac{1}{\mathcal{N}_k} \sum_{l=1}^{L-1} \sin(kl) |l\rangle, \qquad \text{with} \qquad k = \frac{n\pi}{L}, \tag{92}$$

for $n = 1, \ldots, L-1$, and $\mathcal{N}_k$ a normalization factor. These *delocalized domain-walls* have an energy:

$$\mu_k = 2J - 2h \cos k. \tag{93}$$

We notice that if we expand the excitation energies $\epsilon_k$ in Eq. (55) up to lowest order in $h/J$ we obtain $\epsilon_k \approx \mu_k + \mathcal{O}((h/J)^2)$. So, we see that a state with a single quasiparticle $\hat{\gamma}_k^\dagger$ has the same energy as the delocalized domain-wall state in Eq. (92). This justifies the picture that quasiparticles are indeed delocalized domain walls.[16]

Let us continue our perturbative reasoning to see how we can estimate the separation between the two ground states originating from the $h = 0$ doublet discussed above, when $h > 0$. As mentioned above, the states $|+, +, \cdots, +\rangle$ and $|-, -, \cdots, -\rangle$ are degenerate for $h = 0$, and this doublet is separated from the other states by a gap $\geq 2J$. The degenerate states $|+\rangle = |+, +, \cdots, +\rangle$ and $|-\rangle = |-, -, \cdots, -\rangle$ are coupled only at order $L$ in perturbation theory: we need to flip $L$ spins, with the $\hat{\sigma}_j^z$ operators, to couple one to the other. Hence, we expect their splitting to be $\Delta E_0 \sim (h/J)^L$, i.e., exponentially small in the system size $L$ for small $|h|$: The resulting eigenstates $|\psi_\pm(h)\rangle$, even and odd under parity, approach the two eigenstates in Eq. (88) for $h \to 0$. This energy splitting is exactly the quantity $\Delta E_0$ discussed in Sec. 4.1. So, in the thermodynamic limit, we break the $\mathbb{Z}_2$ symmetry. At any finite size we have the symmetry preserving ground states $|\psi_\pm(h)\rangle$ which tend to Eq. (88) for $h \to 0$. These states can be regarded as superpositions of two macroscopically ordered states $|\pm\rangle_h = \frac{1}{\sqrt{2}}(|\psi_+(h)\rangle \pm |\psi_-(h)\rangle)$, where "macroscopically ordered" means that the longitudinal magnetization $\hat{M}^x = \sum_j \hat{\sigma}_j^x$ has an expectation value which is extensive in $L$. So, in the subspace generated by $|\psi_\pm(h)\rangle$ there can be an explicit symmetry-breaking[17] of $\mathbb{Z}_2$ only in the thermodynamic limit, where the two states are degenerate and the slightest local perturbation selects one of the two macroscopically ordered superpositions $|\pm\rangle_h$.

# 5 Connection with the Onsager solution of the 2d classical Ising model

Although the quantum Ising chain in a transverse field is the primary interest of these lecture notes, it is important to understand its connection with the classical Ising model in two dimensions, whose celebrated exact solution was given by Onsager in 1944 [68]. This connection is representative of a general relationship between classical statistical mechanics in $d$ dimensions, and quantum mechanics in $d-1$ dimensions [69]. We will see that the ground state properties of the one-dimensional quantum Ising model in a transverse field precisely mirror the finite temperature statistical mechanics properties of the classical Ising model in two-dimensions, in the high-anisotropy limit [69]. The critical exponents of the two models are identical, and even the numerical value of the transition temperature predicted by Onsager is perfectly described by the quantum critical point of the quantum Ising chain.

Consider the classical two-dimensional Ising model sketched in Fig. 5 for a square lattice with $L$ sites in the x-direction and $N$ in the y-direction, with periodic boundary conditions

---

[16]We performed our analysis for the case of OBC. The case with PBC is similar, the only difference being that the lowest-energy excitations for $h = 0$ have two domain walls. Nevertheless, with an analysis very similar to the one above, one can show that the excited states for $|h| \ll J$ have energy $2\mu_k$ and can be interpreted as states with two quasiparticles.

[17]Nevertheless, all the states in this doublet, $|\phi\rangle = \alpha|\psi_+(h)\rangle + \beta|\psi_-(h)\rangle$ with $|\alpha|^2 + |\beta|^2 = 1$, show long-range correlations also at finite size. Indeed, the correlator $\langle\phi|\hat{\sigma}_j^x \hat{\sigma}_{j+l}^x|\phi\rangle$ is always finite (equal to 1 in the limit $h \to 0$), and

$$\lim_{l \to \infty} \lim_{L \to \infty} |\langle\phi|\hat{\sigma}_j^x \hat{\sigma}_{j+l}^x|\phi\rangle| \neq 0,$$

expressing the long-range order associated with symmetry-breaking. See the related problem in Sec. 9.

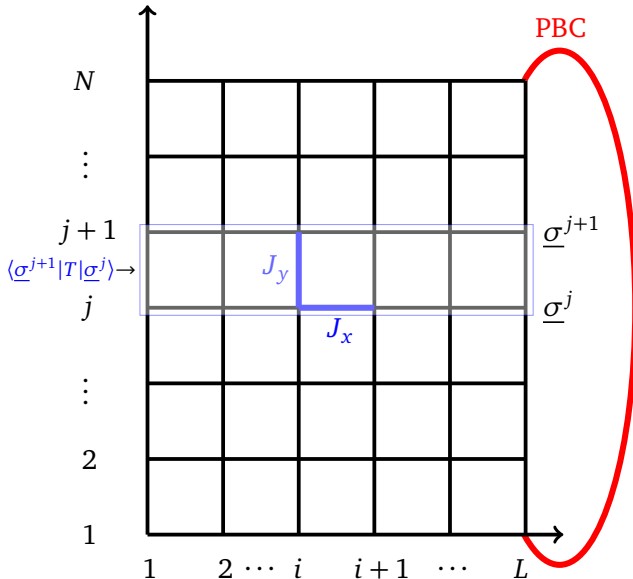

Figure 5: The 2-dimensional classical Ising model on a square lattice with $L \times N$ sites, denoted as $(i, j)$ with $1, \cdots, L$ and $j = 1, \cdots, N$. PBC are enforced in the $y$-direction: You can think of the system as "living on a cylinder" with axis along the $x$-direction. Boundary conditions along the $x$-direction are left unspecified: They could be *open*, *fixed*, or *periodic*. In the latter case, you might picture the system as "living on a torus". The blue lines highlight the couplings $J_x$, connecting sites $(i, j)$ and $(i+1, j)$, and $J_y$, connecting $(i, j)$ and $(i, j+1)$. The shaded rectangle highlights the Boltzmann weights included in the definition of the "transfer matrix" $\langle \underline{\sigma}^{j+1} | \mathrm{T} | \underline{\sigma}^j \rangle$.

(PBC) in the latter. Each lattice since $(i, j)$ is associated with an Ising spin $\sigma_i^j = \pm 1$. The partition function of the model is given by: [70]

$$Z = \sum_{\underline{\sigma}^1, \cdots, \underline{\sigma}^N} \langle \underline{\sigma}^1 | \mathrm{T} | \underline{\sigma}^N \rangle \langle \underline{\sigma}^N | \mathrm{T} | \underline{\sigma}^{N-1} \rangle \cdots \langle \underline{\sigma}^{j+1} | \mathrm{T} | \underline{\sigma}^j \rangle \cdots \langle \underline{\sigma}^2 | \mathrm{T} | \underline{\sigma}^1 \rangle \equiv \mathrm{Tr}\, \mathrm{T}^N, \qquad (94)$$

where the trace emerges from the PBC choice of boundary conditions along the $y$-direction. Here, $\underline{\sigma}^j$ denotes the $j$-th row configuration, comprising the $L$ spins in the x-direction, which we denote as $\underline{\sigma}^j = (\sigma_1^j, \sigma_2^j, \cdots, \sigma_L^j)$, and $\langle \underline{\sigma}^{j+1} | \mathrm{T} | \underline{\sigma}^j \rangle$ is a matrix element of the so-called *transfer matrix* T, a $2^L \times 2^L$ matrix collecting all the Boltzmann weights pertaining to rows $j$ and $j+1$:

$$\langle \underline{\sigma}^{j+1} | \mathrm{T} | \underline{\sigma}^j \rangle = e^{\beta \sum_{i=1}^{L} \left( J_y \sigma_i^j \sigma_i^{j+1} + \frac{J_x}{2} \left( \sigma_i^j \sigma_{i+1}^j + \sigma_i^{j+1} \sigma_{i+1}^{j+1} \right) + \frac{h}{2} \left( \sigma_i^j + \sigma_i^{j+1} \right) \right)}. \qquad (95)$$

We can regard the row configuration $\underline{\sigma}^j$ as a computation basis state for a spin chain with $L$ sites, i.e., simply the eigenstates of Pauli spin operators $\hat{\sigma}_i^z$:

$$\hat{\sigma}_i^z | \underline{\sigma}^j \rangle = \sigma_i^j | \underline{\sigma}^j \rangle.$$

It is simple to write a quantum operator which faithfully reproduces the matrix elements of

the transfer matrix:[18]

$$\widehat{T} = e^{\frac{\beta}{2}\sum_{i=1}^{L}\left(J_x\hat{\sigma}_i^z\hat{\sigma}_{i+1}^z + h\hat{\sigma}_i^z\right)}\left[\prod_{i=1}^{L}\left(e^{\beta J_y}\mathbb{1}_i + e^{-\beta J_y}\hat{\sigma}_i^x\right)\right]e^{\frac{\beta}{2}\sum_{i=1}^{L}\left(J_x\hat{\sigma}_i^z\hat{\sigma}_{i+1}^z + h\hat{\sigma}_i^z\right)}$$

$$= C^L\, e^{\frac{\beta}{2}\sum_{i=1}^{L}\left(J_x\hat{\sigma}_i^z\hat{\sigma}_{i+1}^z + h\hat{\sigma}_i^z\right)}\, e^{\Gamma\sum_{i=1}^{L}\hat{\sigma}_i^x}\, e^{\frac{\beta}{2}\sum_{i=1}^{L}\left(J_x\hat{\sigma}_i^z\hat{\sigma}_{i+1}^z + h\hat{\sigma}_i^z\right)}. \tag{97}$$

Although exact, the quantum expression for $\widehat{T}$ is not easy to handle: The three different terms do not commute and you cannot rewrite $\widehat{T}$ as the exponential of a single quantum Hamiltonian operator. Suppose, however, that the coupling constants $J_x$ and $J_y$ are *highly anisotropic*.

---

ⓘ

**The high-anisotropy limit.** More precisely, assume that:

$$\text{High-anistropy limit:}\quad \begin{cases} \beta J_x = \epsilon J\,, \\[4pt] \beta h = \epsilon h^{\|}\,, \\[4pt] \Gamma = \epsilon h^{\perp} \implies \beta J_y = -\dfrac{1}{2}\log\tanh\epsilon h^{\perp}\,, \end{cases} \tag{98}$$

where $\epsilon$ (with dimensions of "time/$\hbar$") is "**small**", and $J$, $h^{\|}$, $h^{\perp}$ (with dimensions of "energy") are suitable constants. More properly, the dimensionless combinations $\epsilon J$, $\epsilon h^{\|}$, and $\epsilon h^{\perp}$ are assumed to be small. These assumptions imply that when $\beta J_x$ is "**small**", then $\beta J_y = -\frac{1}{2}\log\tanh\epsilon h^{\perp}$ is "**large**", justifying the terminology "**high-anisotropy limit**", when $\epsilon \to 0$.

---

In terms of these quantities, let us now define two quantum operators:

$$\widehat{H}_z = -\sum_{i=1}^{L}\left(J\hat{\sigma}_i^z\hat{\sigma}_{i+1}^z + h^{\|}\hat{\sigma}_i^z\right)\,, \qquad \text{and} \qquad \widehat{H}_x = -h^{\perp}\sum_{i=1}^{L}\hat{\sigma}_i^x\,, \tag{99}$$

such that the exact transfer matrix can be written as

$$\widehat{T} = C^L\, e^{-\frac{\epsilon}{2}\widehat{H}_z}e^{-\epsilon\widehat{H}_x}e^{-\frac{\epsilon}{2}\widehat{H}_z}\,. \tag{100}$$

In high-anisotropy limit $\epsilon \to 0$, the three exponentials can be combined by using the relationship

$$e^{\epsilon\hat{A}}e^{\epsilon\hat{B}} = e^{\epsilon\hat{A}+\epsilon\hat{B}+\frac{\epsilon^2}{2}[\hat{A},\hat{B}]+\cdots} = e^{\epsilon(\hat{A}+\hat{B})} + O(\epsilon^2)\,, \tag{101}$$

which is the simplest instance of the Baker-Campbell-Hausdorff formula.

---

[18]The first expression is easy to establish from the requirement in Eq. (95). The second expression follows by rewriting:

$$\left(e^{\beta J_y}\mathbb{1}_i + e^{-\beta J_y}\hat{\sigma}_i^x\right) \equiv C\, e^{\Gamma\hat{\sigma}_i^x} \equiv C\left(\mathbb{1}_i\cosh\Gamma + \hat{\sigma}_i^x\sinh\Gamma\right)\,,$$

where we used $(\hat{\sigma}^x)^{2n} = \mathbb{1}$, and $(\hat{\sigma}^x)^{2n+1} = \hat{\sigma}^x$ in the second equality, to expand the exponential. To find the constants $C$ and $\Gamma$ we write explicitly:

$$\begin{cases} C\cosh\Gamma = e^{\beta J_y}\,, \\[4pt] C\sinh\Gamma = e^{-\beta J_y} \end{cases} \implies \begin{cases} \tanh\Gamma = e^{-2\beta J_y}\,, \\[4pt] C^2 = \dfrac{2}{\sinh 2\Gamma}\,. \end{cases} \tag{96}$$

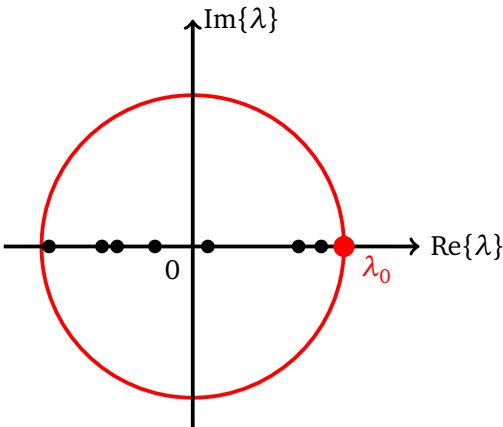

Figure 6: The Perron-Frobenius theorem. The red dot denotes $\lambda_0$, the Perron root, which is the maximum eigenvalue of T, real and positive. All other eigenvalues (smaller black dots) stay within the circle of radius $\lambda_0$ in the complex plane. For the symmetric T considered here, all other eigenvalues are real as well.

**The transfer matrix in the high-anisotropy limit.** This leads us to our final expression:

$$\widehat{T} \overset{\epsilon \to 0}{\simeq} C^L \, e^{-\epsilon \widehat{H}}, \quad \text{with} \quad \widehat{H} = \widehat{H}_z + \widehat{H}_x = -\sum_{i=1}^{L}\left(J\hat{\sigma}_i^z\hat{\sigma}_{i+1}^z + h^{\parallel}\hat{\sigma}_i^z + h^{\perp}\hat{\sigma}_i^x\right). \quad (102)$$

So, in the high-anisotropy limit, the classical transfer matrix has been mapped onto the *imaginary-time*[19] evolution operator of a *quantum Ising chain in a transverse field* $h^{\perp}$. The $y$-direction of the classical problem — for which we chose periodic boundary conditions — becomes the imaginary-time direction of the quantum problem.

To better understand the deep relationship between the statistical mechanics of the classical Ising model in two dimensions, and the physics of the quantum Ising chain, let us return to the classical transfer matrix T. Since T is a positive real matrix — and even symmetric, by construction — we can use Perron-Frobenius theorem, which guarantees that T has a unique (positive) eigenstate $|\lambda_0\rangle$ with a *maximum eigenvalue* $\lambda_0$ — the so-called *Perron root* — which is itself *real and positive* and greater than the modulus of any other eigenvalue, $\lambda_0 > |\lambda_{\alpha>0}|$, in general complex, but in the present case real, because T is symmetric. We can write, in terms of the eigenstates $|\lambda_\alpha\rangle$ of T, the spectral decomposition of T as:

$$T = \sum_{\alpha=0}^{2^L-1} \lambda_\alpha |\lambda_\alpha\rangle\langle\lambda_\alpha| \qquad \Longrightarrow \qquad Z = \text{Tr}\,T^N = \sum_{\alpha=0}^{2^L-1} \lambda_\alpha^N,$$

where the last equation follows because $T^N$ is easy to calculate on the basis of the eigenvectors of T, and its trace, leading to $Z$, is also simply the sum of all $\lambda_\alpha^N$.

---

[19]The usual real-time evolution operator $e^{-it\widehat{H}/\hbar}$ becomes, under the substitution $t \to -i\tau$, the imaginary-time evolution operator $e^{-\tau\widehat{H}/\hbar}$, very important in numerical Quantum Monte Carlo approaches, see Ref. [12], and in statistical mechanics, see Ref. [70].

**Exponential dominance of the Perron root.** Let us denote the maximum eigenvalue of T by $\lambda_0$. In the limit $N \to \infty$, the partition sum is *exponentially dominated* by $\lambda_0$:

$$\frac{\beta F}{N} = -\frac{1}{N} \log Z \xrightarrow{N \to \infty} -\log \lambda_0.$$

In the high-anisotropy limit $\epsilon \to 0$, we rewrite the partition function as:

$$Z = \operatorname{Tr} T^N \equiv \operatorname{Tr} \widehat{T}^N \overset{\epsilon \to 0}{\simeq} C^{LN} \operatorname{Tr} e^{-\epsilon N \widehat{H}}. \tag{103}$$

The Perron-Frobenius theorem, in this context, tells us that the largest eigenvalue of T, connected to the ground state energy of $\widehat{H}$, $\lambda_0 \overset{\epsilon \to 0}{\simeq} C e^{-\epsilon E_0}$, dominates the partition sum. Indeed, for any finite $L$, the quantum Hamiltonian $\widehat{H}$ has a finite gap $\Delta_E = E_1 - E_0$ above its (non-degenerate) ground state energy $E_0 = L e_0$, so that the next eigenvalue makes a negligible contribution in the limit $N \to \infty$ (as long as $\epsilon > 0$):

$$\lambda_1^N \overset{\epsilon \to 0}{\simeq} C^N e^{-\epsilon N E_1} = C^N e^{-\epsilon N E_0} e^{-\epsilon N \Delta_E} \approx \lambda_0^N e^{-\epsilon N \Delta_E}.$$

All in all, for large $N$ (and fixed small $\epsilon$) we would write:

$$Z = e^{-\beta F} \overset{\epsilon \to 0}{\simeq} C^{LN} e^{-\epsilon N E_0} \left(1 + e^{-\epsilon N \Delta_E} + \cdots\right) \approx C^{LN} e^{-\epsilon L N e_0} + \cdots.$$

Taking the logarithm and dividing by $N_{2d} = NL$ we get the free-energy per spin $f$:

$$\beta f = \lim_{N_{2d} \to \infty} \frac{\beta F}{N_{2d}} = -\lim_{N_{2d} \to \infty} \frac{\log Z}{N_{2d}} \overset{\epsilon \to 0}{\simeq} \epsilon e_0 - \log C, \tag{104}$$

where the constant $C$ plays a minor role: It is simply an additive contribution to the free energy.

**Singularities of $f$ correspond to singularities of $e_0$.** The important point is that we expect that the *singularities* of the classical free-energy per spin $f$ (in the thermodynamical limit) should be reflected by the singularities of $e_0$, the *ground state energy* per spin of the quantum model.

Let us check this prediction. In zero longitudinal field ($h = 0$) and for uniform couplings with PBC in both directions, the classical Ising model in $d = 2$ has been solved by Onsager, who succeeded in diagonalizing the exact transfer matrix T [68]. Onsager's solution predicts that the 2d Ising model in zero longitudinal field has a transition temperature $T_c$ (where the free-energy $f$ shows singularities) given by:

$$\sinh(2\beta_c J_x) \sinh(2\beta_c J_y) = 1. \tag{105}$$

Let us briefly summarize the physics of this classical model. For $T > T_c$ the system is in a disordered phase, where the thermal average of the local spin at any site **R** vanishes in zero longitudinal field ($h = 0$): $\langle \sigma_R \rangle_0 = 0$. For $T < T_c$, on the contrary, the $\mathbb{Z}_2$ symmetry of the classical model in zero field is *spontaneously broken* in the thermodynamic limit and a non-vanishing *local order parameter m* develops:

$$m = \langle \sigma_R \rangle_0 = \lim_{h \to 0^+} \lim_{N_{2d} \to \infty} \langle \sigma_R \rangle > 0. \tag{106}$$

The way in which $m$ vanishes when $T \to T_c^-$, the critical temperature, is captured by a *critical exponent* traditionally denoted as $\beta$ (not to be confused with $1/k_B T$):

$$m(T) \propto (T_c - T)^\beta, \quad T \le T_c. \tag{107}$$

For the Ising model in $d = 2$, its exact value is $\beta = \frac{1}{8}$. When the symmetry is spontaneously broken, the large-distance limit of the correlation function of the order parameter tends exponentially fast to the *square* of the local order parameter:

$$\langle \sigma_{\mathbf{R}} \sigma_{\mathbf{R}'} \rangle_0 \xrightarrow{|\mathbf{R}-\mathbf{R}'| \to \infty} m^2 + O\left(e^{-|\mathbf{R}-\mathbf{R}'|/\xi(T)}\right), \tag{108}$$

where $\xi(T)$ is a temperature-dependent *correlation length*. This exponentially fast decrease of correlations is indeed true both in the broken symmetry phase, where $m \ne 0$, as well as in the high-temperature symmetric phase, where $m = 0$, suggesting that it is convenient to define the *connected correlation functions* as

$$C_{\mathbf{x}}^{\text{conn}} = \langle \sigma_{\mathbf{x}} \sigma_{\mathbf{0}} \rangle_0 - m^2, \tag{109}$$

which decays exponentially fast to zero both for $T < T_c$, and for $T > T_c$. A power-law behaviour emerges at $T_c$, where $\xi(T_c) = \infty$ (in the thermodynamic limit). A scaling *Ansatz* proposed by M. Fisher [71, 72] tells us that:

$$C_{\mathbf{x}}^{\text{conn}} = \langle \sigma_{\mathbf{x}} \sigma_{\mathbf{0}} \rangle_0 - m^2 \propto \frac{e^{-|\mathbf{x}|/\xi(T)}}{|\mathbf{x}/a|^{d-2+\eta}}, \tag{110}$$

where the last expression is valid for $|\mathbf{x}| \gg a$, the lattice spacing. Here $d$ is the dimensionality of the lattice, $\xi(T)$ is the *correlation length*, and $\eta$ is the *anomalous exponent* for correlations. The correlation length $\xi(T)$ is finite for $T \ne T_c$, but *diverges* at $T_c$ with a critical exponent $\nu$

$$\xi(T) \sim \frac{1}{|T - T_c|^\nu}. \tag{111}$$

For the Ising model in $d = 2$, $\nu = 1$ and $\eta = \frac{1}{4}$.

Let us now switch to the quantum Ising chain. The quantum critical point of the uniform quantum Ising chain, where singularities of

$$e_0 = \lim_{L \to \infty} \frac{E_0}{L} = -\int_0^\pi \frac{dk}{2\pi} \epsilon_k, \tag{112}$$

are present, occurs at $h_c^\perp = J$. Indeed, if you look at the bands in Fig. 2 you immediately see that they are very smooth for $h^\perp \ne J$ (panels b and d), but develop a *cusp* for $h^\perp = J$ (panel c). This corresponds, in terms of Eq. (98), to

$$h_c^\perp = J \iff \Gamma_c = \beta_c J_x \iff e^{-2\beta_c J_y} = \tanh \beta_c J_x. \tag{113}$$

The correspondence of parameters is such that the high-temperature (disordered) phase for $T > T_c$, and the low-temperature (ordered) phase for $T < T_c$ are mapped into the quantum phases with $h^\perp > J$, and $h^\perp < J$, respectively.

Remarkably, all exact critical exponents of the zero-field 2d Ising model are reproduced by the quantum Ising chain. Let us consider the correlation length critical exponent $\nu$. Following Kogut [69][Sec. III], consider the spin-spin correlation function between sites with the same x-coordinate, and a distance $na$ away in the y-direction: $\mathbf{R} = a(i, 1)$ and $\mathbf{R}' = a(i, n + 1)$.

Translational invariance implies that the result depends only on $n$. By calculating the thermal average using the transfer matrix, and taking the high-anisotropy limit, we get:

$$C_n = \langle \sigma_i^{n+1} \sigma_i^1 \rangle = \frac{1}{\mathrm{Tr}\,\mathrm{T}^N} \mathrm{Tr}(\mathrm{T}^{N-n} \sigma_i^{n+1} \mathrm{T}^n \sigma_i^1)$$

$$\xrightarrow{\text{small } \epsilon} \frac{1}{\mathrm{Tr}\,\mathrm{e}^{-\epsilon N \widehat{H}}} \mathrm{Tr}(\mathrm{e}^{-\epsilon(N-n)\widehat{H}} \hat{\sigma}_i^z \, \mathrm{e}^{-\epsilon n \widehat{H}} \hat{\sigma}_i^z). \tag{114}$$

In the limit $N \to \infty$, the quantum ground state $|\psi_0\rangle$ dominates the trace, hence:

$$C_n \xrightarrow{\text{small } \epsilon,\, N \to \infty} \frac{1}{\mathrm{e}^{-\epsilon N E_0}} \langle \psi_0 | \mathrm{e}^{-\epsilon(N-n)\widehat{H}} \hat{\sigma}_i^z \mathrm{e}^{-\epsilon n \widehat{H}} \hat{\sigma}_i^z | \psi_0 \rangle$$

$$= \mathrm{e}^{\epsilon n E_0} \langle \psi_0 | \hat{\sigma}_i^z \mathrm{e}^{-\epsilon n \widehat{H}} \hat{\sigma}_i^z | \psi_0 \rangle$$

$$= \sum_{m=0}^{\infty} \langle \psi_0 | \hat{\sigma}_i^z \mathrm{e}^{-\epsilon n (\widehat{H} - E_0)} | \psi_m \rangle \langle \psi_m | \hat{\sigma}_i^z | \psi_0 \rangle$$

$$= \sum_{m=0}^{\infty} \mathrm{e}^{-\epsilon n (E_m - E_0)} \left| \langle \psi_m | \hat{\sigma}_i^z | \psi_0 \rangle \right|^2, \tag{115}$$

where we have inserted a resolution of the identity with eigenstates $|\psi_m\rangle$ of $\widehat{H}$. In the limit of large $n$, it is appropriate to keep only the ground and first excited state $|\psi_1\rangle$ out of the infinite sum, hence:

$$C_n \xrightarrow{\text{small } \epsilon,\, n \text{ large}} \left| \langle \psi_0 | \hat{\sigma}_i^z | \psi_0 \rangle \right|^2 + \mathrm{e}^{-\epsilon n (E_1 - E_0)} \left| \langle \psi_1 | \hat{\sigma}_i^z | \psi_0 \rangle \right|^2 + \cdots. \tag{116}$$

The spectral gap $\Delta_E = E_1 - E_0$ emerges as the crucial quantity determining the large $n$ behaviour of correlations. By comparing this expression with the general form in Eq. (108), we realize that:

$$m = \langle \psi_0 | \hat{\sigma}_i^z | \psi_0 \rangle, \qquad \text{and} \qquad \frac{a}{\xi} = \epsilon \Delta_E. \tag{117}$$

Since $\Delta_E$ vanishes linearly at the critical point, see Eq. (80), we conclude that $\nu = 1$.

Concerning the order parameter $m$, the calculation by Pfeuty [65][Eq. 3.12] shows that

$$m = \langle \psi_0 | \hat{\sigma}_i^z | \psi_0 \rangle = \begin{cases} \left( 1 - \left( h^\perp / J \right)^2 \right)^{\frac{1}{8}}, & \text{for} \qquad |h^\perp / J| < 1, \\ 0, & \text{for} \qquad |h^\perp / J| \geq 1. \end{cases} \tag{118}$$

Hence $\beta = \frac{1}{8}$.

As for the anomalous exponent $\eta$, it should be extracted from the calculation of *spin-spin correlations at the critical point*. We will address this calculation explicitly in Sec. 9. The result will be that $\eta = \frac{1}{4}$, as expected.

Finally, concerning the specific heat singularities, observe that one expects, at a general second-order critical point: [70]

$$c_v(T) = -T \frac{\partial^2 f}{\partial T^2} \propto |T - T_c|^{-\alpha}, \tag{119}$$

where $\alpha = 0$ (a logarithmic singularity) for the 2d Ising model. The quantum equivalent of this singularity shows up in the second derivative of the ground state energy per spin, $e_0$, with respect to the transverse field $h^\perp$.

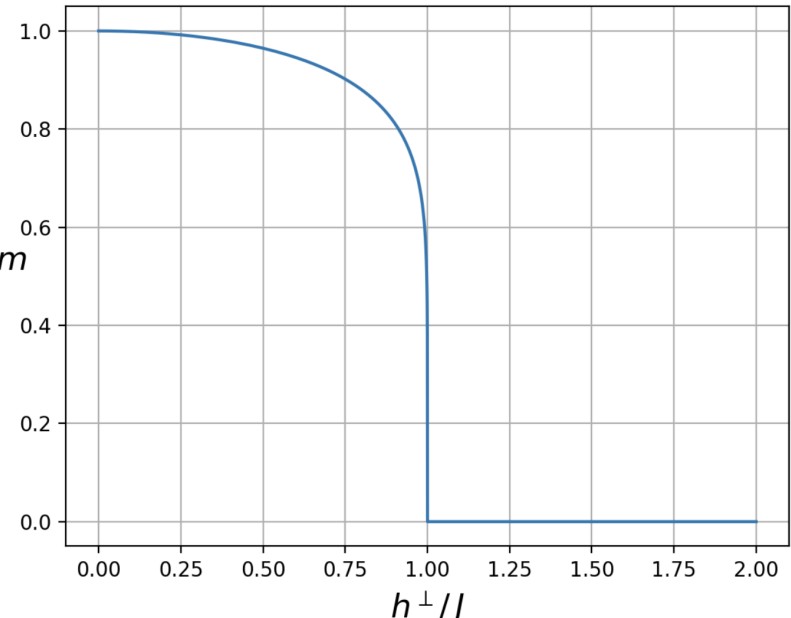

Figure 7: Plot of order parameter $m$ for the Ising chain in a transverse field, according to Eq. (118).

---

**Problem 2. Singularities of the ground state energy.**

Calculate the second derivative of $e_0$ with respect to $h^\perp$, and show that it has a logarithmic divergence, in the thermodynamic limit, for $h^\perp \to J$.

---

Quantitatively, you might wonder how close are the two predictions for $T_c$ — the one deduced from Onsager's solution, Eq. (105), and the one deduced from the critical point of the quantum Ising chain, Eq. (113) —- as a function of the anisotropy of the couplings $J_x/J_y$ (which should be "small" for the quantum mapping to be in principle valid). The two results are shown in Fig. 8: Rather surprisingly, Eqs. (105) and (113) give *precisely*[20] the same $T_c$ *for all values of the anisotropy $J_x/J_y$.*

> ℹ
> **Summary of classical to quantum mapping.** To summarise, the quantum Ising chain in a transverse field captures perfectly well the critical singularities of the classical two-dimensional Ising model. As a bonus (but this is not general), we even get a quantitatively perfect prediction for the critical point temperature $T_c$, well beyond the high-anisotropy limit. Interestingly, the Jordan-Wigner mapping is unable to solve the 1d quantum Ising chain precisely in the case Onsager's solution cannot deal with: In the presence of a longitudinal field.

---

[20]If you call $z = e^{-2\beta_c J_y}$, from the Onsager relation Eq. (105), solving a simple quadratic equation, you get:

$$z = e^{-2\beta_c J_y} = \frac{\sqrt{S^2 + 4} - 2}{S}, \quad \text{with} \quad S = 2\sinh(2\beta_c J_x).$$

Straightforward algebra, using duplication formulas $\cosh(2\beta_c J_x) = \cosh^2(\beta_c J_x) + \sinh^2(\beta_c J_x)$ and $\sinh(2\beta_c J_x) = 2\sinh(\beta_c J_x)\cosh(\beta_c J_x)$, leads then to the result stated in Eq. (113).

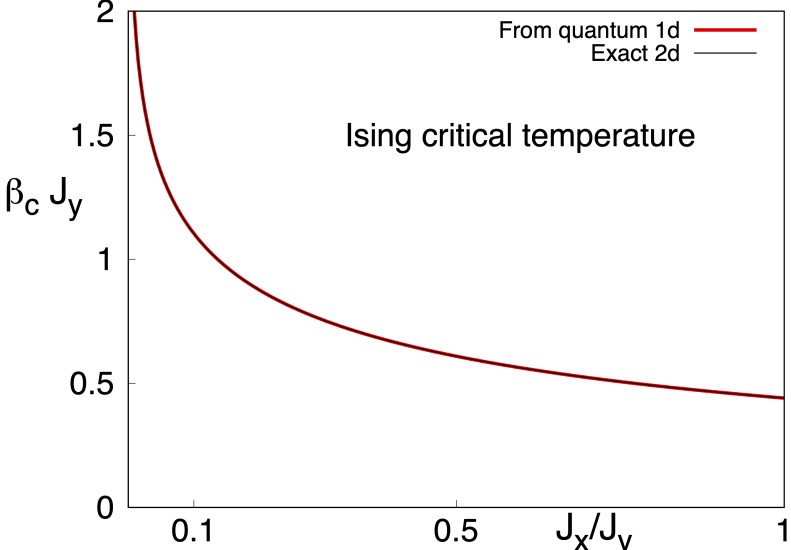

Figure 8: Plot of the critical coupling $\beta_c J_y = J_y/(k_B T_c)$ from the 2d Onsager's solution, Eq. (105), compared to the quantum-mapped prediction of Eq. (113), versus the anisotropy of the lattice $J_x/J_y$. The two predictions *coincide*.

## 6 Nambu formalism for the general disordered case

As we have seen, in the ordered case the Hamiltonian can be diagonalized by a Fourier transformation, reducing the problem to a collection of $2 \times 2$ "pseudo-spin-1/2" problems, followed by a Bogoliubov transformation, as first shown in Refs. [65,73,74]. In the disordered case, we can proceed similarly, but we cannot reduce ourselves to $2 \times 2$ problems in a simple way.[21] By using the Nambu formalism, we define a column vector $\widehat{\boldsymbol{\Psi}}$ and its Hermitian conjugate row vector $\widehat{\boldsymbol{\Psi}}^\dagger$, each of length $2L$, by

$$
\widehat{\boldsymbol{\Psi}} = \begin{pmatrix} \hat{c}_1 \\ \vdots \\ \hat{c}_L \\ \hat{c}_1^\dagger \\ \vdots \\ \hat{c}_L^\dagger \end{pmatrix} = \begin{pmatrix} \hat{\mathbf{c}} \\ \hat{\mathbf{c}}^\dagger \end{pmatrix}, \qquad \widehat{\boldsymbol{\Psi}}^\dagger = \left( \hat{c}_1^\dagger, \cdots, \hat{c}_L^\dagger, \hat{c}_1, \cdots, \hat{c}_L \right) = \left( \hat{\mathbf{c}}^\dagger, \hat{\mathbf{c}} \right), \tag{120}
$$

or $\widehat{\boldsymbol{\Psi}}_j = \hat{c}_j$, $\widehat{\boldsymbol{\Psi}}_{j+L} = \hat{c}_j^\dagger$, and $\widehat{\boldsymbol{\Psi}}_j^\dagger = \hat{c}_j^\dagger$, $\widehat{\boldsymbol{\Psi}}_{j+L}^\dagger = \hat{c}_j$, for $j \leq L$.[22]

---

[21]For the time-independent case, a theorem due to Bloch and Messiah guarantees that there is always an appropriate basis in which the problem reduces to $2 \times 2$ blocks, but this is not very useful if you are willing to tackle dynamical problems. See Sec. A.

[22]The notation for $\hat{\mathbf{c}}$ might be a bit confusing and should be intended as a shorthand, rather than a column vector. We are not consistently assuming, for instance, that $\hat{\mathbf{c}}^\dagger$ is a row vector. The same shorthanded but imperfect notations will be assumed later on for the Bogoliubov rotated operators $\hat{\boldsymbol{\gamma}}$ and $\hat{\boldsymbol{\gamma}}^\dagger$.

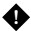

**Warning:** Notice that the $\widehat{\boldsymbol{\Psi}}$ satisfy quite standard fermionic anti-commutation relations

$$\{\widehat{\boldsymbol{\Psi}}_j, \widehat{\boldsymbol{\Psi}}_{j'}^\dagger\} = \delta_{j,j'}, \tag{121}$$

for $j, j' = 1, ..., 2L$, except that $\{\widehat{\boldsymbol{\Psi}}_j, \widehat{\boldsymbol{\Psi}}_{j+L}\} = 1$ for all $j \leq L$, which brings about certain factors 2 in the Heisenberg's equations of motion (see later).

It is useful, for later purposes, to introduce the $2L \times 2L$ *swap matrix* $\mathbb{S}$:

$$\mathbb{S} = \begin{pmatrix} \mathbf{0}_{L \times L} & \mathbf{1}_{L \times L} \\ \mathbf{1}_{L \times L} & \mathbf{0}_{L \times L} \end{pmatrix}, \tag{122}$$

in terms of which $\widehat{\boldsymbol{\Psi}}^\dagger = (\mathbb{S}\widehat{\boldsymbol{\Psi}})^{\mathrm{T}}$.

Consider now a general fermionic quadratic form

$$\widehat{H} = \sum_{jj'} \left( \mathbf{A}_{j'j} \hat{c}_j^\dagger \hat{c}_j + \mathbf{A}_{j'j}^* \hat{c}_j^\dagger \hat{c}_{j'} \right) + \sum_{jj'} \left( \mathbf{B}_{j'j} \hat{c}_j^\dagger \hat{c}_j^\dagger + \mathbf{B}_{j'j}^* \hat{c}_j \hat{c}_{j'} \right), \tag{123}$$

where $\mathbf{A}_{j'j} = \mathbf{A}_{jj'}^*$, i.e., $\mathbf{A} = \mathbf{A}^\dagger$ is Hermitian, and $\mathbf{B}_{jj'} = -\mathbf{B}_{j'j}$, i.e., $\mathbf{B} = -\mathbf{B}^{\mathrm{T}}$ is anti-symmetric because $\hat{c}_j \hat{c}_{j'}$ is anti-symmetric under exchange of the two operators, and any symmetric part of $\mathbf{B}$ would not contribute. It is simple to verify that $\widehat{H}$ can be expressed in terms of $\widehat{\boldsymbol{\Psi}}$, omitting an irrelevant constant term $\mathrm{Tr}\,\mathbf{A}$, as:

$$\widehat{H} = \widehat{\boldsymbol{\Psi}}^\dagger \mathbb{H} \widehat{\boldsymbol{\Psi}} = \left( \hat{\mathbf{c}}^\dagger, \hat{\mathbf{c}} \right) \begin{pmatrix} \mathbf{A} & \mathbf{B} \\ -\mathbf{B}^* & -\mathbf{A}^* \end{pmatrix} \begin{pmatrix} \hat{\mathbf{c}} \\ \hat{\mathbf{c}}^\dagger \end{pmatrix}. \tag{124}$$

There is an intrinsic particle-hole symmetry in a fermionic Hamiltonian having this form. This symmetry, further discussed in Sec. 7.1, is connected with the fact that the Hermitian $2L \times 2L$ matrix $\mathbb{H}$ satisfies:

$$\mathbb{H}\mathbb{S} = -\mathbb{S}\mathbb{H}^*. \tag{125}$$

In the XY-Ising case, all couplings are real and we have two different fermionic Hamiltonians, one for each parity sector $\mathrm{p} = 0, 1$, which we report here for convenience, using also $2\hat{n}_j - 1 = \hat{c}_j^\dagger \hat{c}_j - \hat{c}_j \hat{c}_j^\dagger$:

$$\widehat{\mathbb{H}}_{\mathrm{p}=0,1} = -\sum_{j=1}^{L} \left( J_j^+ \hat{c}_j^\dagger \hat{c}_{j+1} + J_j^- \hat{c}_j^\dagger \hat{c}_{j+1}^\dagger + \mathrm{H.c.} \right) + \sum_{j=1}^{L} h_j (\hat{c}_j^\dagger \hat{c}_j - \hat{c}_j \hat{c}_j^\dagger), \tag{126}$$

with the boundary condition:

$$\hat{c}_{L+1} = (-1)^{\mathrm{p}+1} \hat{c}_1. \tag{127}$$

The corresponding $2L \times 2L$ matrices $\mathbb{H}_{\mathrm{p}}$ are now real and symmetric. Hence $\mathbf{A}$ is real and symmetric ($\mathbf{A} = \mathbf{A}^* = \mathbf{A}^{\mathrm{T}}$), and $\mathbf{B}$ is real and anti-symmetric ($\mathbf{B} = \mathbf{B}^* = -\mathbf{B}^{\mathrm{T}}$):

$$\mathbb{H} = \begin{pmatrix} \mathbf{A} & \mathbf{B} \\ -\mathbf{B}^* & -\mathbf{A}^* \end{pmatrix} \xrightarrow{\text{Ising}} \mathbb{H}_{\mathrm{p}} = \begin{pmatrix} \mathbf{A} & \mathbf{B} \\ -\mathbf{B} & -\mathbf{A} \end{pmatrix}. \tag{128}$$

The structure of the two blocks $\mathbf{A}$ and $\mathbf{B}$ is given, in the Ising case, by:

$$\begin{cases} \mathbf{A}_{j,j} = h_j, \\ \mathbf{A}_{j,j+1} = \mathbf{A}_{j+1,j} = -\dfrac{J_j^+}{2} = -\dfrac{J_j}{2}, \end{cases} \qquad \begin{cases} \mathbf{B}_{j,j} = 0, \\ \mathbf{B}_{j,j+1} = -\mathbf{B}_{j+1,j} = -\dfrac{J_j^-}{2} = -\dfrac{\kappa J_j}{2}, \end{cases} \tag{129}$$

where we have assumed, once again, that $J_j^x = J_j(1 + \kappa)/2$ and $J_j^y = J_j(1 - \kappa)/2$. In the PBC-spin case, we have additional matrix elements:

$$\mathbf{A}_{L,1} = \mathbf{A}_{1,L} = (-1)^p \frac{J_L^+}{2} = (-1)^p \frac{J_L}{2}, \tag{130}$$

and

$$\mathbf{B}_{L,1} = -\mathbf{B}_{1,L} = (-1)^p \frac{J_L^-}{2} = (-1)^p \frac{\kappa J_L}{2}, \tag{131}$$

both depending on the fermion parity p. The OBC case is recovered by simply setting $J_L = 0$, which makes $\mathbb{H}_1 = \mathbb{H}_0$.

# 7   Diagonalisation of $\widehat{H}$: The time-independent case

We start considering the eigenvalue problem for a general Hermitian $2L \times 2L$ matrix showing that the intrinsic particle-hole symmetry of the problem leads to the Bogoliubov-de Gennes (BdG) equations. See Refs. [75–77]. We remark that one recovers the results of Sec. 4 when the couplings are uniform and the matrices $\mathbf{A}$ and $\mathbf{B}$ have a simple translationally-invariant structure.

## 7.1   The Bogoliubov-de Gennes equations

Let us consider the eigenvalue problem for a general Hermitian $2L \times 2L$ matrix $\mathbb{H}$

$$\mathbb{H} \begin{pmatrix} \mathbf{u}_\mu \\ \mathbf{v}_\mu \end{pmatrix} = \begin{pmatrix} \mathbf{A} & \mathbf{B} \\ -\mathbf{B}^* & -\mathbf{A}^* \end{pmatrix} \begin{pmatrix} \mathbf{u}_\mu \\ \mathbf{v}_\mu \end{pmatrix} = \epsilon_\mu \begin{pmatrix} \mathbf{u}_\mu \\ \mathbf{v}_\mu \end{pmatrix}, \tag{132}$$

where $\mathbf{u}, \mathbf{v}$ are $L$-dimensional column vectors, composing the $2L$-dimensional column vector $\begin{pmatrix} \mathbf{u}_\mu \\ \mathbf{v}_\mu \end{pmatrix}$, and the $\mu$ index refers to $\mu$-th eigenvector. By explicitly writing the previous system, we find the so-called Bogoliubov-de Gennes equations:

$$\begin{cases} \mathbf{A} \, \mathbf{u}_\mu + \mathbf{B} \, \mathbf{v}_\mu = \epsilon_\mu \mathbf{u}_\mu, \\ -\mathbf{B}^* \mathbf{u}_\mu - \mathbf{A}^* \mathbf{v}_\mu = \epsilon_\mu \mathbf{v}_\mu. \end{cases} \tag{133}$$

It is easy to verify that if $(\mathbf{u}_\mu, \mathbf{v}_\mu)^{\mathrm{T}}$ is eigenvector with eigenvalue $\epsilon_\mu$, then $(\mathbf{v}_\mu^*, \mathbf{u}_\mu^*)^{\mathrm{T}}$ is an eigenvector with eigenvalue $-\epsilon_\mu$.[23] In the Ising case, $\mathbf{A} = \mathbf{A}^*$ and $\mathbf{B} = \mathbf{B}^*$, and we can always take the solutions to be real.

We can organize the eigenvectors in a unitary (orthogonal, if the solutions are real) $2L \times 2L$ matrix

$$\mathbb{U} = \left( \begin{array}{ccc|ccc} \mathbf{u}_1 & \cdots & \mathbf{u}_L & \mathbf{v}_1^* & \cdots & \mathbf{v}_L^* \\ \mathbf{v}_1 & \cdots & \mathbf{v}_L & \mathbf{u}_1^* & \cdots & \mathbf{u}_L^* \end{array} \right) = \begin{pmatrix} \mathbf{U} & \mathbf{V}^* \\ \mathbf{V} & \mathbf{U}^* \end{pmatrix}, \tag{134}$$

---

[23]Indeed:

$$\begin{cases} \mathbf{A} \, \mathbf{v}_\mu^* + \mathbf{B} \, \mathbf{u}_\mu^* = -\epsilon_\mu \mathbf{v}_\mu^*, \\ -\mathbf{B}^* \mathbf{v}_\mu^* - \mathbf{A}^* \mathbf{u}_\mu^* = -\epsilon_\mu \mathbf{u}_\mu^*, \end{cases}$$

coincides exactly with Eq. (133), after taking a complex conjugation, exchanging the two equations and reshuffling the terms. An alternative derivation uses the fact that

$$\mathbb{S} \begin{pmatrix} \mathbf{u} \\ \mathbf{v} \end{pmatrix} = \begin{pmatrix} \mathbf{v} \\ \mathbf{u} \end{pmatrix},$$

and that $\mathbb{H}\mathbb{S} = -\mathbb{S}\mathbb{H}^*$, see Eq. (125).

$\mathbf{U}$ and $\mathbf{V}$ being $L \times L$ matrices (real, as we can choose to be, in the Ising case) with the $\mathbf{u}_j$ and $\mathbf{v}_j$ as columns. As a consequence:

$$\mathbb{U}^\dagger \mathbb{H} \mathbb{U} = \left(\begin{array}{cccc|cccc} \epsilon_1 & 0 & \cdots & 0 & 0 & 0 & \cdots & 0 \\ 0 & \epsilon_2 & \cdots & 0 & 0 & 0 & \cdots & 0 \\ \vdots & \vdots & \cdots & \vdots & \vdots & \vdots & \cdots & \vdots \\ 0 & 0 & \cdots & \epsilon_L & 0 & 0 & \cdots & 0 \\ \hline 0 & 0 & \cdots & 0 & -\epsilon_1 & 0 & \cdots & 0 \\ 0 & 0 & \cdots & 0 & 0 & -\epsilon_2 & \cdots & 0 \\ \vdots & \vdots & \cdots & \vdots & \vdots & \vdots & \cdots & \vdots \\ 0 & 0 & \cdots & 0 & 0 & 0 & \cdots & -\epsilon_L \end{array}\right) \equiv \operatorname{diag}(\epsilon_\mu, -\epsilon_\mu) = \mathbb{E}_{\text{diag}}. \quad (135)$$

If we define the new Nambu fermion[24] operators $\widehat{\boldsymbol{\Phi}}$ and $\widehat{\boldsymbol{\Phi}}^\dagger$ in such way that

$$\widehat{\boldsymbol{\Psi}} = \mathbb{U} \widehat{\boldsymbol{\Phi}}, \quad (136)$$

we can write $\widehat{H}$ in diagonal form

$$\widehat{H} = \widehat{\boldsymbol{\Psi}}^\dagger \mathbb{H} \widehat{\boldsymbol{\Psi}} = \widehat{\boldsymbol{\Phi}}^\dagger \mathbb{U}^\dagger \mathbb{H} \mathbb{U} \widehat{\boldsymbol{\Phi}} = \widehat{\boldsymbol{\Phi}}^\dagger \mathbb{E}_{\text{diag}} \widehat{\boldsymbol{\Phi}}. \quad (137)$$

Similarly to $\widehat{\boldsymbol{\Psi}}$, we can define new fermion operators $\hat{\boldsymbol{\gamma}}$ such that

$$\widehat{\boldsymbol{\Phi}} = \left(\begin{array}{c} \hat{\boldsymbol{\gamma}} \\ \hat{\boldsymbol{\gamma}}^\dagger \end{array}\right) = \mathbb{U}^\dagger \widehat{\boldsymbol{\Psi}} = \left(\begin{array}{cc} \mathbf{U}^\dagger & \mathbf{V}^\dagger \\ \mathbf{V}^{\mathsf{T}} & \mathbf{U}^{\mathsf{T}} \end{array}\right) \left(\begin{array}{c} \hat{\mathbf{c}} \\ \hat{\mathbf{c}}^\dagger \end{array}\right). \quad (138)$$

More explicitly, we can write:[25]

$$\begin{cases} \hat{\gamma}_\mu = \sum_{j=1}^L (\mathbf{U}_{j\mu}^* \hat{c}_j + \mathbf{V}_{j\mu}^* \hat{c}_j^\dagger), \\ \hat{\gamma}_\mu^\dagger = \sum_{j=1}^L (\mathbf{V}_{j\mu} \hat{c}_j + \mathbf{U}_{j\mu} \hat{c}_j^\dagger), \end{cases} \quad (140)$$

which can be easily inverted, remembering that $\widehat{\boldsymbol{\Psi}} = \mathbb{U} \widehat{\boldsymbol{\Phi}}$, to express the $\hat{c}_j$ operators in terms of the $\hat{\gamma}_\mu$:

$$\begin{cases} \hat{c}_j = \sum_\mu (\mathbf{U}_{j\mu} \hat{\gamma}_\mu + \mathbf{V}_{j\mu}^* \hat{\gamma}_\mu^\dagger), \\ \hat{c}_j^\dagger = \sum_\mu (\mathbf{V}_{j\mu} \hat{\gamma}_\mu + \mathbf{U}_{j\mu}^* \hat{\gamma}_\mu^\dagger). \end{cases} \quad (141)$$

---

[24]We have:

$$\{\widehat{\boldsymbol{\Phi}}_\mu, \widehat{\boldsymbol{\Phi}}_{\mu'}^\dagger\} = \{\sum_{j'} \mathbb{U}_{\mu j'}^\dagger \widehat{\boldsymbol{\Psi}}_{j'}, \sum_j \widehat{\boldsymbol{\Psi}}_j^\dagger \mathbb{U}_{j\mu'}\} = \sum_{jj'} \mathbb{U}_{\mu j'}^\dagger \mathbb{U}_{j\mu'} \{\widehat{\boldsymbol{\Psi}}_{j'}, \widehat{\boldsymbol{\Psi}}_j^\dagger\}$$

$$= \sum_j \mathbb{U}_{\mu j}^\dagger \mathbb{U}_{j\mu'} = (\mathbb{U}^\dagger \mathbb{U})_{\mu\mu'} = \delta_{\mu\mu'}.$$

[25]The conditions for the transformation in Eq. (140) to be canonical are:

$$\mathbb{U}^\dagger \mathbb{U} = \left(\begin{array}{c|c} \mathbf{U}^\dagger \mathbf{U} + \mathbf{V}^\dagger \mathbf{V} & \mathbf{U}^\dagger \mathbf{V}^* + \mathbf{V}^\dagger \mathbf{U}^* \\ \hline \mathbf{V}^{\mathsf{T}} \mathbf{U} + \mathbf{U}^{\mathsf{T}} \mathbf{V} & \mathbf{V}^{\mathsf{T}} \mathbf{V}^* + \mathbf{U}^{\mathsf{T}} \mathbf{U}^* \end{array}\right) = \left(\begin{array}{c|c} \mathbf{1} & \mathbf{0} \\ \hline \mathbf{0} & \mathbf{1} \end{array}\right) \quad \Rightarrow \quad \begin{cases} \mathbf{U}^\dagger \mathbf{U} + \mathbf{V}^\dagger \mathbf{V} = \mathbf{1}, \\ \mathbf{V}^{\mathsf{T}} \mathbf{U} + \mathbf{U}^{\mathsf{T}} \mathbf{V} = \mathbf{0}, \end{cases} \quad (139)$$

since you realise that the block 22 is simply the $*$ of block 11 and block 12 is the $\dagger$ of block 21. Interestingly, the condition $\mathbf{V}^{\mathsf{T}} \mathbf{U} + \mathbf{U}^{\mathsf{T}} \mathbf{V} = \mathbf{0}$ tells us that $\mathbf{V}^{\mathsf{T}} \mathbf{U}$ is anti-symmetric, and the same happens for $\mathbf{U}^{\mathsf{T}} \mathbf{V}$. From the fact that $\mathbb{U} \mathbb{U}^\dagger$ must also equal the identity matrix, we might deduce that $\mathbf{U} \mathbf{U}^\dagger + \mathbf{V}^* \mathbf{V}^{\mathsf{T}} = \mathbf{1}$ and $\mathbf{U} \mathbf{V}^\dagger + \mathbf{V}^* \mathbf{U}^{\mathsf{T}} = \mathbf{0}$.

Finally $\widehat{H}$ in terms of the $\hat{\boldsymbol{\gamma}}$ operators reads, assuming we have taken $\epsilon_\mu > 0$:

$$\widehat{H} = \sum_{\mu=1}^{L} \left( \epsilon_\mu \hat{\gamma}_\mu^\dagger \hat{\gamma}_\mu - \epsilon_\mu \hat{\gamma}_\mu \hat{\gamma}_\mu^\dagger \right) = \sum_{\mu=1}^{L} 2\epsilon_\mu \left( \hat{\gamma}_\mu^\dagger \hat{\gamma}_\mu - \frac{1}{2} \right), \tag{142}$$

and the ground state is the state annihilated by all $\hat{\gamma}_\mu$, which we denote by $|\emptyset_\gamma\rangle$:

$$\hat{\gamma}_\mu |\emptyset_\gamma\rangle = 0 \quad \forall\, \mu \quad \implies \quad \widehat{H}|\emptyset_\gamma\rangle = E_0 |\emptyset_\gamma\rangle, \quad \text{with} \quad E_0 = -\sum_{\mu=1}^{L} \epsilon_\mu. \tag{143}$$

The $2^L$ eigenstates can be expressed as:

$$|\psi_{\{n_\mu\}}\rangle = \prod_{\mu=1}^{L} \left( \hat{\gamma}_\mu^\dagger \right)^{n_\mu} |\emptyset_\gamma\rangle, \quad \text{with} \quad n_\mu = 0, 1,$$

$$E_{\{n_\mu\}} = E_0 + 2 \sum_\mu n_\mu \epsilon_\mu. \tag{144}$$

> ❖ **Warning:** The previous discussion applies to a generic quadratic fermion Hamiltonian $\widehat{H}$. Consequently, it also applies to the two different parity Hamiltonians $\widehat{\mathbb{H}}_{\mathrm{p}}$ relevant for the Ising case, which one could express as:
>
> $$\widehat{\mathbb{H}}_{\mathrm{p}} = \sum_{\mu=1}^{L} \left( \epsilon_{\mathrm{p},\mu} \hat{\gamma}_{\mathrm{p},\mu}^\dagger \hat{\gamma}_{\mathrm{p},\mu} - \epsilon_{\mathrm{p},\mu} \hat{\gamma}_{\mathrm{p},\mu} \hat{\gamma}_{\mathrm{p},\mu}^\dagger \right) = \sum_{\mu=1}^{L} 2\epsilon_{\mathrm{p},\mu} \left( \hat{\gamma}_{\mathrm{p},\mu}^\dagger \hat{\gamma}_{\mathrm{p},\mu} - \frac{1}{2} \right). \tag{145}$$
>
> This implies that there are two distinct Bogoliubov vacuum states $|\emptyset_{\mathrm{p}}\rangle$, one for each set of operators $\hat{\gamma}_{\mathrm{p},\mu}$. Recall, however, that the block Hamiltonian $\widehat{H}_{\mathrm{p}} = \widehat{P}_{\mathrm{p}} \widehat{\mathbb{H}}_{\mathrm{p}} \widehat{P}_{\mathrm{p}}$ involves projectors on the appropriate sub-sectors, which must be handled appropriately. Moreover, the possible presence of zero-energy eigenvalues must be appropriately taken care of: see below. This is important in calculating thermal averages, as further discussed in Sec. 11.

Before ending, a note on zero-energy eigenvalues, which has a practical relevance when calculating thermal averages. If you calculate the eigenvalues $\{\epsilon_\mu\}$ by a numerical diagonalization routine, the presence of zero-energy eigenvalues complicates the story. Indeed, the zero-energy eigenvalues, if present, must come in an *even number*. This is rather clear from the fact that the total dimension is $2L$ and that every non-zero positive eigenvalue $\epsilon_\mu > 0$ must have a negative partner $-\epsilon_\mu < 0$. Unfortunately, the computer will produce eigenvectors associated with the degenerate zero-energy eigenvalues which *do not* have the structure alluded at in Eq. (134). To enforce such a structure you can exploit the swap matrix $\mathbb{S}$.

ℹ **Info:** Let us consider the Ising case, where $\mathbb{H}$ is real and particle-hole symmetry reads $\mathbb{H}\mathbb{S} = -\mathbb{S}\mathbb{H}$. Hence if $\left(\mathbf{u}_\mu, \mathbf{v}_\mu\right)^{\mathsf{T}}$ is a zero-energy state, so is $\mathbb{S}\left(\mathbf{u}_\mu, \mathbf{v}_\mu\right)^{\mathsf{T}} = \left(\mathbf{v}_\mu, \mathbf{u}_\mu\right)^{\mathsf{T}}$. Hence the zero-energy subspace — the so-called $\mathrm{Ker}\,\mathbb{H}$, whose *even* dimension we denote by $N_0$ — is invariant for $\mathbb{S}$. Hence you can restrict $\mathbb{S}$ to such $N_0$-dimensional subspace, and diagonalize it there. But $\mathbb{S}$ is such that $\mathbb{S}^2 = \mathbb{1}$, hence its eigenvalues can be only $\pm 1$, the eigenstates of $\mathbb{S}$ being even or odd under swap of the first and last $L$ components. Even more: You can show that $\mathbb{S}$ must have exactly as many $+1$ as $-1$ eigenvalues in that subspace. Now, if $(\mathbf{u}, \mathbf{u})^{\mathsf{T}}$ is a zero-energy even-swap eigenstate, and $(\mathbf{v}, -\mathbf{v})^{\mathsf{T}}$ a zero-energy odd-swap eigenstate — both normalised and orthogonal — then the two combinations:

$$\frac{1}{\sqrt{2}}\begin{pmatrix} \mathbf{u}+\mathbf{v} \\ \mathbf{u}-\mathbf{v} \end{pmatrix}, \qquad \text{and} \qquad \frac{1}{\sqrt{2}}\begin{pmatrix} \mathbf{u}-\mathbf{v} \\ \mathbf{u}+\mathbf{v} \end{pmatrix}, \tag{146}$$

are both normalised, orthogonal, and have precisely the structure shown in Eq. (134). These states should be used to enforce the required structure of Eq. (134), crucial to fulfilling the correct anti-commutation rules.

---

**Problem 3. Tight-binding formulation of the BdG equations.**

Show that the static Bogoliubov-de Gennes equations in Eq. (133) are equivalent, for general couplings, to the diagonalization of the following tight-binding problem for the two-component spinor $\mathbf{W}_{j\mu} \stackrel{\text{def}}{=} \begin{pmatrix} \mathbf{U}_{j\mu} \\ \mathbf{V}_{j\mu} \end{pmatrix}$:

$$-\frac{J_j}{2}\left(\hat{\tau}^z + i\kappa\hat{\tau}^y\right)\mathbf{W}_{j+1,\mu} - \frac{J_{j-1}}{2}\left(\hat{\tau}^z - i\kappa\hat{\tau}^y\right)\mathbf{W}_{j-1,\mu} + h_j\hat{\tau}^z\mathbf{W}_{j\mu} = \epsilon_\mu\mathbf{W}_{j\mu},$$

where $\hat{\tau}$ are pseudo-spin Pauli matrices acting on the two components of $\mathbf{W}_{j\mu}$.

Next, consider the uniform case $J_j = J$ and $h_j = h$. Use Fourier transforms $\mathbf{W}_{j\mu} = \frac{1}{L}\sum_k e^{ikj}\mathbf{W}_{k\mu}$ (where the $k$-vectors used depend, as usual, from the boundary conditions) to show that:

$$\left(\mathbf{H}_k - 2\epsilon_\mu\right)\mathbf{W}_{k\mu} = 0,$$

where $\mathbf{H}_k = (2\kappa J \sin k)\hat{\tau}^y + 2(h - J\cos k)\hat{\tau}^z$ as in Eq. (52). This shows that the correct correspondence between the general BdG approach of Sec. 7 and the $k$-space approach of Sec. 4, is given by $2\epsilon_\mu \to \epsilon_k$.

---

**Problem 4. Bound-state excitations for an Ising chain with an impurity.**

Consider now a uniform Ising chain with $\kappa = 1$ and a single impurity on-site $j = l$. Take $J_j \equiv J > 0$, $h_j = h - h_{\text{imp}} \delta_{jl}$, with $0 < h_{\text{imp}} \ll h$, $h_{\text{imp}} \ll J$, $J \neq \pm h$. Show that the impurity induces two bound-state excitations, one above and one below the continuum $[2|J-h|, 2|J+h|]$ of the extended excitations of the uniform chain.

**Answer:** $\quad 2\epsilon_\mu^\pm = 2|J \pm h| \pm \frac{hJ}{|J \pm h|} \left(\frac{h_{\text{imp}}}{J}\right)^2 + o\left(\frac{h_{\text{imp}}}{J}\right)^2$.

**Hint:** Assuming that $2\epsilon_\mu$ is outside of the spectrum of the unperturbed uniform chain, and using Fourier transforms, show that you arrive at the following equation determining the non-trivial solutions $\epsilon_\mu$:

$$\left(1 - \frac{2h_{\text{imp}}}{L} \sum_k \left(\mathbf{H}_k - 2\epsilon_\mu\right)^{-1} \hat{\tau}^z\right) \mathbf{W}_{l\mu} = 0.$$

Assuming $h_{\text{imp}} \ll h$, show that, for $L \to \infty$, $1 \simeq 2h_{\text{imp}} \int_0^{2\pi} \frac{dk}{2\pi} \text{Tr}\left(\left(\mathbf{H}_k - 2\epsilon_\mu\right)^{-1} \hat{\tau}^z\right)$. Calculate the trace using $\mathbf{H}_k = \epsilon_k e^{i\theta_k \hat{\tau}^x/2} \hat{\tau}^z e^{-i\theta_k \hat{\tau}^x/2}$ with $\tan \theta_k = \frac{J \sin k}{h - J \cos k}$, and perform the integral over $k$. Give an expression for $\epsilon_\mu$ approximated to second order in $h_{\text{imp}}/J$.

---

The next problem helps in understanding that the presence of disorder changes the nature of the eigenstates, from being extended in space (plane-wave-like) to being space localized.

---

**Problem 5. Anderson localization of states for the disordered Ising chain.**

Consider the model with the disorder in both $J_j$ and $h_j$. Assume that $J_j \in [J_{\min}, 1]$ and $h_j \in [0, h_{\max}]$ are uniformly distributed, with $J_{\min} > 0$. Numerically solve the Bogoliubov-de Gennes equations Eq. (133) and show that, whatever the choice of $h_{\max}$ and $J_{\min}$, the spinor eigenfunctions $\mathbf{W}_{j\mu} \stackrel{\text{def}}{=} \begin{pmatrix} \mathbf{U}_{j\mu} \\ \mathbf{V}_{j\mu} \end{pmatrix}$ are localized in space. This means that these eigenfunctions are uniformly bounded by a function exponentially decaying over a characteristic length-scale $\xi_{\text{loc}}$, the so-called localization length. More formally, fixing $h_{\max}$ and $J_{\min}$, there exists a $\xi_{\text{loc}}$ such that

$$\sqrt{|\mathbf{U}_{j\mu}|^2 + |\mathbf{V}_{j\mu}|^2} \leq C e^{-|j - l_\mu|/\xi_{\text{loc}}}, \qquad \forall \mu, \tag{147}$$

where $l_\mu$ depends on $\mu$ and $C$ is a constant. This phenomenon can be seen when the system size exceeds the localization length, $L > \xi_{\text{loc}}$. Study localization also using the inverse participation ratio [78, 79]

$$\text{IPR}_\mu = \sum_j \left||\mathbf{W}_{j\mu}|^2\right|^2 = \sum_j \left||\mathbf{U}_{j\mu}|^2 + |\mathbf{V}_{j\mu}|^2\right|^2. \tag{148}$$

Average $\text{IPR}_\mu$ over $\mu$ and verify that it tends towards a constant value, for increasing $L$.[26]

---

The space localization phenomenon discussed in the previous problem is an example of *Anderson localization*, see Refs. [78,79]. The space localization of the eigenstates has profound consequences on the physics of the problem. As shown in Ref. [81], there is still a region of

---

[26]Observe that plane-wave delocalized states have IPR $= 1/L$, while fully localized states have IPR $= 1$. The problem of Anderson localization in the Kitaev model – the fermionic representation of the quantum ising chain – has been considered in [80].

transverse fields where the system shows ferromagnetic long-range order. Contrary to the ordered case — where long-range order is seen in the ground state only and is immediately lost in higher excited states — the presence of disorder, with the associated space-localized excitations, leads to the consequence that the whole spectrum shows long-ranged spin-spin correlations. See Ref. [82].

## 7.2 Open boundary conditions and Majorana fermions

The case of a chain with open boundary conditions is particularly interesting, because *Majorana fermions*, and the associated zero-energy modes, emerge quite naturally from the discussion [19]. In this section, we work out explicitly the case of a chain with open boundary conditions, introduce the Majorana fermions first as a formal device to perform the diagonalization, and then discuss the physical role they have as boundary excitations at vanishing energy in the broken symmetry phase.

For illustration purposes, let us consider the case in which the spin chain has $L = 4$ sites, and couplings $J_1, J_2, J_3 > 0$, while $J_4 = 0$ (as dictated by OBC).[27] The $2L \times 2L$ Hamiltonian matrix (in this case, an $8 \times 8$ matrix) will have the form (we fix the anisotropy parameter $\kappa = 1$):

$$\mathbb{H} = \frac{1}{2} \left( \begin{array}{cccc|cccc} 2h & -J_1 & 0 & 0 & 0 & -J_1 & 0 & 0 \\ -J_1 & 2h & -J_2 & 0 & J_1 & 0 & -J_2 & 0 \\ 0 & -J_2 & 2h & -J_3 & 0 & J_2 & 0 & -J_3 \\ 0 & 0 & -J_3 & 2h & 0 & 0 & J_3 & 0 \\ \hline 0 & J_1 & 0 & 0 & -2h & J_1 & 0 & 0 \\ -J_1 & 0 & J_2 & 0 & J_1 & -2h & J_2 & 0 \\ 0 & -J_2 & 0 & J_3 & 0 & J_2 & -2h & J_3 \\ 0 & 0 & -J_3 & 0 & 0 & 0 & J_3 & -2h \end{array} \right). \tag{149}$$

Let us consider first the case with $h = 0$, corresponding to the classical Ising model with the given couplings. The corresponding eigenvalues/eigenvectors (disregarding the ordering of the non-zero eigenvalues) are found to be:

$$\left| \begin{array}{cccc|cccc} J_1 & J_2 & J_3 & 0 & -J_1 & -J_2 & -J_3 & 0 \\ \hline -\frac{1}{2} & 0 & 0 & \frac{1}{\sqrt{2}} & \frac{1}{2} & 0 & 0 & 0 \\ \frac{1}{2} & -\frac{1}{2} & 0 & 0 & \frac{1}{2} & \frac{1}{2} & 0 & 0 \\ 0 & \frac{1}{2} & -\frac{1}{2} & 0 & 0 & \frac{1}{2} & \frac{1}{2} & 0 \\ 0 & 0 & \frac{1}{2} & 0 & 0 & 0 & \frac{1}{2} & -\frac{1}{\sqrt{2}} \\ \hline \frac{1}{2} & 0 & 0 & \frac{1}{\sqrt{2}} & -\frac{1}{2} & 0 & 0 & 0 \\ \frac{1}{2} & \frac{1}{2} & 0 & 0 & \frac{1}{2} & -\frac{1}{2} & 0 & 0 \\ 0 & \frac{1}{2} & \frac{1}{2} & 0 & 0 & \frac{1}{2} & -\frac{1}{2} & 0 \\ 0 & 0 & \frac{1}{2} & 0 & 0 & 0 & \frac{1}{2} & \frac{1}{\sqrt{2}} \end{array} \right|, \tag{150}$$

where you should observe that the structure of Eq. (134) is correctly respected, except for the two zero eigenvalues, which the diagonalization routine has decided to give us in this particular form. This form itself is particularly interesting. It suggests that the following fermionic

---

[27]If you want to do numerical tests, we suggest you take *different* values for $J_j$, for instance, $J_1 = 1$, $J_2 = 2$ and $J_3 = 3$, to avoid degeneracies, which might lead to a mixing of the corresponding eigenvectors.

combinations naturally emerge:

$$
\begin{aligned}
\epsilon_1 = J_1 &\rightarrow \widehat{\mathbf{\Phi}}_1 = \hat{\gamma}_1 = \tfrac{1}{2}(\hat{c}_2^\dagger + \hat{c}_2) + \tfrac{1}{2}(\hat{c}_1^\dagger - \hat{c}_1), \\
\epsilon_2 = J_2 &\rightarrow \widehat{\mathbf{\Phi}}_2 = \hat{\gamma}_2 = \tfrac{1}{2}(\hat{c}_3^\dagger + \hat{c}_3) + \tfrac{1}{2}(\hat{c}_2^\dagger - \hat{c}_2), \\
\epsilon_3 = J_3 &\rightarrow \widehat{\mathbf{\Phi}}_3 = \hat{\gamma}_3 = \tfrac{1}{2}(\hat{c}_4^\dagger + \hat{c}_4) + \tfrac{1}{2}(\hat{c}_3^\dagger - \hat{c}_3), \\
\epsilon_4 = 0 &\rightarrow \widehat{\mathbf{\Phi}}_4' = \frac{1}{\sqrt{2}}(\hat{c}_1^\dagger + \hat{c}_1), \\
-\epsilon_1 = -J_1 &\rightarrow \widehat{\mathbf{\Phi}}_5 = \hat{\gamma}_1^\dagger = \tfrac{1}{2}(\hat{c}_2^\dagger + \hat{c}_2) - \tfrac{1}{2}(\hat{c}_1^\dagger - \hat{c}_1), \\
-\epsilon_2 = -J_2 &\rightarrow \widehat{\mathbf{\Phi}}_6 = \hat{\gamma}_2^\dagger = \tfrac{1}{2}(\hat{c}_3^\dagger + \hat{c}_3) - \tfrac{1}{2}(\hat{c}_2^\dagger - \hat{c}_2), \\
-\epsilon_3 = -J_3 &\rightarrow \widehat{\mathbf{\Phi}}_7 = \hat{\gamma}_3^\dagger = \tfrac{1}{2}(\hat{c}_4^\dagger + \hat{c}_4) - \tfrac{1}{2}(\hat{c}_3^\dagger - \hat{c}_3), \\
-\epsilon_4 = 0 &\rightarrow \widehat{\mathbf{\Phi}}_8' = \frac{1}{\sqrt{2}}(\hat{c}_4^\dagger - \hat{c}_4).
\end{aligned}
\tag{151}
$$

Several things strike our attention. First: $\widehat{\mathbf{\Phi}}_4'$ is Hermitian and $\widehat{\mathbf{\Phi}}_8'$ is anti-Hermitian, and they are not Hermitian conjugate pairs, contrary to all other $(\widehat{\mathbf{\Phi}}_j, \widehat{\mathbf{\Phi}}_{j+4})$ pairs. If you want to construct ordinary fermionic operators, then you should redefine:

$$
\begin{aligned}
\widehat{\mathbf{\Phi}}_4' &\rightarrow \widehat{\mathbf{\Phi}}_4 = \hat{\gamma}_4 = \tfrac{1}{2}(\hat{c}_1^\dagger + \hat{c}_1) + \tfrac{1}{2}(\hat{c}_4^\dagger - \hat{c}_4), \\
\widehat{\mathbf{\Phi}}_8' &\rightarrow \widehat{\mathbf{\Phi}}_8 = \hat{\gamma}_4^\dagger = \tfrac{1}{2}(\hat{c}_1^\dagger + \hat{c}_1) - \tfrac{1}{2}(\hat{c}_4^\dagger - \hat{c}_4),
\end{aligned}
\tag{152}
$$

with an orthogonal transformation which leaves the subspace of two degenerate eigenvalues 0 invariant, precisely as alluded to in the last info-box of 7.1. Second: certain Hermitian combinations seem to play a peculiar role. In particular, let us define the *Majorana fermions*:[28]

$$
\check{c}_{j,1} = (\hat{c}_j^\dagger + \hat{c}_j), \qquad \text{and} \qquad \check{c}_{j,2} = i(\hat{c}_j^\dagger - \hat{c}_j).
\tag{154}
$$

These operators are manifestly Hermitian. They allow us to express the original fermions as:

$$
\hat{c}_j = \tfrac{1}{2}(\check{c}_{j,1} + i\check{c}_{j,2}), \qquad \text{and} \qquad \hat{c}_j^\dagger = \tfrac{1}{2}(\check{c}_{j,1} - i\check{c}_{j,2}),
\tag{155}
$$

and satisfy the anti-commutation relations:

$$
\{\check{c}_{j,\alpha}, \check{c}_{j',\alpha'}\} = 2\delta_{j,j'}\delta_{\alpha,\alpha'}.
\tag{156}
$$

Notice, in particular, that this implies that different Majorana anti-commute, but $(\check{c}_{j,\alpha})^2 = 1$.

In terms of these operators, we have:

$$
\begin{aligned}
\widehat{\mathbf{\Phi}}_1 = \hat{\gamma}_1 &= \tfrac{1}{2}(\hat{c}_2^\dagger + \hat{c}_2) + \tfrac{1}{2}(\hat{c}_1^\dagger - \hat{c}_1) = \tfrac{1}{2}(\check{c}_{2,1} - i\check{c}_{1,2}), \\
\widehat{\mathbf{\Phi}}_2 = \hat{\gamma}_2 &= \tfrac{1}{2}(\hat{c}_3^\dagger + \hat{c}_3) + \tfrac{1}{2}(\hat{c}_2^\dagger - \hat{c}_2) = \tfrac{1}{2}(\check{c}_{3,1} - i\check{c}_{2,2}), \\
\widehat{\mathbf{\Phi}}_3 = \hat{\gamma}_3 &= \tfrac{1}{2}(\hat{c}_4^\dagger + \hat{c}_4) + \tfrac{1}{2}(\hat{c}_3^\dagger - \hat{c}_3) = \tfrac{1}{2}(\check{c}_{4,1} - i\check{c}_{3,2}), \\
\widehat{\mathbf{\Phi}}_4 = \hat{\gamma}_4 &= \tfrac{1}{2}(\hat{c}_1^\dagger + \hat{c}_1) + \tfrac{1}{2}(\hat{c}_4^\dagger - \hat{c}_4) = \tfrac{1}{2}(\check{c}_{1,1} - i\check{c}_{4,2}), \\
\widehat{\mathbf{\Phi}}_5 = \hat{\gamma}_1^\dagger &= \tfrac{1}{2}(\hat{c}_2^\dagger + \hat{c}_2) - \tfrac{1}{2}(\hat{c}_1^\dagger - \hat{c}_1) = \tfrac{1}{2}(\check{c}_{2,1} + i\check{c}_{1,2}), \\
\widehat{\mathbf{\Phi}}_6 = \hat{\gamma}_2^\dagger &= \tfrac{1}{2}(\hat{c}_3^\dagger + \hat{c}_3) - \tfrac{1}{2}(\hat{c}_2^\dagger - \hat{c}_2) = \tfrac{1}{2}(\check{c}_{3,1} + i\check{c}_{2,2}), \\
\widehat{\mathbf{\Phi}}_7 = \hat{\gamma}_3^\dagger &= \tfrac{1}{2}(\hat{c}_4^\dagger + \hat{c}_4) - \tfrac{1}{2}(\hat{c}_3^\dagger - \hat{c}_3) = \tfrac{1}{2}(\check{c}_{4,1} + i\check{c}_{3,2}), \\
\widehat{\mathbf{\Phi}}_8 = \hat{\gamma}_4^\dagger &= \tfrac{1}{2}(\hat{c}_1^\dagger + \hat{c}_1) - \tfrac{1}{2}(\hat{c}_4^\dagger - \hat{c}_4) = \tfrac{1}{2}(\check{c}_{1,1} + i\check{c}_{4,2}).
\end{aligned}
\tag{157}
$$

---

[28]This definition is non-standard. The standard definition used by Kitaev [19] duplicates the sites and defines the Majorana fermions as living on even/odd sites as:

$$
\check{c}_{2j-1} = (\hat{c}_j^\dagger + \hat{c}_j), \qquad \text{and} \qquad \check{c}_{2j} = i(\hat{c}_j^\dagger - \hat{c}_j).
\tag{153}
$$

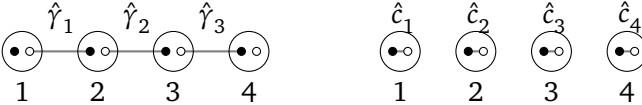

Figure 9: Left: An $L = 4$ open chain with the off-site Majorana pairing leading to the Bogoliubov vacuum. Right: The on-site Majorana pairing leads to the ordinary vacuum for $h_j > 0$.

There is something simple behind the previous story. If you rewrite the original Ising coupling in terms of fermions, you realize that for instance:

$$-J_j \hat{\sigma}_j^x \hat{\sigma}_{j+1}^x \rightarrow -J_j(\hat{c}_j^\dagger \hat{c}_{j+1} + \hat{c}_j^\dagger \hat{c}_{j+1}^\dagger + \text{H.c.}) = -J_j(\hat{c}_j^\dagger - \hat{c}_j)(\hat{c}_{j+1}^\dagger + \hat{c}_{j+1})$$
$$\equiv -iJ_j \check{c}_{j,2} \check{c}_{j+1,1}, \tag{158}$$

i.e., the Ising term couples in a precise way neighbouring Majorana operators. All we have done, to diagonalise it, is to introduce the appropriate combination $\hat{\gamma}_j = \frac{1}{2}(\check{c}_{j+1,1} - i\check{c}_{j,2})$ and $\hat{\gamma}_j^\dagger = \frac{1}{2}(\check{c}_{j+1,1} + i\check{c}_{j,2})$ and re-express the coupling term as:

$$-J_j \hat{\sigma}_j^x \hat{\sigma}_{j+1}^x \rightarrow -iJ_j \check{c}_{j,2} \check{c}_{j+1,1} = J_j(\hat{\gamma}_j^\dagger \hat{\gamma}_j - \hat{\gamma}_j \hat{\gamma}_j^\dagger), \tag{159}$$

which suggests that the ground state is the *vacuum* of those $\hat{\gamma}_j$ operators.[29]

There is a second simple case we can deal with. Take all $h_j > 0$ and $J_j = 0$, so that the Hamiltonian is now:

$$\widehat{H} = \sum_{j=1}^{L} h_j(2\hat{n}_j - 1) = \sum_{j=1}^{L} h_j(\hat{c}_j^\dagger \hat{c}_j - \hat{c}_j \hat{c}_j^\dagger) \rightarrow i \sum_{j=1}^{L} h_j \check{c}_{j,1} \check{c}_{j,2}. \tag{160}$$

This shows that the ground state, now the vacuum of the $\hat{c}_j$, still involves a "pairing" of Majorana fermions, but now *on the same site $j$*. The two different Majorana pairings are sketched in Fig. 9.

Returning to the previous case with $h_j = 0$, the ground states certainly verify

$$\hat{\gamma}_j |\emptyset\rangle = 0, \qquad \text{for} \qquad j = 1, \cdots, L-1(=3). \tag{161}$$

But there are *two states* satisfying such a condition, a degeneracy that is ultimately related to the presence of *unpaired* Majorana operators at the end of the chain, as emerging from Fig. 9 (left). Indeed, one such state is also the vacuum of $\hat{\gamma}_{L=4}$:

$$\hat{\gamma}_j |\emptyset_0\rangle = 0, \qquad \text{for} \qquad j = 1, \cdots, L(=4). \tag{162}$$

On such a state, we have (generalising now to arbitrary even $L$)

$$\hat{\gamma}_L \hat{\gamma}_L^\dagger |\emptyset_0\rangle = |\emptyset_0\rangle \qquad \Longrightarrow \qquad i\check{c}_{1,1} \check{c}_{L,2} |\emptyset_0\rangle = |\emptyset_0\rangle. \tag{163}$$

The second possible ground state is $|\emptyset_1\rangle = \hat{\gamma}_L^\dagger |\emptyset_0\rangle$ for which:

$$\hat{\gamma}_L^\dagger \hat{\gamma}_L |\emptyset_1\rangle = |\emptyset_1\rangle \qquad \Longrightarrow \qquad i\check{c}_{1,1} \check{c}_{L,2} |\emptyset_1\rangle = -|\emptyset_1\rangle. \tag{164}$$

---

[29]Interestingly, in the vacuum we gain an energy $-J_j$ from each bond. Breaking that, we would get a contribution $+J_j$ from the bond, hence an energy cost, referred to the vacuum, of $2J_j$: This explains, for instance, the factor 2 in front of $\epsilon_\mu$ in Eq. (142).

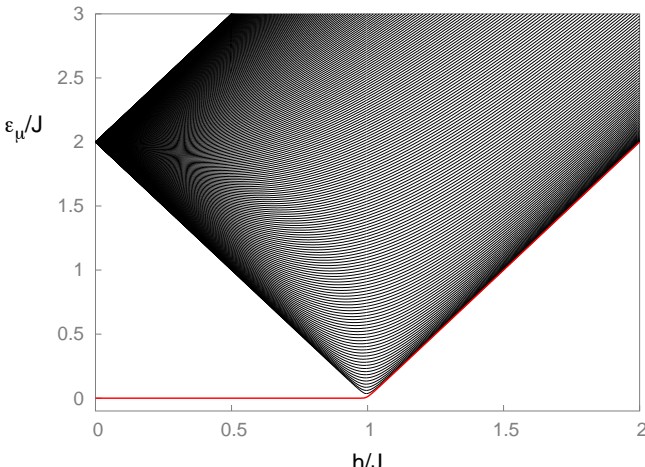

Figure 10: The spectrum of eigenvalues $\varepsilon_\mu = 2\epsilon_\mu \geq 0$ of an ordered Ising chain with OBC, versus the transverse field $h$. (We show only half of the particle-hole symmetric spectrum $\pm\epsilon_\mu$.) Here $L = 256$. Notice the zero-energy eigenvalue for $h < h_c = J$. This eigenvalue is exponentially small in the length $L$ for all $h < h_c$.

These two ground states have opposite fermion parity,[30] because they differ by the application of $\hat{\gamma}_L^\dagger$.

> 🛈 **Info:** As discussed by Kitaev [19], the two zero-modes survive for $0 < h < J$, with a splitting which is exponentially small in the length of the chain, as long the broken symmetry leads to two possible ground states.
>
> The existence of these modes is deeply related to the topological considerations done when discussing Fig. 3. Indeed, a fermionic chain with $|h| < J$ and OBC is equivalent to surrounding the chain with the fermionic vacuum — in turn, equivalent to an Ising chain with $h \to \infty$. But one cannot go continuously from a phase with a winding index of 1 to a phase with an index of 0. Therefore, at the border between two phases with different indexes, the gap must close to enforce this discontinuity (we saw when discussing Fig. 3, the deep connection between the discontinuity of the index and the closing of the gap). Hence, the gap must close at the boundary, and this effect appears as two zero-energy boundary modes which behave as Majorana excitations. As we saw above, there are only two ways of combining them into fermionic excitations, which are indeed very non-local objects. For any finite system size $L$, the two Majorana fermions have an overlap exponentially small in $L$. If we combine them into fermionic excitations, we find a gap between them which is exponentially small in the system size. This is the same gap we found in Secs. 4.1 and 4.2, which vanishes in the thermodynamic limit and leads to symmetry-breaking. Now we appreciate its intimate connection with topology.

---

[30]Notice, incidentally, that the fermionic parity can be expressed as:

$$\widehat{\mathcal{P}} = \prod_{j=1}^{L}(1 - 2\hat{n}_j) = \prod_{j=1}^{L}\left((\hat{c}_j^\dagger + \hat{c}_j)(\hat{c}_j^\dagger - \hat{c}_j)\right) = \prod_{j=1}^{L}(-i\check{c}_{j,1}\check{c}_{j,2}). \tag{165}$$

To visualize these facts, we show in Fig. 10 the spectrum of eigenvalues $\epsilon_\mu \geq 0$ (evaluated numerically) of an ordered Ising chain with OBC. We mark in red one of the two zero-energy modes we have discussed above, surviving for all $h \leq h_c = J$. The mode is not exactly at zero, but, rather, exponentially small in $L$. Finite-size effects (here $L = 256$) lead to a visible rounding effect in the proximity of $h_c$.

---

**Problem 6. Majorana fermion wave-functions.**

Consider a uniform Ising chain in a transverse field, with open boundary conditions. Consider the following expectation value of Majorana fermion operators:

$$\psi_{j,\alpha} = \langle \emptyset_1 | \check{c}_{j,\alpha} | \emptyset_0 \rangle \,,$$

where $|\emptyset_0\rangle$ is the Bogoljubov vacuum of the $\hat{\gamma}_\mu$ and $|\emptyset_1\rangle = \hat{\gamma}^\dagger_{\mu=0}|\emptyset_0\rangle$, with the convention that $\hat{\gamma}^\dagger_{\mu=0}$ creates a Bogoljubov fermion with eigenvalue $\epsilon_{\mu=0} = 0^+$, the "zero-energy" eigenvalue. By using the transformation between the $\hat{c}_j$ and $\hat{\gamma}_\mu$, show that:

$$\psi_{j,1} = \left( U^*_{j,\mu=0} + V^*_{j,\mu=0} \right), \qquad \text{and} \qquad \psi_{j,2} = i\left( U^*_{j,\mu=0} - V^*_{j,\mu=0} \right).$$

Show that $|\psi_{j,\alpha}|^2$ are both normalized to 1. By evaluating numerically the relevant quantities, plot $|\psi_{j,\alpha}|^2$ versus $j$ for a few values of the transverse field $h$, including $h = 0, 0.1, 0.9, 1$.

---

## 7.3 The BCS form of the ground state

The next problem we would like to solve is how to write the Bogoliubov vacuum $|\emptyset_\gamma\rangle$ in terms of the $\hat{c}^\dagger_j$ in the general non-homogeneous case, in a way that generalizes the simple BCS form we have in $k$-space:

$$|\emptyset_\gamma\rangle^{\text{ABC}} = \prod_{k>0}^{\text{ABC}} \left( u_k + v_k \hat{c}^\dagger_k \hat{c}^\dagger_{-k} \right)|0\rangle \,. \tag{166}$$

For that purpose, let us make the *Ansatz* that $|\emptyset_\gamma\rangle$ can be written as a Gaussian state of the form:

$$|\emptyset_\gamma\rangle = \mathcal{N} \exp\left( \frac{1}{2} \sum_{j_1 j_2} \mathbf{Z}_{j_1 j_2} \hat{c}^\dagger_{j_1} \hat{c}^\dagger_{j_2} \right)|0\rangle \equiv \mathcal{N}\, e^{\mathcal{Z}}\, |0\rangle \,, \tag{167}$$

where $\mathcal{Z}$ will be our shorthand notation for the quadratic fermion form we exponentiate. Clearly, since $\hat{c}^\dagger_{j_1} \hat{c}^\dagger_{j_2} = -\hat{c}^\dagger_{j_2} \hat{c}^\dagger_{j_1}$ we can take the matrix $\mathbf{Z}$ to be *antisymmetric* (but complex, in general): Any symmetric part of $\mathbf{Z}$ would give no contribution. The conditions that $\mathbf{Z}$ has to satisfy should be inferred from the fact that we require that $\hat{\gamma}_\mu |\emptyset_\gamma\rangle = 0$, hence:

$$\mathcal{N} \sum_{j=1}^{L} \left( \mathbf{U}^*_{j\mu} \hat{c}_j + \mathbf{V}^*_{j\mu} \hat{c}^\dagger_j \right) e^{\mathcal{Z}} |0\rangle = 0\,, \qquad \forall\, \mu\,. \tag{168}$$

Since $\mathcal{Z}$ is made of *pairs* of $\hat{c}^\dagger$s, it commutes with $\hat{c}^\dagger_j$, hence, $\hat{c}^\dagger_j e^{\mathcal{Z}}|0\rangle = e^{\mathcal{Z}} \hat{c}^\dagger_j |0\rangle$. The first term, containing $\hat{c}_j e^{\mathcal{Z}}|0\rangle$, is more problematic. We would like to commute $\hat{c}_j$ through $e^{\mathcal{Z}}$ to bring it

towards the fermionic vacuum state $|0\rangle$, where it annihilates. To do so, let us start calculating:

$$\left[\hat{c}_j, \mathcal{Z}\right] = \frac{1}{2}\left[\hat{c}_j, \sum_{j_1 j_2} \mathbf{Z}_{j_1 j_2} \hat{c}_{j_1}^\dagger \hat{c}_{j_2}^\dagger\right] = \sum_{j'} \mathbf{Z}_{jj'} \hat{c}_{j'}^\dagger, \tag{169}$$

where we have used the antisymmetry of $\mathbf{Z}$. We see, therefore, that $[\hat{c}_j, \mathcal{Z}]$, being a combination of $\hat{c}_{j'}^\dagger$ commutes with $\mathcal{Z}$ and with any function of $\mathcal{Z}$. It takes then little algebra[31] to show that:

$$\left[\hat{c}_j, e^{\mathcal{Z}}\right] = \left[\hat{c}_j, \mathcal{Z}\right] e^{\mathcal{Z}} = e^{\mathcal{Z}}\left[\hat{c}_j, \mathcal{Z}\right] \quad \Rightarrow \quad \hat{c}_j e^{\mathcal{Z}} = e^{\mathcal{Z}}\left(\hat{c}_j + [\hat{c}_j, \mathcal{Z}]\right). \tag{170}$$

The conditions in Eq. (168) therefore read:

$$\mathcal{N} e^{\mathcal{Z}} \sum_{j=1}^{L}\left(\mathbf{U}_{j\mu}^*\left(\hat{c}_j + [\hat{c}_j, \mathcal{Z}]\right) + \mathbf{V}_{j\mu}^* \hat{c}_j^\dagger\right)|0\rangle = 0, \qquad \forall \mu. \tag{171}$$

Noticing that $\hat{c}_j|0\rangle = 0$, substituting Eq. (169), and omitting irrelevant prefactors we therefore have:

$$\left(\sum_{jj'} \mathbf{U}_{j'\mu}^* \mathbf{Z}_{j'j} \hat{c}_j^\dagger + \sum_j \mathbf{V}_{j\mu}^* \hat{c}_j^\dagger\right)|0\rangle = 0, \qquad \forall \mu, \tag{172}$$

where we have exchanged the dummy indices $j$ and $j'$ in the first term. Next, we collect the two terms by writing:

$$\sum_j \left((\mathbf{U}^\dagger \mathbf{Z})_{\mu j} + (\mathbf{V}^\dagger)_{\mu j}\right) \hat{c}_j^\dagger |0\rangle = 0 \quad \Rightarrow \quad \mathbf{Z} = -(\mathbf{U}^\dagger)^{-1} \mathbf{V}^\dagger. \tag{173}$$

This is the condition that $\mathbf{Z}$ has to verify for the state $|\emptyset_\gamma\rangle$ to be annihilated by all $\hat{\gamma}_\mu$. This is the so-called *Thouless formula* [83]. It takes very little algebra[32] to verify that, indeed, such a form of $\mathbf{Z}$ is *antisymmetric*.

We will see in Sec. 8.1, see Eq. (207), that the Gaussian form just derived applies also to a time-dependent state $|\psi(t)\rangle$ when the dynamics follows a unitary Schrödinger evolution with an Hamiltonian $\widehat{H}(t)$ which is quadratic in the fermionic operators.

According to a theorem of linear algebra [84] any antisymmetric matrix can always be reduced to a "standard canonical" form by applying a unitary matrix $\mathbf{D}$ as follows:

$$\mathbf{Z} = \mathbf{D}\boldsymbol{\Lambda}\mathbf{D}^{\mathrm{T}}, \qquad \text{with} \qquad \boldsymbol{\Lambda} = \begin{pmatrix} 0 & \lambda_1 & 0 & 0 & \cdots \\ -\lambda_1 & 0 & 0 & 0 & \cdots \\ 0 & 0 & 0 & \lambda_2 & \cdots \\ 0 & 0 & -\lambda_2 & 0 & \cdots \\ \vdots & \vdots & \vdots & \vdots & \vdots \end{pmatrix}_{L \times L}, \tag{175}$$

---

[31]Simply expand the exponential in the usual way, realise that

$$[\hat{c}_j, \mathcal{Z}^n] = n[\hat{c}_j, \mathcal{Z}]\mathcal{Z}^{n-1},$$

because $[\hat{c}_j, \mathcal{Z}]$ commutes with all powers of $\mathcal{Z}$, and reconstruct the exponential to get the result.

[32]Observe that:

$$\mathbf{Z}^{\mathrm{T}} = -(\mathbf{V}^\dagger)^{\mathrm{T}}\left((\mathbf{U}^\dagger)^{-1}\right)^{\mathrm{T}} = -\mathbf{V}^*\left((\mathbf{U}^\dagger)^{\mathrm{T}}\right)^{-1} = -\mathbf{V}^*(\mathbf{U}^*)^{-1}.$$

However, from block 12 in Eq. (139) we get:

$$\mathbf{U}^\dagger \mathbf{V}^* = -\mathbf{V}^\dagger \mathbf{U}^* \quad \Rightarrow \quad \mathbf{Z}^{\mathrm{T}} = -\mathbf{V}^*(\mathbf{U}^*)^{-1} = (\mathbf{U}^\dagger)^{-1}\mathbf{V}^\dagger = -\mathbf{Z}. \tag{174}$$

where in general the $\lambda_p$ are complex. If $L$ is *even*, there are $\frac{L}{2}$ blocks $2 \times 2$ with some $\lambda_p$, while if $L$ is *odd*, $\Lambda$ has an extra row/column of zeroes. The unitary matrix $\mathbf{D}$ allows us to define combinations of the fermions $c_j^\dagger$ which form natural "BCS-paired" orbitals,

$$\hat{d}_p^\dagger = \sum_j (\mathbf{D}^\mathsf{T})_{pj} \hat{c}_j^\dagger = \sum_j \mathbf{D}_{jp} \hat{c}_j^\dagger . \tag{176}$$

Labelling the consecutive columns of $\mathbf{D}$ as $1, \overline{1}, 2, \overline{2}, \cdots, p, \overline{p}, \cdots$, with $p$ up to $L/2$, one can readily check that in terms of the $d^\dagger$s the Bogoliubov vacuum $|\emptyset_\gamma\rangle$ reads:

$$|\emptyset_\gamma\rangle = \mathcal{N} \, \exp\Big( \sum_{p=1}^{L/2} \lambda_p \hat{d}_p^\dagger \hat{d}_{\overline{p}}^\dagger \Big) |0\rangle = \mathcal{N} \prod_{p=1}^{L/2} \Big( 1 + \lambda_p \hat{d}_p^\dagger \hat{d}_{\overline{p}}^\dagger \Big) |0\rangle . \tag{177}$$

It remains to evaluate the normalization constant $\mathcal{N}$. Now we calculate:[33]

$$
\begin{aligned}
1 = \langle \emptyset_\gamma | \emptyset_\gamma \rangle &= |\mathcal{N}|^2 \, \langle 0| \prod_{p=1}^{L/2} \Big( 1 + \lambda_p^* \hat{d}_{\overline{p}} \hat{d}_p \Big) \Big( 1 + \lambda_p \hat{d}_p^\dagger \hat{d}_{\overline{p}}^\dagger \Big) |0\rangle \\
&= |\mathcal{N}|^2 \prod_{p=1}^{L/2} \Big( 1 + |\lambda_p|^2 \Big) = |\mathcal{N}|^2 \Big[ \det\Big( \mathbf{1} + \mathbf{\Lambda}\mathbf{\Lambda}^\dagger \Big) \Big]^{1/2} = |\mathcal{N}|^2 \Big[ \det\Big( \mathbf{1} + \mathbf{Z}\mathbf{Z}^\dagger \Big) \Big]^{1/2} \\
&= |\mathcal{N}|^2 \Big[ \det\Big( \mathbf{1} + (\mathbf{U}^\dagger)^{-1} \mathbf{V}^\dagger \mathbf{V} \mathbf{U}^{-1} \Big) \Big]^{1/2} \\
&= |\mathcal{N}|^2 \Big[ \det\Big( (\mathbf{U}^\dagger)^{-1} (\mathbf{U}^\dagger \mathbf{U} + \mathbf{V}^\dagger \mathbf{V}) \mathbf{U}^{-1} \Big) \Big]^{1/2} = |\mathcal{N}|^2 \Big[ \det\Big( (\mathbf{U}\mathbf{U}^\dagger)^{-1} \Big) \Big]^{1/2} \\
&= |\mathcal{N}|^2 \frac{1}{|\det(\mathbf{U})|} \qquad \Rightarrow \qquad |\mathcal{N}| = \sqrt{|\det(\mathbf{U})|} .
\end{aligned}
\tag{178}
$$

Summarising, we have derived the so-called *Onishi formula* [83], which states that:

$$\Big| \langle 0 | \emptyset_\gamma \rangle \Big|^2 = |\mathcal{N}|^2 = |\det(\mathbf{U})| . \tag{179}$$

If we express the Bogoliubov vacuum in terms of the $\lambda_p$ we have:

$$|\emptyset_\gamma\rangle = \prod_{p=1}^{L/2} \frac{1}{\sqrt{1 + |\lambda_p|^2}} \Big( 1 + \lambda_p \hat{d}_p^\dagger \hat{d}_{\overline{p}}^\dagger \Big) |0\rangle = \prod_{p=1}^{L/2} \Big( u_p + v_p \hat{d}_p^\dagger \hat{d}_{\overline{p}}^\dagger \Big) |0\rangle , \tag{180}$$

where we have defined $u_p = 1/\sqrt{1 + |\lambda_p|^2}$ and $v_p = \lambda_p/\sqrt{1 + |\lambda_p|^2}$, which verify $|u_p|^2 + |v_p|^2 = 1$.

Further details about the overlap between BCS states of the form discussed previously are given in Appendix A.

---

[33]In the derivation we use that:

$$
\mathbf{\Lambda}\mathbf{\Lambda}^\dagger = \begin{pmatrix}
|\lambda_1|^2 & 0 & 0 & 0 & \cdots \\
0 & |\lambda_1|^2 & 0 & 0 & \cdots \\
0 & 0 & |\lambda_2|^2 & 0 & \cdots \\
0 & 0 & 0 & |\lambda_2|^2 & \cdots \\
\vdots & \vdots & \vdots & \vdots & \vdots
\end{pmatrix}_{L \times L} .
$$

# 8 Schrödinger dynamics in the time-dependent case

A time dependence of the Hamiltonian can come from many different sources [7, 9, 10, 85]. The simplest case, which is used in the so-called *quantum annealing* approach [86–88], consists in assuming that the transverse fields are time-dependent $h_j(t)$: for instance, they might be slowly annealed from a very large value towards zero. Alternatively, the Hamiltonian couplings might be changed in some time-periodic fashion [89], as further discussed in Sec. 8.3. In all these cases, the elements of the matrices **A** and **B** become time-dependent and consequently $\widehat{H} \to \widehat{H}(t)$. We proceed now in general, assuming $\mathbf{A}(t)$ and $\mathbf{B}(t)$.

Start from Schrödinger's equation:

$$i\hbar \frac{d}{dt}|\psi(t)\rangle = \widehat{H}(t)|\psi(t)\rangle. \tag{181}$$

Since the norm of $|\psi(t)\rangle$ must be conserved, this implies the existence of a unitary evolution operator $\widehat{U}(t, t_0)$ such that $|\psi(t)\rangle = \widehat{U}(t, t_0)|\psi(t_0)\rangle$, which satisfies the same equation:

$$i\hbar \frac{d}{dt}\widehat{U}(t, t_0) = \widehat{H}(t)\widehat{U}(t, t_0), \qquad \text{with} \quad \widehat{U}(t_0, t_0) = \hat{1}. \tag{182}$$

Next, consider the expectation value of a time-dependent operator $\hat{O}(t)$ in the Schrödinger's picture

$$\langle \hat{O}(t) \rangle \equiv \langle \psi(t)|\hat{O}(t)|\psi(t)\rangle = \langle \psi(t_0)|\widehat{U}^\dagger(t, t_0)\hat{O}(t)\widehat{U}(t, t_0)|\psi(t_0)\rangle$$
$$\equiv \langle \psi(t_0)|\hat{O}_{\text{H}}(t)|\psi(t_0)\rangle, \tag{183}$$

where we have introduced Heisenberg's picture operator

$$\hat{O}_{\text{H}}(t) \equiv \widehat{U}^\dagger(t, t_0)\hat{O}(t)\widehat{U}(t, t_0). \tag{184}$$

Therefore the equation of motion of an operator in Heisenberg's picture for the general case of a time-dependent Hamiltonian reads:[34]

$$i\hbar \frac{d}{dt}\hat{O}_{\text{H}}(t) = \widehat{U}^\dagger(t, t_0)\left( \left[\hat{O}(t), \widehat{H}(t)\right] + i\hbar \frac{\partial}{\partial t}\hat{O}(t) \right)\widehat{U}(t, t_0). \tag{186}$$

## 8.1 The time-dependent Bogoliubov-de Gennes equations

Let's write Heisenberg's equation of motion for operator $\hat{c}_j$

$$i\hbar \frac{d}{dt}\hat{c}_{j_{\text{H}}}(t) = \widehat{U}^\dagger(t, t_0)\left[\hat{c}_j, \widehat{H}(t)\right]\widehat{U}(t, t_0). \tag{187}$$

---

[34]Here we use:

$$i\hbar \frac{d}{dt}\widehat{U}(t, t_0) = \widehat{H}(t)\widehat{U}(t, t_0), \quad \text{and} \quad -i\hbar \frac{d}{dt}\widehat{U}^\dagger(t, t_0) = \widehat{U}^\dagger(t, t_0)\widehat{H}(t).$$

Notice that if $\widehat{H}$ and $\hat{O}$ are time-independent

$$\left[\widehat{U}, \widehat{H}\right] = \left[\widehat{U}^\dagger, \widehat{H}\right] = 0, \quad \text{and} \quad i\hbar \frac{\partial}{\partial t}\hat{O} = 0,$$

then Eq. (186) takes the well-known form:

$$i\hbar \frac{d}{dt}\hat{O}_{\text{H}} = \left[\hat{O}_{\text{H}}, \widehat{H}\right]. \tag{185}$$

By calculating the commutator

$$
\begin{aligned}
\left[\hat{c}_j, \widehat{H}(t)\right] &= \sum_{l,l'=1}^{2L} \mathbb{H}_{ll'}(t)\left[\hat{c}_j, \widehat{\boldsymbol{\Psi}}_l^\dagger \widehat{\boldsymbol{\Psi}}_{l'}\right] \\
&= \sum_{l,l'=1}^{2L} \mathbb{H}_{ll'}(t)\left(\left\{\hat{c}_j, \widehat{\boldsymbol{\Psi}}_l^\dagger\right\}\widehat{\boldsymbol{\Psi}}_{l'} - \widehat{\boldsymbol{\Psi}}_l^\dagger\left\{\hat{c}_j, \widehat{\boldsymbol{\Psi}}_{l'}\right\}\right) \\
&= \sum_{l,l'=1}^{2L} \mathbb{H}_{ll'}(t)\left(\delta_{l,j}\widehat{\boldsymbol{\Psi}}_{l'} - \widehat{\boldsymbol{\Psi}}_l^\dagger \delta_{l',j+L}\right) \\
&= 2\sum_{j'=1}^{L}\left(\mathbf{A}_{jj'}(t)\hat{c}_{j'} + \mathbf{B}_{jj'}(t)\hat{c}_{j'}^\dagger\right),
\end{aligned}
\tag{188}
$$

we see that we have a *linear* equation of motion

$$
i\hbar\frac{d}{dt}\hat{c}_{j\text{H}}(t) = 2\sum_{j'=1}^{L}\left(\mathbf{A}_{jj'}(t)\,\hat{c}_{j'\text{H}}(t) + \mathbf{B}_{jj'}(t)\,\hat{c}_{j'\text{H}}^\dagger(t)\right),
\tag{189}
$$

and analogously for the operator $\hat{c}_j^\dagger$. With a more compact notation, one can write the linear Heisenberg equations of motion for the elementary fermionic operators as:

$$
i\hbar\frac{d}{dt}\widehat{\boldsymbol{\Psi}}_{\text{H}}(t) = 2\,\mathbb{H}(t)\widehat{\boldsymbol{\Psi}}_{\text{H}}(t),
\tag{190}
$$

the factor 2 on the right-hand side originating from the off-diagonal $\{\widehat{\boldsymbol{\Psi}}_j, \widehat{\boldsymbol{\Psi}}_{j+L}\} = 1$ for $j = 1\cdots L$. The initial condition for these equations can be written as:

$$
\widehat{\boldsymbol{\Psi}}_{\text{H}}(t = t_0) \equiv \widehat{\boldsymbol{\Psi}} = \mathbb{U}_0\begin{pmatrix}\hat{\boldsymbol{\gamma}}\\\hat{\boldsymbol{\gamma}}^\dagger\end{pmatrix} = \mathbb{U}_0\,\widehat{\boldsymbol{\Phi}},
\tag{191}
$$

where $\hat{\boldsymbol{\gamma}}$ are the Bogoliubov fermions that diagonalise $\widehat{H}(t_0)$, and $\mathbb{U}_0$ the corresponding rotation matrix.

We are not quite done: We have an explicit linear equation for $\widehat{\boldsymbol{\Psi}}_{\text{H}}(t)$, but we need an *explicit solution* for this equation, obtained by some "simple enough" integration of a finite-dimensional linear problem. There are now at least two ways of getting the desired result.

**First route.** We make the *Ansatz* that $|\psi(t)\rangle$, the time-evolved state of the system, is a *Bogoliubov vacuum* annihilated by a set of *time-dependent* quasi-particle annihilation operators $\hat{\gamma}_\mu(t)$

$$
\hat{\gamma}_\mu(t)\,|\psi(t)\rangle = 0, \qquad \forall\,\mu, \quad \forall\,t.
\tag{192}
$$

This requirement immediately implies, by taking a total time derivative, that:

$$
\begin{aligned}
0 &= i\hbar\frac{d}{dt}\left(\hat{\gamma}_\mu(t)\,|\psi(t)\rangle\right) \\
&= \left(i\hbar\frac{\partial}{\partial t}\hat{\gamma}_\mu(t)\right)|\psi(t)\rangle + \hat{\gamma}_\mu(t)\left(i\hbar\frac{d}{dt}|\psi(t)\rangle\right) \\
&= \left(i\hbar\frac{\partial}{\partial t}\hat{\gamma}_\mu(t) + \hat{\gamma}_\mu(t)\widehat{H}(t) - \widehat{H}(t)\hat{\gamma}_\mu(t)\right)|\psi(t)\rangle,
\end{aligned}
\tag{193}
$$

where we have added, in the last step, a term $\hat{\gamma}_\mu(t)\,|\psi(t)\rangle = 0$. The last expression implies:[35]

$$i\hbar\frac{\partial}{\partial t}\hat{\gamma}_\mu(t) = -\left[\hat{\gamma}_\mu(t),\widehat{H}(t)\right]. \tag{194}$$

By considering the equation of motion of the Heisenberg operator $\hat{\gamma}_{\mu\mathrm{H}}(t)$ we have

$$i\hbar\frac{d}{dt}\hat{\gamma}_{\mu\mathrm{H}}(t) = \widehat{U}^\dagger(t,t_0)\left(\left[\hat{\gamma}_\mu(t),\widehat{H}(t)\right] + i\hbar\frac{\partial}{\partial t}\hat{\gamma}_\mu(t)\right)\widehat{U}(t,t_0) \equiv 0, \tag{195}$$

where we have used Eq. (194) in the last step. So, since $\hat{\gamma}_{\mu\mathrm{H}}$ does not depend on $t$, it must coincide with its $t = t_0$ value. Let's call this value $\hat{\gamma}_\mu = \hat{\gamma}_{\mu\mathrm{H}} = \hat{\gamma}_\mu(t = t_0)$.

Let us assume now, inspired by Eq. (141), that the $\hat{c}_{j\mathrm{H}}(t)$ are indeed expressed by

$$\hat{c}_{j\mathrm{H}}(t) = \sum_{\mu=1}^{L}\left(\mathbf{U}_{j\mu}(t)\,\hat{\gamma}_\mu + \mathbf{V}^*_{j\mu}(t)\,\hat{\gamma}^\dagger_\mu\right), \tag{196}$$

and let us see if this expression solves the required Heisenberg equations in Eq. (189) for an appropriate choice of the time-dependent coefficients $\mathbf{U}_{j\mu}(t)$ and $\mathbf{V}_{j\mu}(t)$. Substituting in Eq. (189) we get:

$$\sum_{\mu=1}^{L}\left(i\hbar\left(\frac{d}{dt}\mathbf{U}_{j\mu}(t)\right)\hat{\gamma}_\mu + i\hbar\left(\frac{d}{dt}\mathbf{V}^*_{j\mu}(t)\right)\hat{\gamma}^\dagger_\mu\right) = 2\sum_{j=1}^{L}\mathbf{A}_{ij}(t)\left(\mathbf{U}_{j\mu}(t)\hat{\gamma}_\mu + \mathbf{V}^*_{j\mu}(t)\hat{\gamma}^\dagger_\mu\right)$$
$$+ 2\sum_{j=1}^{L}\mathbf{B}_{ij}(t)\left(\mathbf{V}_{j\mu}(t)\hat{\gamma}_\mu + \mathbf{U}^*_{j\mu}(t)\hat{\gamma}^\dagger_\mu\right). \tag{197}$$

By equating the coefficients of $\hat{\gamma}_\mu$ and $\hat{\gamma}^\dagger_\mu$ we obtain the *time-dependent Bogoliubov-de Gennes equations*:

$$\begin{cases} i\hbar\dfrac{d}{dt}\mathbf{U}_{j\mu}(t) = \ \ 2\displaystyle\sum_{j'=1}^{L}\left(\mathbf{A}_{jj'}(t)\mathbf{U}_{j'\mu}(t) + \mathbf{B}_{jj'}(t)\mathbf{V}_{j'\mu}(t)\right), \\[4mm] i\hbar\dfrac{d}{dt}\mathbf{V}_{j\mu}(t) = -2\displaystyle\sum_{j'=1}^{L}\left(\mathbf{B}^*_{jj'}(t)\mathbf{U}_{j'\mu}(t) + \mathbf{A}^*_{jj'}(t)\mathbf{V}_{j'\mu}(t)\right), \end{cases} \tag{198}$$

or more compactly, collecting together $\mu = 1,\cdots,L$ solutions in $L \times L$ blocks $\mathbf{U}$ and $\mathbf{V}$:[36]

$$i\hbar\frac{d}{dt}\begin{pmatrix}\mathbf{U}(t)\\ \mathbf{V}(t)\end{pmatrix} = 2\,\mathbb{H}(t)\begin{pmatrix}\mathbf{U}(t)\\ \mathbf{V}(t)\end{pmatrix}. \tag{200}$$

Notice that if $\left(\mathbf{u}_\mu(t),\mathbf{v}_\mu(t)\right)^{\mathsf{T}}$ is solution of Eq. (198) then $\left(\mathbf{v}^*_\mu(t),\mathbf{u}^*_\mu(t)\right)^{\mathsf{T}}$ is also a solution, so we need to find only $\mu = 1,\cdots,L$ solutions, as indeed alluded by the compact form (200), not $2L$. Once we have the first $L$, it is automatically guaranteed that:

$$i\hbar\frac{d}{dt}\begin{pmatrix}\mathbf{U}(t) & \mathbf{V}^*(t)\\ \mathbf{V}(t) & \mathbf{U}^*(t)\end{pmatrix} = 2\,\mathbb{H}(t)\begin{pmatrix}\mathbf{U}(t) & \mathbf{V}^*(t)\\ \mathbf{V}(t) & \mathbf{U}^*(t)\end{pmatrix}. \tag{201}$$

---

[35]A mathematician would complain, here, that this is not a valid implication: An arbitrary linear combination of $\hat{\gamma}_\mu(t)$ could be added that, acting on $|\psi(t)\rangle$, gives 0. We are a bit swift here, but the result is correct. We will get to the same result by a second route in a short while.

[36]In the time-independent case, the solution is equivalent to solving the time-independent Bogoliubov-de Gennes equations. Indeed in this case the time evolution of the solution is

$$\mathbb{H}\begin{pmatrix}\mathbf{u}_\mu\\ \mathbf{v}_\mu\end{pmatrix} = \epsilon_\mu\begin{pmatrix}\mathbf{u}_\mu\\ \mathbf{v}_\mu\end{pmatrix} \quad\Rightarrow\quad \begin{pmatrix}\mathbf{u}_\mu(t)\\ \mathbf{v}_\mu(t)\end{pmatrix} = \mathrm{e}^{-2i\epsilon_\mu t/\hbar}\begin{pmatrix}\mathbf{u}_\mu\\ \mathbf{v}_\mu\end{pmatrix}, \tag{199}$$

and, as you can easily verify, the same result can be obtained by using directly Eq. (200) with $\mathbb{H}(t) = \mathbb{H}$.

**Second route.** It is reassuring to get to the same time-dependent Bogoliubov-de Gennes equations by a second, quicker, route. Let us recall the linear equation we want to solve, with its initial condition:

$$i\hbar\frac{d}{dt}\widehat{\mathbf{\Psi}}_{\text{H}}(t) = 2\,\mathbb{H}(t)\widehat{\mathbf{\Psi}}_{\text{H}}(t)\,,$$

$$\widehat{\mathbf{\Psi}}_{\text{H}}(t=t_0) \equiv \widehat{\mathbf{\Psi}} \;=\; \mathbb{U}_0\left(\begin{array}{c}\hat{\boldsymbol{\gamma}}\\\hat{\boldsymbol{\gamma}}^\dagger\end{array}\right) = \mathbb{U}_0\,\widehat{\mathbf{\Phi}}\,,$$

where $\hat{\boldsymbol{\gamma}}$ are the Bogoliubov fermions that diagonalise $\widehat{H}(t_0)$, and $\mathbb{U}_0$ the corresponding $2L\times2L$ transformation matrix. Inspired by the form of the initial condition, let us search for a solution of the same form:

$$\widehat{\mathbf{\Psi}}_{\text{H}}(t) = \mathbb{U}(t)\left(\begin{array}{c}\hat{\boldsymbol{\gamma}}\\\hat{\boldsymbol{\gamma}}^\dagger\end{array}\right) = \mathbb{U}(t)\,\widehat{\mathbf{\Phi}}\,, \tag{202}$$

with the *same* $\widehat{\mathbf{\Phi}}$ used to diagonalise the initial $t=t_0$ problem. For this to be a solution, the time-dependent coefficients $\mathbb{U}(t)$ must satisfy the linear Bogoliubov-de Gennes time-dependent equations:

$$i\hbar\frac{d}{dt}\mathbb{U}(t) = 2\,\mathbb{H}(t)\,\mathbb{U}(t)\,, \tag{203}$$

with initial conditions $\mathbb{U}(t=t_0) = \mathbb{U}_0$. The latter form is just a compact way of expressing Eq. (201).

It is easy to verify that this implies that the operators $\hat{\gamma}_\mu(t)$ in the Schrödinger picture are time-dependent and annihilate the state $|\psi(t)\rangle$: this was indeed the starting point of the Bogoliubov *Ansatz* presented in the first route. The *Ansatz* here is that the Heisenberg operators associated with the Bogoliubov fermions are *time-independent*, coinciding with the original $\hat{\gamma}_\mu$ at time $t=t_0$. This amounts to saying that:

$$\left(\begin{array}{c}\hat{\boldsymbol{\gamma}}_{\text{H}}\\\hat{\boldsymbol{\gamma}}_{\text{H}}^\dagger\end{array}\right) = \left(\begin{array}{c}\hat{\boldsymbol{\gamma}}\\\hat{\boldsymbol{\gamma}}^\dagger\end{array}\right) = \mathbb{U}^\dagger(t)\left(\begin{array}{c}\hat{\mathbf{c}}_{\text{H}}(t)\\\hat{\mathbf{c}}_{\text{H}}^\dagger(t)\end{array}\right)\,. \tag{204}$$

By linearity, it follows that, in the Schrödinger picture:

$$\left(\begin{array}{c}\hat{\boldsymbol{\gamma}}(t)\\\hat{\boldsymbol{\gamma}}^\dagger(t)\end{array}\right) = \mathbb{U}^\dagger(t)\left(\begin{array}{c}\hat{\mathbf{c}}\\\hat{\mathbf{c}}^\dagger\end{array}\right)\,, \tag{205}$$

or more explicitly:

$$\hat{\gamma}_\mu(t) = \sum_{j=1}^{L}\left(\mathbf{U}_{j\mu}^*(t)\,\hat{c}_j + \mathbf{V}_{j\mu}^*(t)\,\hat{c}_j^\dagger\right)\,. \tag{206}$$

If we go back to Sec. 7.3, we realize that the algebra carried out there is perfectly applicable here, and allows us to write the time-dependent state $|\psi(t)\rangle$ in the explicit Gaussian form:

$$|\psi(t)\rangle = \mathcal{N}(t)\,\exp\left(\frac{1}{2}\sum_{j_1 j_2}\mathbf{Z}_{j_1 j_2}(t)\hat{c}_{j_1}^\dagger\hat{c}_{j_2}^\dagger\right)|0\rangle\,, \tag{207}$$

with the anti-symmetric matrix $\mathbf{Z}(t)$ given by:

$$\mathbf{Z}(t) = -\left(\mathbf{U}^\dagger(t)\right)^{-1}\mathbf{V}^\dagger(t)\,. \tag{208}$$

It is not very hard to explicitly verify that such a state satisfies the Schrödinger equation:

$$i\hbar\frac{d}{dt}|\psi(t)\rangle = \widehat{H}(t)|\psi(t)\rangle\,, \tag{209}$$

provided $\mathbf{U}(t)$ and $\mathbf{V}(t)$ satisfy the time-dependent BdG equations in Eq. (200). Indeed, the time derivative of the state $|\psi(t)\rangle$ is simply:

$$i\hbar\frac{d}{dt}|\psi(t)\rangle = i\hbar\left(\frac{1}{2}(\hat{\mathbf{c}}^\dagger)^{\mathrm{T}}\dot{\mathbf{Z}}(t)(\hat{\mathbf{c}}^\dagger) + \frac{\dot{\mathcal{N}}(t)}{\mathcal{N}(t)}\right)|\psi(t)\rangle\,.$$

On the right-hand side, the Hamiltonian terms can be rewritten by using that, for instance:

$$\sum_{jj'}\hat{c}_{j'}^\dagger\mathbf{A}_{j'j}\hat{c}_j\,\mathrm{e}^{\mathcal{Z}(t)}|0\rangle = \sum_{jj'}\hat{c}_{j'}^\dagger(\mathbf{AZ})_{j'j}\hat{c}_j^\dagger\,\mathrm{e}^{\mathcal{Z}(t)}|0\rangle\,.$$

Rewriting all the Hamiltonian terms we get:

$$\widehat{H}(t)|\psi(t)\rangle = \left((\hat{\mathbf{c}}^\dagger)^{\mathrm{T}}\left(\mathbf{B}+\mathbf{AZ}+\mathbf{ZA}+\mathbf{ZB}^*\mathbf{Z}\right)(\hat{\mathbf{c}}^\dagger) - \mathrm{Tr}\mathbf{A} - \mathrm{Tr}\mathbf{B}^*\mathbf{Z}\right)|\psi(t)\rangle\,.$$

By explicitly calculating the derivative of $\mathbf{Z}(t)$ using the BdG equations one can check, after some lengthy algebra, that the two expressions indeed coincide.

---

**Problem 7. Time-dependent BdG equations for a uniform chain.**

Consider the uniform model with anisotropy $\kappa = 1$. Take $J = 1$ (constant in time, and taken as our unit of energy), and consider a time-dependent transverse magnetic field $h(t)$. Show that, in analogy with the form of the ABC ground state in Eq. (70), the time-dependent state

$$|\psi(t)\rangle = \prod_{k>0}^{\mathrm{ABC}}\left(u_k(t) + v_k(t)\hat{c}_k^\dagger\hat{c}_{-k}^\dagger\right)|0\rangle\,,$$

solves the time-dependent Schrödinger equation $i\hbar|\dot{\psi}(t)\rangle = \widehat{\mathbb{H}}_0(t)|\psi(t)\rangle$ in the fermionic ABC sector provided $u_k(t)$ and $v_k(t)$ satisfy, for all $k$, the following BdG equations:

$$i\hbar\dot{\boldsymbol{\psi}}_k(t) = \mathbf{H}_k(t)\boldsymbol{\psi}_k(t)\,,\qquad \text{with}\qquad \boldsymbol{\psi}_k(t) = \begin{pmatrix}v_k(t)\\u_k(t)\end{pmatrix}\,.$$

---

**Problem 8. Out-of-equilibrium protocol: crossing the critical point.**

Consider now a slow annealing of the transverse field $h(t)$ from the initial value $h_{\mathrm{i}} \gg J$ at time $t = 0$, to the final value $h_{\mathrm{f}} = 0$ at time $t = \tau$, for instance linearly: $h(t) = h_{\mathrm{i}}(1-t/\tau)$. Initialize the system in the ground state of $\widehat{H}_k(t = 0)$, and numerically study the BdG evolution for all $k$, for a sufficiently large $L$. Consider now the expectation value of the density of defects over the ferromagnetic ground state at the end of the non-equilibrium protocol

$$\rho_{\mathrm{def}}(\tau) = \frac{1}{2L}\sum_{j=1}^{L}\langle\psi(\tau)|(1-\hat{\sigma}_j^x\hat{\sigma}_{j+1}^x)|\psi(\tau)\rangle\,.$$

Show that $\rho_{\mathrm{def}}(\tau) \sim \tau^{-1/2}$ for sufficiently large $\tau$, provided $L$ is large enough.[37] This is the so-called Kibble-Zurek scaling, see Refs. [35, 90] for details and further references on this topic.

---

[37]The power-law scaling of $\rho_{\mathrm{def}}(\tau)$ holds for $\tau \ll \tau_L^*$ where $\tau_L^* \propto L^2$ is a characteristic time where the finite-size critical gap, scaling as $1/L$, starts to be visible. For larger $\tau$, the finite-size critical gap at $h_c/J = 1$ becomes relevant, and the density of defects starts decaying faster.

## 8.2 Calculating time-dependent expectation values

Once we have a solution to the time-dependent BdG equations, we can calculate time-dependent expectations of operators quite easily. Consider, for instance, the elementary one-body Green's function:

$$
\begin{aligned}
\mathbf{G}_{jj'}(t) &\equiv \langle\psi(t)|\hat{c}_j\hat{c}^\dagger_{j'}|\psi(t)\rangle = \langle\psi(t_0)|\hat{c}_{j_\mathrm{H}}(t)\hat{c}^\dagger_{j'_\mathrm{H}}(t)|\psi(t_0)\rangle, \\
\mathbf{F}_{jj'}(t) &\equiv \langle\psi(t)|\hat{c}_j\hat{c}_{j'}|\psi(t)\rangle = \langle\psi(t_0)|\hat{c}_{j_\mathrm{H}}(t)\hat{c}_{j'_\mathrm{H}}(t)|\psi(t_0)\rangle.
\end{aligned}
\tag{210}
$$

We assume that the initial state $|\psi(t_0)\rangle$ is the Bogoliubov vacuum of the operators $\gamma$ which diagonalise $\widehat{H}(t_0)$, i.e., $|\psi(t_0)\rangle = |\emptyset_\gamma\rangle$. The algebra is most directly carried out by working with the $2L \times 2L$ Nambu one-body Green's function matrix

$$
\mathbb{G}_{jj'}(t) \equiv \langle\psi(t)|\widehat{\boldsymbol{\Psi}}_j\widehat{\boldsymbol{\Psi}}^\dagger_{j'}|\psi(t)\rangle = \langle\psi(t_0)|\widehat{\boldsymbol{\Psi}}_{j_\mathrm{H}}(t)\widehat{\boldsymbol{\Psi}}^\dagger_{j'_\mathrm{H}}(t)|\psi(t_0)\rangle,
\tag{211}
$$

by using the fact that the corresponding transformed Green's function is simple, since $|\psi(t_0)\rangle = |\emptyset_\gamma\rangle$:

$$
\mathbb{G}^\gamma_{\mu\mu'} \equiv \langle\psi(t_0)|\widehat{\boldsymbol{\Phi}}_\mu\widehat{\boldsymbol{\Phi}}^\dagger_{\mu'}|\psi(t_0)\rangle = \left(\begin{array}{c|c} \mathbf{1} & \mathbf{0} \\ \hline \mathbf{0} & \mathbf{0} \end{array}\right).
\tag{212}
$$

In matrix form, we immediately calculate:

$$
\begin{aligned}
\mathbb{G}(t) &= \langle\psi(t_0)|\widehat{\boldsymbol{\Psi}}_\mathrm{H}(t)\widehat{\boldsymbol{\Psi}}^\dagger_\mathrm{H}(t)|\psi(t_0)\rangle = \mathbb{U}(t)\langle\psi(t_0)|\widehat{\boldsymbol{\Phi}}\ \widehat{\boldsymbol{\Phi}}^\dagger|\psi(t_0)\rangle\,\mathbb{U}^\dagger(t) \\
&= \mathbb{U}(t)\left(\begin{array}{c|c} \mathbf{1} & \mathbf{0} \\ \hline \mathbf{0} & \mathbf{0} \end{array}\right)\mathbb{U}^\dagger(t) = \left(\begin{array}{c|c} \mathbf{U}(t)\mathbf{U}^\dagger(t) & \mathbf{U}(t)\mathbf{V}^\dagger(t) \\ \hline \mathbf{V}(t)\mathbf{U}^\dagger(t) & \mathbf{V}(t)\mathbf{V}^\dagger(t) \end{array}\right).
\end{aligned}
\tag{213}
$$

Summarising, the four $L \times L$ blocks of $\mathbb{G}$ read:

$$
\mathbb{G}(t) = \left(\begin{array}{c|c} \mathbf{G}(t) & \mathbf{F}(t) \\ \hline \mathbf{F}^\dagger(t) & \mathbf{1} - \mathbf{G}^\mathrm{T}(t) \end{array}\right) = \left(\begin{array}{c|c} \mathbf{U}(t)\mathbf{U}^\dagger(t) & \mathbf{U}(t)\mathbf{V}^\dagger(t) \\ \hline \mathbf{V}(t)\mathbf{U}^\dagger(t) & \mathbf{V}(t)\mathbf{V}^\dagger(t) \end{array}\right).
\tag{214}
$$

> ℹ **Info:** Notice that, quite generally, $\mathbf{G}$ is Hermitian, while $\mathbf{F}$, as a consequence of the fermionic anti-commutations is anti-symmetric:
>
> $$
> \mathbf{G}(t) = \mathbf{U}(t)\mathbf{U}^\dagger(t) = \mathbf{G}^\dagger(t), \qquad \text{and} \qquad \mathbf{F}(t) = \mathbf{U}(t)\mathbf{V}^\dagger(t) = -\mathbf{F}^\mathrm{T}(t).
> \tag{215}
> $$

Expectation values of more complicated operators can be reduced to sums of products of Green's functions through the application of Wick's theorem [1]. This fact will be explicitly used later on, for instance, when calculating expectation values for spin-spin correlation functions, see Sec. 9, or the Entanglement entropy, see Sec. 10. Moreover, time-correlation functions with Heisenberg operators at different times can be calculated similarly.

## 8.3 Floquet time-dependent case

A particular case of dynamics is that in which the Hamiltonian is periodic in time, i.e., a period $\tau$ exists such that $\widehat{H}(t + \tau) = \widehat{H}(t)$. The whole field of *Floquet engineering* is recently actively pursuing this strategy to construct interesting phases of matter, sometimes without a counterpart in equilibrium physics. See Refs. [89, 91] and reference therein for more details.

The Floquet theorem [92, 93] guarantees the existence in the Hilbert space of a complete basis of solutions of the time-dependent Schrödinger equation which are *periodic* "up to a phase factor", i.e., such that:

$$
|\psi_{\mathrm{F}\alpha}(t)\rangle = e^{-i\mathcal{E}_\alpha t/\hbar}|\psi_{\mathrm{P}\alpha}(t)\rangle, \quad \text{with} \quad |\psi_{\mathrm{P}\alpha}(t)\rangle = |\psi_{\mathrm{P}\alpha}(t + \tau)\rangle.
\tag{216}
$$

This way of writing is closely reminiscent of the time-independent case, except that the state $|\psi_{P\alpha}(t)\rangle$, known as *Floquet mode*, is now *periodic* in time rather than a time-independent eigenstate of the Hamiltonian; the $\mathcal{E}_\alpha$, which plays the role of the eigenenergy, is known as *Floquet quasi-energy*. There are $2^L$, as many as the dimension of the Hilbert space, Floquet solutions of this type, and these solutions can be used as a convenient time-dependent basis to expand states. Their usefulness consists in the fact that if we expand a general initial state as

$$|\psi(0)\rangle = \sum_\alpha |\psi_{P\alpha}(0)\rangle \langle\psi_{P\alpha}(0)|\psi(0)\rangle,$$

then the time-evolution can be written, for free, in a form that is reminiscent of the time-independent case, i.e.:

$$|\psi(t)\rangle = \underbrace{\sum_{\alpha=1}^{2^L} e^{-i\mathcal{E}_\alpha t/\hbar} |\psi_{P\alpha}(t)\rangle \langle\psi_{P\alpha}(0)|\psi(0)\rangle}_{\widehat{U}(t)}. \tag{217}$$

Explicit construction of the many-body Floquet states can be obtained through a Floquet analysis of the time-dependent Bogoliubov-de Gennes (BdG) equations, in a way similar to that, used to construct the energy eigenstates from the solution of the static BdG equations (see Sec. 7). To do that, let us write the BdG equations (203)

$$i\hbar \frac{d}{dt} \begin{pmatrix} \mathbf{U}(t) \\ \mathbf{V}(t) \end{pmatrix} = 2\,\mathbb{H}(t) \begin{pmatrix} \mathbf{U}(t) \\ \mathbf{V}(t) \end{pmatrix}. \tag{218}$$

Since $\mathbb{H}(t+\tau) = \mathbb{H}(t)$ is a periodic $2L \times 2L$ matrix, the Floquet theorem guarantees the existence of a complete set of $2L$ solutions which are periodic up to a phase. $L$ of them have the form:

$$e^{-i\epsilon_\mu t/\hbar} \begin{pmatrix} \mathbf{u}_{P\mu}(t) \\ \mathbf{v}_{P\mu}(t) \end{pmatrix}, \qquad \text{for } \mu = 1 \cdots L, \qquad \text{with} \qquad \begin{cases} \mathbf{u}_{P\mu}(t+\tau) = \mathbf{u}_{P\mu}(t), \\ \mathbf{v}_{P\mu}(t+\tau) = \mathbf{v}_{P\mu}(t), \end{cases}$$

and the remaining $L$, by particle-hole symmetry, are automatically obtained as

$$e^{i\epsilon_\mu t/\hbar} \begin{pmatrix} \mathbf{v}_{P\mu}^*(t) \\ \mathbf{u}_{P\mu}^*(t) \end{pmatrix}.$$

Collecting all the quasi-energies $\epsilon_\mu$ into a diagonal matrix $\boldsymbol{\epsilon} = \text{diag}(\epsilon_\mu)$, and the various column vectors $\mathbf{u}_{P\mu}(t)$ and $\mathbf{v}_{P\mu}(t)$ into a $L \times L$ matrices $\mathbf{U}_P(t)$ and $\mathbf{V}_P(t)$, it is straightforward to show that the structure of the Floquet solutions of the BdG solutions is[38]

$$\mathbb{U}_F(t) = \left( \begin{array}{c|c} \mathbf{U}_F(t) & \mathbf{V}_F^*(t) \\ \hline \mathbf{V}_F(t) & \mathbf{U}_F^*(t) \end{array} \right) = \left( \begin{array}{c|c} \mathbf{U}_P(t)e^{-i\boldsymbol{\epsilon}t/\hbar} & \mathbf{V}_P^*(t)e^{i\boldsymbol{\epsilon}t/\hbar} \\ \hline \mathbf{V}_P(t)e^{-i\boldsymbol{\epsilon}t/\hbar} & \mathbf{U}_P^*(t)e^{i\boldsymbol{\epsilon}t/\hbar} \end{array} \right). \tag{219}$$

Using these solutions, we can construct the Bogoliubov operators $\hat{\gamma}_{F\mu}(t)$ which annihilate a vacuum Floquet state $|\emptyset_F(t)\rangle$ through the standard method employed in the general time-dependent case (see Eq. 140):

$$\begin{pmatrix} \hat{\boldsymbol{\gamma}}_F(t) \\ \hat{\boldsymbol{\gamma}}_F^\dagger(t) \end{pmatrix} = \mathbb{U}_F^\dagger(t) \begin{pmatrix} \hat{\mathbf{c}} \\ \hat{\mathbf{c}}^\dagger \end{pmatrix}, \tag{220}$$

---

[38]Notice that the quasi-energy phase factors have to stay on the *right* of the periodic part, for the ordinary rules of matrix multiplication to give the correct phase-factor to each *column* of the matrix.

or, more explicitly, for $\mu = 1, \cdots, L$:

$$\hat{\gamma}_{\text{F}\mu}(t) = e^{i\epsilon_\mu t/\hbar} \sum_{j=1}^{L} \left( (\mathbf{U}_{\text{P}}^*(t))_{j\mu} \hat{c}_j + (\mathbf{V}_{\text{P}}^*(t))_{j\mu} \hat{c}_j^\dagger \right) \quad \Rightarrow \quad \hat{\gamma}_{\text{F}\mu}(t + \tau) = e^{i\epsilon_\mu \tau/\hbar} \hat{\gamma}_{\text{F}\mu}(t), \quad \forall\, t. \tag{221}$$

The Floquet vacuum state $|\emptyset_{\text{F}}(t)\rangle$ annihilated by all the $\hat{\gamma}_{\text{F}\mu}(t)$ has the Gaussian form (see Eq. (207)):

$$|\emptyset_{\text{F}}(t)\rangle = \mathcal{N}_{\text{F}}(t) \exp\left( \frac{1}{2} \sum_{j_1 j_2} (\mathbf{Z}_{\text{F}}(t))_{j_1 j_2} \hat{c}_{j_1}^\dagger \hat{c}_{j_2}^\dagger \right) |0\rangle, \tag{222}$$

where, see Secs. 7 and 8, the Thouless and Onishi formulas hold:

$$\mathbf{Z}_{\text{F}}(t) = -(\mathbf{U}_{\text{F}}^\dagger(t))^{-1} \mathbf{V}_{\text{F}}^\dagger(t), \qquad \text{and} \qquad \mathcal{N}_{\text{F}}(t) = \sqrt{|\det(\mathbf{U}_{\text{F}}(t))|}. \tag{223}$$

Let us show that the Floquet vacuum state is periodic, i.e.,

$$|\emptyset_{\text{F}}(t + \tau)\rangle = |\emptyset_{\text{F}}(t)\rangle,$$

or, to put it differently, its many-body quasi-energy is $\mathcal{E}_0 = 0$. To this aim, it suffices to show that $\mathbf{Z}_{\text{F}}(t)$ and $\mathcal{N}_{\text{F}}(t)$ are both periodic. From $\mathbf{V}_{\text{F}} = \mathbf{V}_{\text{P}} e^{-i\epsilon t/\hbar}$ and $\mathbf{U}_{\text{F}} = \mathbf{U}_{\text{P}} e^{-i\epsilon t/\hbar}$ we immediately derive that $\mathbf{V}_{\text{F}}^\dagger(t) = e^{i\epsilon t/\hbar} \mathbf{V}_{\text{P}}^\dagger(t)$ and $(\mathbf{U}_{\text{F}}^\dagger(t))^{-1} = (\mathbf{U}_{\text{P}}^\dagger(t))^{-1} e^{-i\epsilon t/\hbar}$. From these relationships, in turn, it follows immediately that the quasi-energy phase-factors cancel in $\mathbf{Z}_{\text{F}}$, i.e.:

$$\mathbf{Z}_{\text{F}}(t) = -(\mathbf{U}_{\text{F}}^\dagger(t))^{-1} \mathbf{V}_{\text{F}}^\dagger(t) = -(\mathbf{U}_{\text{P}}^\dagger(t))^{-1} \mathbf{V}_{\text{P}}^\dagger(t), \tag{224}$$

which is manifestly periodic in time, $\mathbf{Z}_{\text{F}}(t + \tau) = \mathbf{Z}_{\text{F}}(t)$, because both $\mathbf{U}_{\text{P}}$ and $\mathbf{V}_{\text{P}}$ are periodic. The periodicity of $\mathcal{N}_{\text{F}}(t)$ follows because

$$\left| \det(\mathbf{U}_{\text{F}}(t)) \right| = \left| \det(\mathbf{U}_{\text{P}}(t)) \det(e^{-i\epsilon t/\hbar}) \right| = \left| \det(\mathbf{U}_{\text{P}}(t)) \right| \left| e^{-i\sum_\mu \epsilon_\mu t/\hbar} \right| = \left| \det(\mathbf{U}_{\text{P}}(t)) \right|,$$

i.e., once again something manifestly periodic in time. At this point, we can easily, in principle, construct all the $2^L$ many-body Floquet states by simply applying any product of $\hat{\gamma}_{\text{F}\mu}^\dagger(t)$ to $|\emptyset_{\text{F}}(t)\rangle$:[39]

$$|\psi_{\text{F}\{n_\mu\}}(t)\rangle = \prod_{\mu=1}^{L} \left( \hat{\gamma}_{\text{F}\mu}^\dagger(t) \right)^{n_\mu} |\emptyset_{\text{F}}(t)\rangle, \tag{225}$$

where $n_\mu = 0$ or $1$ is the occupation number of the $\hat{\gamma}_{\text{F}\mu}^\dagger(t)$ operator. From Eq. (221) and the periodicity of the Floquet vacuum, it follows that the quasi-energy of $|\psi_{\text{F}\{n_\mu\}}(t)\rangle$ is given by:

$$\mathcal{E}_{\{n_\mu\}} = \sum_{\mu=1}^{L} n_\mu \epsilon_\mu. \tag{226}$$

## 9 Spin-spin correlation functions

As discussed in Sec. 5, the thermal physics of the classical Ising model in two dimensions is reproduced by the ground state physics of the quantum Ising chain. Here we consider spin-spin correlation functions. From the general theory we know that, see Eq. (110), the ground

---

[39]Some care should be exercised if the boundary conditions depend on the fermionic parity. In that case, one should work separately in the two subsectors with even and odd fermionic parity, starting from the corresponding vacuum state.

state expectation value of the spin-spin (order-parameter) correlations for a translationally invariant system should have the form:[40]

$$C_{j,j+n}^{xx} = \langle \hat{\sigma}_j^x \hat{\sigma}_{j+n}^x \rangle \xrightarrow{n \text{ large}} m^2 + \text{const} \times \frac{e^{-an/\xi}}{n^\eta} \,, \tag{227}$$

where $m$ is the order-parameter $m = \langle \hat{\sigma}_j^x \rangle$, $\xi$ the correlation length, and $\eta$ the anomalous exponent.

The question we address here is how to calculate spin-spin correlation functions for the quantum Ising chain, see Refs. [74, 94] for a more in-depth and general discussion. Specifically, we want to show how to calculate spin-spin correlations for the three possible directions in spin-space:

$$C_{j_1,j_2}^{xx} = \langle \hat{\sigma}_{j_1}^x \hat{\sigma}_{j_2}^x \rangle, \qquad C_{j_1,j_2}^{yy} = \langle \hat{\sigma}_{j_1}^y \hat{\sigma}_{j_2}^y \rangle, \qquad C_{j_1,j_2}^{zz} = \langle \hat{\sigma}_{j_1}^z \hat{\sigma}_{j_2}^z \rangle. \tag{228}$$

Here we discuss in detail the equilibrium case only.

The $C_{j_1,j_2}^{zz}$ does not involve the Jordan-Wigner non-local string, and comes automatically from using Wick's theorem, once you have calculated the single-particle Green's functions, including the anomalous term, see Sec. 4.1.2 and Sec. 8.2.

---

**Problem 9.  The zz correlations.**

By a direct application of the Jordan-Wigner mapping $\hat{\sigma}_j^z = 1 - \hat{c}_j^\dagger \hat{c}_j$ and of Wick's theorem, show that:

$$C_{j_1,j_2}^{zz} = 4\Big(G_{j_1,j_1} - \frac{1}{2}\Big)\Big(G_{j_2,j_2} - \frac{1}{2}\Big) + 4G_{j_1,j_2}(\delta_{j_1,j_2} - G_{j_2,j_1}) + 4|F_{j_1,j_2}|^2 \,. \tag{229}$$

Verify that $C_{j_1,j_1}^{zz} = 1$. Show that:

$$\langle \hat{\sigma}_j^z \rangle = 2\Big(G_{j,j} - \frac{1}{2}\Big) \,. \tag{230}$$

Recognise that the first term in $C_{j_1,j_2}^{zz}$ is simply $\langle \hat{\sigma}_{j_1}^z \rangle \langle \hat{\sigma}_{j_2}^z \rangle$. As a consequence, the so-called *connected* correlation function is given by:

$$C_{j_1,j_2}^{zz,\text{conn}} \stackrel{\text{def}}{=} C_{j_1,j_2}^{zz} - \langle \hat{\sigma}_{j_1}^z \rangle \langle \hat{\sigma}_{j_2}^z \rangle = 4G_{j_1,j_2}(\delta_{j_1,j_2} - G_{j_2,j_1}) + 4|F_{j_1,j_2}|^2 \,. \tag{231}$$

For the translationally invariant critical Ising case ($\kappa = 1$, $h = J$), using the Green's functions derived in Sec. 4.1.2, verify that:

$$C_{1,1+n}^{zz,\text{conn}} = \frac{4}{\pi^2} \frac{1}{4n^2 - 1} \,. \tag{232}$$

Compare this with Eq. (3.3) of Ref. [65].

---

Consider now the calculation of the $\hat{\sigma}_{j_1}^x \hat{\sigma}_{j_2}^x$ correlations[41] for $j_2 > j_1$. Using the Jordan-Wigner mapping we get:

$$C_{j_1,j_2}^{xx} = \langle \hat{\sigma}_{j_1}^x \hat{\sigma}_{j_2}^x \rangle = \langle (\hat{c}_{j_1}^\dagger + \hat{c}_{j_1}) \exp\Big(i\pi \sum_{j=j_1}^{j_2-1} \hat{n}_j\Big)(\hat{c}_{j_2}^\dagger + \hat{c}_{j_2}) \rangle \,. \tag{233}$$

---

[40]Recall that the rotation in spin-space we have performed is such that the longitudinal direction is given by $\hat{\sigma}^x$.

[41]Recall that, in our convention, $x$ is the spin direction in which the symmetry-breaking occurs.

Recall now that:

$$e^{i\pi\hat{n}_j} = 1 - 2\hat{n}_j = \hat{c}_j\hat{c}_j^\dagger - \hat{c}_j^\dagger\hat{c}_j = (\hat{c}_j^\dagger + \hat{c}_j)(\hat{c}_j^\dagger - \hat{c}_j) = \widehat{A}_j\widehat{B}_j, \tag{234}$$

where the last expression involves the definitions [75]

$$\widehat{A}_j = (\hat{c}_j^\dagger + \hat{c}_j) \equiv \check{c}_{j,1}, \qquad \text{and} \qquad \widehat{B}_j = (\hat{c}_j^\dagger - \hat{c}_j) = -i\check{c}_{j,2}, \tag{235}$$

closely related to the Majorana fermions. Hence we can write:

$$\begin{aligned}
C_{j_1,j_2}^{xx} = \langle \hat{\sigma}_{j_1}^x \hat{\sigma}_{j_2}^x \rangle &= \langle \widehat{A}_{j_1} \Big(\prod_{j=j_1}^{j_2-1} \widehat{A}_j\widehat{B}_j\Big)\widehat{A}_{j_2} \rangle \\
&= \langle \widehat{B}_{j_1}\widehat{A}_{j_1+1}\widehat{B}_{j_1+1} \cdots \widehat{A}_{j_2-1}\widehat{B}_{j_2-1}\widehat{A}_{j_2} \rangle, \tag{236}
\end{aligned}$$

where we used that $(\widehat{A}_j)^2 = 1$.

At this point, we use Wick's theorem [1], since this is a product of fermion operators averaged on the ground state of a quadratic Hamiltonian (a Gaussian state, as we have seen in Sec. 7.3), or on a thermal state of a quadratic Hamiltonian, depending on whether we are calculating correlations at $T = 0$ or finite $T$. There are $2(j_2 - j_1)$ elements in the product, which we should be contracted pairwise in all possible ways, with the appropriate permutation sign [1].

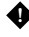

**Recall:** The results of Sec. 8.2 concerning the elementary fermionic Green's functions tell us that:

$$\mathbf{G}_{jj'} \overset{\text{def}}{=} \langle \hat{c}_j \hat{c}_{j'}^\dagger \rangle = \left(\mathbf{U}\mathbf{U}^\dagger\right)_{jj'}, \tag{237}$$

and

$$\mathbf{F}_{jj'} \overset{\text{def}}{=} \langle \hat{c}_j \hat{c}_{j'} \rangle = \left(\mathbf{U}\mathbf{V}^\dagger\right)_{jj'}. \tag{238}$$

Recall also that the anti-commutation of the fermionic destruction operators forces $\mathbf{F}$ to be anti-symmetric. Moreover, if the average is taken over the ground state so that $\mathbf{U}$ and $\mathbf{V}$ can be taken to be both real,[42] then $\mathbf{G} = \mathbf{U}\mathbf{U}^\mathsf{T}$ is *real and symmetric*, and $\mathbf{F} = \mathbf{U}\mathbf{V}^\mathsf{T}$ is *real and anti-symmetric*.

Let us start by observing that

$$\overset{\lceil\quad\rceil}{\widehat{A}_j\widehat{A}_{j'}} = \langle \widehat{A}_j\widehat{A}_{j'} \rangle = \langle(\hat{c}_j^\dagger + \hat{c}_j)(\hat{c}_{j'}^\dagger + \hat{c}_{j'})\rangle = \mathbf{G}_{jj'} + (\delta_{j,j'} - \mathbf{G}_{j'j}) + \mathbf{F}_{jj'} + \mathbf{F}_{j'j}^* = \delta_{j,j'}, \tag{239}$$

where we used that $\mathbf{G} = \mathbf{G}^\mathsf{T}$ and $(\mathbf{F}^*)^\mathsf{T} = -\mathbf{F}$. Similarly, we have that:

$$\overset{\lceil\quad\rceil}{\widehat{B}_j\widehat{B}_{j'}} = \langle \widehat{B}_j\widehat{B}_{j'} \rangle = \langle(\hat{c}_j^\dagger - \hat{c}_j)(\hat{c}_{j'}^\dagger - \hat{c}_{j'})\rangle = -\mathbf{G}_{jj'} - (\delta_{j,j'} - \mathbf{G}_{j'j}) + \mathbf{F}_{jj'} + \mathbf{F}_{j'j}^* = -\delta_{j,j'}. \tag{240}$$

These findings simply eliminate contractions between operators of the same type.[43]

We are therefore left with the contractions of the type:

$$\begin{aligned}
\overset{\lceil\quad\rceil}{\widehat{B}_j\widehat{A}_{j'}} = \langle \widehat{B}_j\widehat{A}_{j'} \rangle = \langle(\hat{c}_j^\dagger - \hat{c}_j)(\hat{c}_{j'}^\dagger + \hat{c}_{j'})\rangle &= -\mathbf{G}_{jj'} + (\delta_{j,j'} - \mathbf{G}_{j'j}) - \mathbf{F}_{jj'} + \mathbf{F}_{j'j}^* \\
&= \delta_{j,j'} - 2(\mathbf{G}_{jj'} + \mathbf{F}_{jj'}) \overset{\text{def}}{=} \mathbf{M}_{j,j'}, \tag{241}
\end{aligned}$$

---

[42]The same results can be easily obtained from thermal averages, see Sec. 11.

[43]Observe that contractions of the type $\langle \widehat{A}_j\widehat{A}_j \rangle$ or $\langle \widehat{B}_j\widehat{B}_j \rangle$ never occur from the Wick's expansion.

and

$$\overset{\frown}{\widehat{A}_j \widehat{B}_{j'}} = \langle \widehat{A}_j \widehat{B}_{j'} \rangle = \langle (\hat{c}_j^\dagger + \hat{c}_j)(\hat{c}_{j'}^\dagger - \hat{c}_{j'}) \rangle = \mathbf{G}_{jj'} - (\delta_{j,j'} - \mathbf{G}_{j'j}) - \mathbf{F}_{jj'} + \mathbf{F}_{j'j}^*$$
$$= 2(\mathbf{G}_{j'j} + \mathbf{F}_{j'j}) - \delta_{j,j'} = -\mathbf{M}_{j',j}, \qquad (242)$$

as perhaps expected.

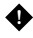 **Warning:** The previous considerations apply to expectations calculated on the ground state or to thermal expectations. When considering *time-dependent* expectations the condition of reality of the matrices $\mathbf{G}$ and $\mathbf{F}$ no longer applies, and appropriate modifications would be needed, see Ref. [94].

We now have to account for all possible contractions of the B-A type, say, with the proper permutation sign. If you think for a while, you realize that you can organize the whole of Wick's sum into the determinant of an appropriate matrix as follows:

$$\text{Wick} \rightarrow \overset{\frown}{\widehat{B}_{j_1}\widehat{A}_{j_1+1}}\overset{\frown}{\widehat{B}_{j_1+1}\widehat{A}_{j_1+2}}\cdots\overset{\frown}{\widehat{B}_{j_2-1}\widehat{A}_{j_2}} + \overset{\frown}{\widehat{B}_{j_1}}\overset{\frown}{\widehat{A}_{j_1+1}\widehat{B}_{j_1+1}}\overset{\frown}{\widehat{A}_{j_1+2}}\cdots\overset{\frown}{\widehat{B}_{j_2-1}\widehat{A}_{j_2}} + \cdots$$

$$= \det \begin{pmatrix} \mathbf{M}_{j_1,j_1+1} & \mathbf{M}_{j_1,j_1+2} & \cdots & \mathbf{M}_{j_1,j_2} \\ \mathbf{M}_{j_1+1,j_1+1} & \mathbf{M}_{j_1+1,j_1+2} & \cdots & \mathbf{M}_{j_1+1,j_2} \\ \vdots & \vdots & \ddots & \vdots \\ \mathbf{M}_{j_2-1,j_1+1} & \mathbf{M}_{j_2-1,j_1+2} & \cdots & \mathbf{M}_{j_2-1,j_2} \end{pmatrix}_{(j_2-j_1)\times(j_2-j_1)}. \qquad (243)$$

Here, to help the reader recognize the various contractions, we have used colours.

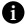 **Info:** A good way to understand the structure of the matrix determinant you see is to notice that the second (column) index is constant — going from $j_1 + 1$ to $j_2$ — and tells you which is the $\widehat{A}$ operator in the contraction: the corresponding first (row) index tells you the $\widehat{B}$ partner in the contraction, and as you see it grows from $j_1$ up to $j_2 - 1$, as appropriate for the $\widehat{B}$ partners.

Summarizing, we can write:

$$C_{j_1,j_2}^{xx} = \det \begin{pmatrix} \mathbf{M}_{j_1,j_1+1} & \mathbf{M}_{j_1,j_1+2} & \cdots & \mathbf{M}_{j_1,j_2-1} & \mathbf{M}_{j_1,j_2} \\ \mathbf{M}_{j_1+1,j_1+1} & \mathbf{M}_{j_1+1,j_1+2} & \mathbf{M}_{j_1+1,j_1+3} & \cdots & \mathbf{M}_{j_1+1,j_2} \\ \vdots & \vdots & \ddots & \vdots & \vdots \\ \mathbf{M}_{j_2-2,j_1+1} & \mathbf{M}_{j_2-2,j_1+2} & \cdots & \mathbf{M}_{j_2-2,j_2-1} & \mathbf{M}_{j_2-2,j_2} \\ \mathbf{M}_{j_2-1,j_1+1} & \mathbf{M}_{j_2-1,j_1+2} & \cdots & \mathbf{M}_{j_2-1,j_2-1} & \mathbf{M}_{j_2-1,j_2} \end{pmatrix}. \qquad (244)$$

---

**Problem 10. The yy correlations.**

With similar arguments, using the Jordan-Wigner mapping of $\hat{\sigma}_j^y$, show that:

$$C_{j_1,j_2}^{yy} = \langle \hat{\sigma}_{j_1}^y \hat{\sigma}_{j_2}^y \rangle = (-1)^{j_2-j_1-1} \langle \widehat{B}_{j_1+1}\widehat{A}_{j_1+1} \cdots \widehat{B}_{j_2-1}\widehat{A}_{j_2-1}\widehat{B}_{j_2}\widehat{A}_{j_1} \rangle. \tag{245}$$

Use Wick's theorem to deduce that:

$$C_{j_1,j_2}^{yy} = \det \begin{pmatrix} \mathbf{M}_{j_1+1,j_1} & \mathbf{M}_{j_1+1,j_1+1} & \cdots & \mathbf{M}_{j_1+1,j_2-2} & \mathbf{M}_{j_1+1,j_2-1} \\ \mathbf{M}_{j_1+2,j_1} & \mathbf{M}_{j_1+2,j_1+1} & \cdots & \mathbf{M}_{j_1+2,j_2-2} & \mathbf{M}_{j_1+2,j_2-1} \\ \vdots & \vdots & \ddots & \vdots & \vdots \\ \mathbf{M}_{j_2-1,j_1} & \mathbf{M}_{j_2-1,j_1+1} & \cdots & \mathbf{M}_{j_2-1,j_2-2} & \mathbf{M}_{j_2-1,j_2-1} \\ \mathbf{M}_{j_2,j_1} & \mathbf{M}_{j_2,j_1+1} & \cdots & \mathbf{M}_{j_2,j_2-2} & \mathbf{M}_{j_2,j_2-1} \end{pmatrix}. \tag{246}$$

---

The translationally invariant case is particularly noteworthy, since the various $\mathbf{M}_{j,j'}$ depend only on the difference of sites. Denoting by $\mathbf{M}_{j-j'} = \mathbf{M}_{j,j'}$, setting $j_1 = 1$ and $j_2 = 1 + n$, we can then write:

$$C_n^{xx} \stackrel{\text{def}}{=} C_{1,1+n}^{xx} = \det \begin{pmatrix} \mathbf{M}_{-1} & \mathbf{M}_{-2} & \cdots & \mathbf{M}_{-(n-1)} & \mathbf{M}_{-n} \\ \mathbf{M}_0 & \mathbf{M}_{-1} & \cdots & \mathbf{M}_{-(n-2)} & \mathbf{M}_{-(n-1)} \\ \vdots & \ddots & \ddots & \ddots & \vdots \\ \mathbf{M}_{n-3} & \mathbf{M}_{n-4} & \cdots & \mathbf{M}_{-1} & \mathbf{M}_{-2} \\ \mathbf{M}_{n-2} & \mathbf{M}_{n-3} & \cdots & \mathbf{M}_0 & \mathbf{M}_{-1} \end{pmatrix}, \tag{247}$$

and

$$C_n^{yy} \stackrel{\text{def}}{=} C_{1,1+n}^{yy} = \det \begin{pmatrix} \mathbf{M}_1 & \mathbf{M}_0 & \cdots & \mathbf{M}_{-(n-3)} & \mathbf{M}_{-(n-2)} \\ \mathbf{M}_2 & \mathbf{M}_1 & \mathbf{M}_0 & \cdots & \mathbf{M}_{-(n-3)} \\ \vdots & \ddots & \ddots & \ddots & \vdots \\ \mathbf{M}_{n-1} & \mathbf{M}_{n-2} & \cdots & \mathbf{M}_1 & \mathbf{M}_0 \\ \mathbf{M}_n & \mathbf{M}_{n-1} & \cdots & \mathbf{M}_2 & \mathbf{M}_1 \end{pmatrix}, \tag{248}$$

both having the form of an $n \times n$ Toeplitz matrix[44] determinant.

---

[44]By definition, a matrix in which every sub-diagonal is constant.

---

**Problem 11. The translationally invariant case.**

By referring to the results for the one-particle Green's functions, show that the critical Ising case ($h = J$ and $\kappa = 1$), and in the thermodynamic limit $L \to \infty$, the elementary contractions are given by:

$$\mathbf{M}_{j,j'} = \mathbf{M}_{j-j'} = \frac{1}{\pi\sqrt{2}} \int_0^\pi dk \, \frac{\cos(k(j-j'+1)) - \cos(k(j-j'))}{\sqrt{1-\cos k}} \,. \tag{249}$$

Evaluate the integral, showing that:

$$\mathbf{M}_{j-j'} = -\frac{2}{\pi} \frac{1}{1+2(j-j')} \,.$$

---

**Problem 12. Spin-spin correlation functions.**

Consider a uniform Ising chain with PBC. By working in the fermionic ABC sector, calculate numerically the spin-spin correlation function $C^{xx}_{1,1+n}$ for the three cases: A) $h/J = 1/2$, b) $h/J = 2$ and c) $h/J = 1$. Verify that the correlations $C^{xx}_{1,1+n}$ tend to decrease with increasing $n$, until $n \sim L/2$, and then increase back towards a value $C^{xx}_{1,L} \equiv C^{xx}_{1,2}$. Explain why this happens. Verify numerically that, when $L \to \infty$ — practically, try increasing $L$ in your calculation — $C^{xx}_{1,L/2}$ tends towards a *finite value* for case a), it goes to zero exponentially fast in case b), and as a power law in case c). Estimate the exponent of such a power law. In the critical case c), compare the results obtained for a finite chain length $L$, and those obtained by using the contractions $\mathbf{M}_{j-j'}$ evaluated in the thermodynamic limit $L \to \infty$ (see previous problem). Repeat the calculations for $C^{yy}_{1,1+n}$. Compare with the analytical solutions provided in Ref. [65].

---

Thanks to the existence of approximate formulae for the asymptotic behavior of Toeplitz matrix determinants [95–99], these exact formulas are the starting point of many large-distance results [29, 65, 74, 94, 100], also related to the 2d classical Ising model [101–103].

## 10 Entanglement entropy

The entanglement entropy is a way to quantify entanglement [104], that plays a fundamental role in many-body physics, both in- and out-of-equilibrium. In the first case, the development of long-range quantum correlations captured by the entanglement entropy is closely related to critical points of second-order quantum-phase transitions [105]. In the case of the Ising model we are discussing here, the situation has been discussed in [106, 107], and it was found that the half-chain entanglement entropy scales logarithmically with the system size at the critical point, in deep connection with conformal field theories [107–109].

On the non-equilibrium side, the entanglement plays an important role in the dynamics of quenched many-body systems. In the case of ergodic thermalizing systems, local observables thermalize, and this happens because the local density matrix becomes mixed, due to the strong entanglement quantum correlations generated by the dynamics (see [110, 111] for

a review). In the case of integrable Hamiltonians, there is no thermalization, but local observables tend to an asymptotic value described by the so-called generalized Gibbs ensemble (GGE) [111, 112]. In this case, the asymptotic local density matrix is less mixed and the dynamics generates less quantum correlations due to local constraints [113–115].

The entanglement entropy quantifies these quantum correlations and is an invaluable tool to distinguish between thermalizing and GGE cases (see, for instance, [116]). Moreover, in case of disorder, many-body localization can appear, an integrable phase where quantum correlations propagate slowly, and half-chain entanglement entropy increases logarithmically in time, in contrast with the linear increase of clean ergodic or integrable systems [117–121]. The Ising model is integrable, and under a quantum quench its half-chain entanglement entropy linearly increases in time, until it reaches an asymptotic value linear in the system size and smaller than the thermal value, as discussed in [27, 115], also in connection with conformal field theories [108].

Finally, the entanglement entropy has a very important role in witnessing the so-called entanglement transitions. If a quantum system undergoes random measurements from a classical environment, the entanglement can show very different behaviours, depending on whether the effect of the quantum dynamics dominates (strong entanglement) or the effect of the onsite measurements (small entanglement). This is reflected in the behaviour of the half-chain entanglement entropy that scales differently with the system size in each entanglement phase. Also in this context the Ising model has been widely employed [45–55], together with similar quadratic fermionic models [45, 52, 122–131], where one evaluates the entanglement entropy using the methods described here.

So, the question of how to calculate the entanglement entropy for the quantum Ising chain is a very important one, and we address it here using the methods developed in [27, 106]. (Equivalent methods leading even to closed expressions for the ground-state entanglement entropy are described in [132–134]. The approach of Ref. [135] for a general real Gaussian density matrix is also worth mentioning.). To explain what the entanglement entropy is, let us start from the concept of reduced density matrix. For a system described by a pure state $|\psi\rangle$, we can equivalently adopt a density matrix formulation [70], using $\hat{\rho} = |\psi\rangle\langle\psi|$. The *reduced density matrix* is a proper (generally non-pure) density matrix obtained by *tracing out* a part of the system. To be concrete, if our Ising chain has $L$ sites, then we can write:

$$\hat{\rho}_l = \text{Tr}_{\{l+1,\cdots,L\}} \hat{\rho}, \qquad (250)$$

where $\text{Tr}_{\{l+1,\cdots,L\}}$ indicates that we take a partial trace over the sites of the chain not belonging to the connected subsystem with sites $\{1, \cdots, l\}$.

The reduced density matrix $\hat{\rho}_l$ must be a positive Hermitian operator acting in the Hilbert space of the $\{1 \cdots l\}$ spins, whose trace is 1. It has $2^l$ non-negative eigenvalues $w_i$ which sum to 1. The only case in which it is itself a pure state is when only one of the $w_i$ is equal to 1, and all the others vanish. This, in turn, is only possible when the state of the two connected subchains $\{1 \cdots l\}$ and $\{l+1 \cdots L\}$ is a *product state*, i.e., a state *without entanglement*. A good way to capture such entanglement is to calculate the so-called *entanglement entropy*

$$S_l = -\text{Tr}_{\{1,\cdots,l\}} \hat{\rho}_l \log \hat{\rho}_l = -\sum_{i=1}^{2^l} w_i \log w_i, \qquad (251)$$

which vanishes exactly when the state is a product state and is positive otherwise.

What do we know about $\hat{\rho}_l$? Recall now that a basis of Hermitian operators for each spin is given by the three Pauli matrices, supplemented by the identity matrix which we denote by $\hat{\sigma}^0 = \mathbf{1}_2$. Hence, we can certainly write $\hat{\rho}_l$ as:

$$\hat{\rho}_l = \frac{1}{2^l} \sum_{\alpha_1,\dots,\alpha_l} C_{\alpha_1\cdots\alpha_l} \hat{\sigma}_1^{\alpha_1} \cdots \hat{\sigma}_l^{\alpha_l}, \qquad \text{with} \quad \alpha_i = 0, x, y, z. \qquad (252)$$

The first question is how the coefficients $C_{\alpha_1 \cdots \alpha_l}$ in this expansion can be written.

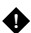

**Info:** Let us recall the analogous problem for the case of a single spin. If we write $\hat{\rho} = \frac{1}{2}\sum_\alpha C_\alpha \hat{\sigma}^\alpha$, then while $C_0 = 1$ due to the unit trace of $\hat{\rho}$, we can reconstruct the other coefficients $C_{\alpha=x,y,z}$ from *measurements* of the different components of the spin, more precisely, from the spin expectation values

$$\langle \hat{\sigma}^\alpha \rangle = \mathrm{Tr}\left(\hat{\sigma}^\alpha \hat{\rho}\right) = \frac{1}{2}\sum_{\alpha'} C_{\alpha'} \mathrm{Tr}\left(\hat{\sigma}^\alpha \hat{\sigma}^{\alpha'}\right) = C_\alpha \,, \tag{253}$$

where we used that $\mathrm{Tr}(\hat{\sigma}^\alpha \hat{\sigma}^{\alpha'}) = 2\delta_{\alpha,\alpha'}$. The name *tomography* is often used in this context: You reconstruct full information on the state by appropriate measurements.

In a very similar way, consider measuring the expectation value of the operator $\hat{\sigma}_1^{\alpha_1} \cdots \hat{\sigma}_l^{\alpha_l}$ on the state $\hat{\rho}_l$. We get:

$$\begin{aligned}
C_{\alpha_1 \cdots \alpha_l} &= \mathrm{Tr}_{\{1 \cdots l\}}\left(\hat{\sigma}_1^{\alpha_1} \cdots \hat{\sigma}_l^{\alpha_l} \hat{\rho}_l\right) \\
&= \mathrm{Tr}_{\{1 \cdots l\}} \mathrm{Tr}_{\{l+1 \cdots L\}}\left(\hat{\sigma}_1^{\alpha_1} \cdots \hat{\sigma}_l^{\alpha_l} \hat{\rho}\right) \\
&= \mathrm{Tr}_{\{1 \cdots L\}}\left(\hat{\sigma}_1^{\alpha_1} \cdots \hat{\sigma}_l^{\alpha_l} \hat{\rho}\right) \equiv \langle \psi | \hat{\sigma}_1^{\alpha_1} \cdots \hat{\sigma}_l^{\alpha_l} | \psi \rangle \,, \tag{254}
\end{aligned}$$

where the second step follows from Eq. (250).

**Remark:** The $4^l$ complex coefficients $C_{\alpha_1 \cdots \alpha_l}$ completely specify the reduced density matrix $\hat{\rho}_l$, an operator in a $2^l$-dimensional space. How to get the eigenvalues of $\hat{\rho}_l$ and extract the entanglement entropy $S_l$, seems a highly non-trivial task, at this stage. Notice also that the previous considerations apply to *any* spin system. There is nothing special, so far, about the quantum Ising chain.

Let us look more closely at an example of such a term. Consider, for a block of $l = 8$ sites the term $C_{000xz00y}$:

$$C_{000xz00y} = \langle \psi | \hat{\sigma}_4^x \hat{\sigma}_5^z \hat{\sigma}_8^y | \psi \rangle \,. \tag{255}$$

First of all observe that such a term respects the parity invariance of the Hamiltonian, as it possesses an *even* number of $\hat{\sigma}^x$ and $\hat{\sigma}^y$ operators: If there was an odd number of them, the coefficient would vanish due to the parity selection rule. Next, we map spins into fermions.

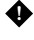 **Info:** To map spins into fermions, recall the Jordan-Wigner transformation:

$$\begin{cases} \hat{\sigma}_j^x = \hat{K}_j \, (\hat{c}_j^\dagger + \hat{c}_j), \\ \hat{\sigma}_j^y = \hat{K}_j \, i(\hat{c}_j^\dagger - \hat{c}_j), \\ \hat{\sigma}_j^z = 1 - 2\hat{n}_j, \end{cases} \qquad \text{with} \qquad \hat{K}_j = \prod_{j'=1}^{j-1}(1 - 2\hat{n}_{j'}). \tag{256}$$

Recall also the following useful ways of writing some of the previous fermionic quantities. First of all, Majorana (Hermitian) combinations appear explicitly in $\hat{\sigma}^x$ and $\hat{\sigma}^y$:

$$\check{c}_{j,1} = (\hat{c}_j^\dagger + \hat{c}_j), \qquad \text{and} \qquad \check{c}_{j,2} = i(\hat{c}_j^\dagger - \hat{c}_j). \tag{257}$$

Second, you can re-express in many ways the operator $1 - 2\hat{n}_j$:

$$1 - 2\hat{n}_j = \hat{c}_j \hat{c}_j^\dagger - \hat{c}_j^\dagger \hat{c}_j = (\hat{c}_j^\dagger + \hat{c}_j)(\hat{c}_j^\dagger - \hat{c}_j) = -i\check{c}_{j,1}\check{c}_{j,2}. \tag{258}$$

Therefore, we get:[45]

$$\begin{aligned} C_{000xz00y} &= \langle\psi|\hat{K}_4\,(\hat{c}_4^\dagger + \hat{c}_4)(1 - 2\hat{n}_5)\hat{K}_8\,i(\hat{c}_8^\dagger - \hat{c}_8)|\psi\rangle \\ &= \langle\psi|(\hat{c}_4^\dagger + \hat{c}_4)(1 - 2\hat{n}_4)(1 - 2\hat{n}_6)(1 - 2\hat{n}_7)\,i(\hat{c}_8^\dagger - \hat{c}_8)|\psi\rangle \\ &= \langle\psi|\check{c}_{4,1}(-i\check{c}_{4,1}\check{c}_{4,2})(-i\check{c}_{6,1}\check{c}_{6,2})(-i\check{c}_{7,1}\check{c}_{7,2})\check{c}_{8,2}|\psi\rangle \\ &= \langle\psi|(-i\check{c}_{4,2})(-i\check{c}_{6,1}\check{c}_{6,2})(-i\check{c}_{7,1}\check{c}_{7,2})\check{c}_{8,2}|\psi\rangle. \end{aligned} \tag{259}$$

⚠ **Warning:** In the notation of Sec. 9, see Eq. (235), such an expectation value would translate into:

$$C_{000xz00y} = i\langle\psi|\widehat{B}_4\widehat{A}_6\widehat{B}_6\widehat{A}_7\widehat{B}_7\widehat{B}_8|\psi\rangle, \tag{260}$$

hence *in equilibrium* (i.e., for a ground state or thermal calculation) it would still vanish, simply because you cannot construct the correct number of non-vanishing Wick's contractions! In any case, you notice how the approach we have undertaken is essentially impossible to carry out. Even after calculating, through an appropriate application of Wick's theorem, all possible non-vanishing coefficients $C_{\alpha_1\cdots\alpha_l}$ calculating the eigenvalues of the corresponding reduced density matrix seems an incredibly difficult task!

But there is something very special about a quantum Ising chain. If $|\psi\rangle$ is the ground state of the quantum Ising chain, the state has a Gaussian form, as explained in Sec. 7.3. Similarly, for a state $|\psi(t)\rangle = \widehat{U}(t,0)|\psi(0)\rangle$, corresponding to a Schrödinger time evolution with an arbitrary quantum Ising chain Hamiltonian, starting from some $|\psi(0)\rangle$ with a Gaussian form. In all these cases, Wick's theorem comes to rescue us in the calculation of the relevant expectation values, which can be expressed as a sum of products of elementary one-particle Green's functions [1]. It turns out that working with Majorana fermions is a good way of handling efficiently ordinary and anomalous fermionic Green's functions. To do that, let us be equipped with a matrix notation for the Majorana as well.

---

[45]Use the fact that $(1-2\hat{n}_j)$ commutes with terms which do not involve fermions at site $j$, and that $(1-2\hat{n}_j)^2 = 1$. This leads, in particular, to a cancellation of the two tails originating from the Jordan-Wigner string operators $\hat{K}_4$ and $\hat{K}_8$. Moreover, recall that the square of a Majorana gives the identity: $(\check{c}_{j,1})^2 = (\check{c}_{j,2})^2 = 1$.

To be consistent with the Nambu notation for the ordinary fermions, we better define the Majorana column vector:[46]

$$
\check{\mathbf{c}} = \begin{pmatrix} \check{c}_{1,1} \\ \check{c}_{2,1} \\ \vdots \\ \check{c}_{L,1} \\ \check{c}_{1,2} \\ \check{c}_{2,2} \\ \vdots \\ \check{c}_{L,2} \end{pmatrix} = \left( \begin{array}{cccc|cccc} 1 & 0 & \cdots & 0 & 1 & 0 & \cdots & 0 \\ 0 & 1 & \cdots & 0 & 0 & 1 & \cdots & 0 \\ \vdots & \vdots & \ddots & \vdots & \vdots & \vdots & \ddots & \vdots \\ 0 & 0 & \cdots & 1 & 0 & 0 & \cdots & 1 \\ \hline -i & 0 & \cdots & 0 & i & 0 & \cdots & 0 \\ 0 & -i & \cdots & 0 & 0 & i & \cdots & 0 \\ \vdots & \vdots & \ddots & \vdots & \vdots & \vdots & \ddots & \vdots \\ 0 & 0 & \cdots & -i & 0 & 0 & \cdots & i \end{array} \right) \begin{pmatrix} \hat{c}_1 \\ \hat{c}_2 \\ \vdots \\ \hat{c}_L \\ \hat{c}_1^\dagger \\ \hat{c}_2^\dagger \\ \vdots \\ \hat{c}_L^\dagger \end{pmatrix} = \mathbb{W}\,\widehat{\boldsymbol{\Psi}}\,, \tag{262}
$$

where we defined the $2L \times 2L$ block matrix:

$$
\mathbb{W} = \left( \begin{array}{c|c} \mathbf{1} & \mathbf{1} \\ \hline -i\mathbf{1} & i\mathbf{1} \end{array} \right)\,. \tag{263}
$$

One can define the Majorana $2L \times 2L$ correlations matrix as:

$$
\mathbb{M}_{nn'} = \langle\psi|\check{\mathbf{c}}_n\check{\mathbf{c}}_{n'}|\psi\rangle = \sum_{j,j'}(\mathbb{W})_{nj}\langle\psi|\widehat{\boldsymbol{\Psi}}_j\widehat{\boldsymbol{\Psi}}_{j'}^\dagger|\psi\rangle(\mathbb{W}^\dagger)_{j'n'}\,, \tag{264}
$$

which in full matrix form is immediately related to the Nambu Green's function $\mathbb{G}$:

$$
\mathbb{M} = \mathbb{W}\,\mathbb{G}\,\mathbb{W}^\dagger\,. \tag{265}
$$

Upon substituting the block-expression for the Nambu Green's function $\mathbb{G}$ in Eq. (214) we obtain, after simple block-matrix algebra:

$$
\mathbb{M} = \left( \begin{array}{c|c} \mathbf{1} + (\mathbf{G} - \mathbf{G}^{\mathrm{T}}) + (\mathbf{F} - \mathbf{F}^*) & i(\mathbf{G} + \mathbf{G}^{\mathrm{T}} - \mathbf{1}) - i(\mathbf{F} + \mathbf{F}^*) \\ \hline -i(\mathbf{G} + \mathbf{G}^{\mathrm{T}} - \mathbf{1}) - i(\mathbf{F} + \mathbf{F}^*) & \mathbf{1} + (\mathbf{G} - \mathbf{G}^{\mathrm{T}}) - (\mathbf{F} - \mathbf{F}^*) \end{array} \right) = \mathbb{1} + i\mathbb{A}\,, \tag{266}
$$

where the $2L \times 2L$ matrix $\mathbb{A}$ is *real* and *anti-symmetric*,[47] and both $\mathbf{G}$ and $\mathbf{F}$ are (in general) complex.

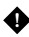 **Remark:** Notice that, quite generally, $\mathbf{F}^{\mathrm{T}} = -\mathbf{F}$ as a consequence of the fermionic anti-commutation. If we are in equilibrium (ground state or thermal) then both $\mathbf{G}$ and $\mathbf{F}$ can be taken to be *real*. Moreover, $\mathbf{G}$ is symmetric, $\mathbf{G} = \mathbf{G}^{\mathrm{T}}$. This implies that

$$
\mathbb{M} = \mathbb{1} + i\left( \begin{array}{c|c} \mathbf{0} & -\mathbf{1} + 2\mathbf{G} - 2\mathbf{F} \\ \hline \mathbf{1} - 2\mathbf{G} - 2\mathbf{F} & \mathbf{0} \end{array} \right)\,, \tag{267}
$$

in agreement with the Majorana equilibrium correlators seen in Sec. 9.

---

[46]The standard definition [19] which in row-vector form would read:

$$
\check{\mathbf{c}} = (\check{c}_1, \check{c}_2, \check{c}_3, \check{c}_4, \cdots, \check{c}_{2L-1}, \check{c}_{2L}) \equiv (\check{c}_{1,1}, \check{c}_{1,2}, \check{c}_{2,1}, \check{c}_{2,2}, \cdots, \check{c}_{L,1}, \check{c}_{L,2})\,, \tag{261}
$$

mixes the different blocks of the Nambu fermions in a way that makes the algebra extremely painful.

[47]The fact that $\mathbb{A}$ is real and anti-symmetric follows from the fact that, quite generally, from $\mathbf{G} = \mathbf{U}\mathbf{U}^\dagger$ and $\mathbf{F} = \mathbf{V}\mathbf{U}^\dagger$ and the unitary nature of the Bogoliubov rotation — see Eq. (8.2) — it follows that $\mathbf{G} + \mathbf{G}^{\mathrm{T}}$ is *real and symmetric*, $\mathbf{F} + \mathbf{F}^*$ is *real and anti-symmetric*, while both $\mathbf{G} - \mathbf{G}^{\mathrm{T}}$ and $\mathbf{F} - \mathbf{F}^*$ are *purely imaginary and anti-symmetric*.

This algebra of re-expressing the Green's functions in terms of Majorana correlators, which in turn involve a single $2L \times 2L$ real and anti-symmetric matrix $\mathbb{A}$, will be important in a short while.[48]

**How to calculate the reduced density matrix spectrum.**   We are now ready to proceed, circumventing the difficulty of calculating all the $4^l$ complex coefficients of the reduced density matrix in Eq. (252) and finding its spectrum.

**Step 1**: The Majorana matrix $\mathbb{M}$ fully determine the (pure) state $|\psi\rangle$, because of the Gaussian nature of the latter and Wick's theorem. We can equivalently write

$$\mathbb{M}_{nn'} = \langle \psi | \check{c}_n \check{c}_{n'} | \psi \rangle = \text{Tr}\left( \check{c}_n \check{c}_{n'} \hat{\rho} \right), \tag{268}$$

where $\hat{\rho} = |\psi\rangle\langle\psi|$ is the pure-state density matrix associated with $|\psi\rangle$.

**Step 2**: Consider now *restricting* the Majorana correlation matrix to the sub-chain $\{1 \cdots l\}$. We will denote such a $2l \times 2l$ matrix as $\mathbb{M}_l$. $\mathbb{M}_l$ is made by four $l \times l$ blocks suitably extracted from the full $2L \times 2L$ matrix $\mathbb{M}$ according to the site-indices involved in $\{1 \cdots l\}$. Most importantly, since it is the block-truncation of an $\mathbb{M} = \mathbb{1} + i\mathbb{A}$, with $\mathbb{A} = \mathbb{A}^* = -\mathbb{A}^{\mathrm{T}}$, it will retain the same structure. More precisely, assuming now $n, n' \in \{1, \cdots, l\}$ we have:

$$\begin{cases} (\mathbb{M}_l)_{n,n'} = \delta_{n,n'} + i\mathbb{A}_{n,n'} (\mathbb{M}_l)_{n,l+n'} = i\mathbb{A}_{n,L+n'}, \\ (\mathbb{M}_l)_{l+n,n'} = i\mathbb{A}_{L+n,n'} (\mathbb{M}_l)_{l+n,l+n'} = \delta_{n,n'} + i\mathbb{A}_{L+n,L+n'}. \end{cases} \tag{269}$$

With a slight leap in the notation, we will now denote these 4 blocks as:

$$\mathbb{M}_l = \mathbb{1}_{2l} + i\mathbb{A}_l, \tag{270}$$

where both $\mathbb{M}_l$ and $\mathbb{A}_l$ are taken to be $2l \times 2l$ and $\mathbb{A}_l$ is real and anti-symmetric.

$\mathbb{M}_l$ contains correlations between Majorana fermions living on the *physical sites* of the reduced chain $\{1 \cdots l\}$. All other sites have been effectively eliminated from the discussion to the point that we might consider restricting our Majorana operators to

$$\check{\mathbf{c}} = \begin{pmatrix} \check{c}_{1,1} \\ \check{c}_{2,1} \\ \vdots \\ \check{c}_{l,1} \\ \check{c}_{1,2} \\ \check{c}_{2,2} \\ \vdots \\ \check{c}_{l,2} \end{pmatrix}_{2l} = \left( \begin{array}{cccc|cccc} 1 & 0 & \cdots & 0 & 1 & 0 & \cdots & 0 \\ 0 & 1 & \cdots & 0 & 0 & 1 & \cdots & 0 \\ \vdots & \vdots & \ddots & \vdots & \vdots & \vdots & \ddots & \vdots \\ 0 & 0 & \cdots & 1 & 0 & 0 & \cdots & 1 \\ \hline -i & 0 & \cdots & 0 & i & 0 & \cdots & 0 \\ 0 & -i & \cdots & 0 & 0 & i & \cdots & 0 \\ \vdots & \vdots & \ddots & \vdots & \vdots & \vdots & \ddots & \vdots \\ 0 & 0 & \cdots & -i & 0 & 0 & \cdots & i \end{array} \right)_{2l \times 2l} \begin{pmatrix} \hat{c}_1 \\ \hat{c}_2 \\ \vdots \\ \hat{c}_l \\ \hat{c}_1^\dagger \\ \hat{c}_2^\dagger \\ \vdots \\ \hat{c}_l^\dagger \end{pmatrix}_{2l}. \tag{271}$$

---

[48]Incidentally, although we will not use it, the same transformation might be applied to the Hamiltonian $\widehat{H}$ to rewrite it in terms of Majorana fermions as follows:

$$\widehat{H} = \widehat{\mathbf{\Psi}}^\dagger \, \mathbb{H} \, \widehat{\mathbf{\Psi}} \; = \frac{1}{4} (\check{\mathbf{c}})^{\mathrm{T}} \, \mathbb{W} \, \mathbb{H} \, \mathbb{W}^\dagger \, (\check{\mathbf{c}}),$$

where use used that $\mathbb{W}^{-1} = \frac{1}{2}\mathbb{W}^\dagger$ and, see Eq. (124):

$$\mathbb{W} \, \mathbb{H} \, \mathbb{W}^\dagger = \left( \begin{array}{c|c} (\mathbf{A} - \mathbf{A}^*) + (\mathbf{B} - \mathbf{B}^*) & i(\mathbf{A} + \mathbf{A}^*) - i(\mathbf{B} + \mathbf{B}^*) \\ \hline -i(\mathbf{A} + \mathbf{A}^*) - i(\mathbf{B} + \mathbf{B}^*) & (\mathbf{A} - \mathbf{A}^*) - (\mathbf{B} - \mathbf{B}^*) \end{array} \right) = i\mathbb{A}_H,$$

with a real anti-symmetric $\mathbb{A}_H = \mathbb{A}_H^* = -\mathbb{A}_H^{\mathrm{T}}$, since $\mathbf{A} = \mathbf{A}^\dagger$ and $\mathbf{B}^{\mathrm{T}} = -\mathbf{B}$. The unitary Bogoliubov rotation would now translate into a real orthogonal rotation of Majorana fermions which transforms the real anti-symmetric matrix $\mathbb{A}_H$ into a standard canonical form.

Hence, we can use the *reduced* density matrix in these averages:

$$(\mathbb{M}_l)_{n,n'} = \text{Tr}\left(\check{\mathbf{c}}_n \check{\mathbf{c}}_{n'} \hat{\rho}\right) \equiv \text{Tr}_{\{1\cdots l\}}\left(\check{\mathbf{c}}_n \check{\mathbf{c}}_{n'} \hat{\rho}_l\right) = \delta_{n,n'} + i(\mathbb{A}_l)_{n,n'}. \tag{272}$$

**Step 3**: We can transform the matrix $\mathbb{A}_l$ to a canonical form, by a (real) orthogonal transformation $\mathbb{R}$. The canonical form of a real anti-symmetric matrix is, see Ref. [84], made of $l$ anti-symmetric $2 \times 2$ blocks along the diagonal:

$$\mathbb{A}_l = \mathbb{R} \not{\bowtie} \mathbb{R}^{\text{T}}, \quad \text{with} \quad \not{\bowtie} = \left(\begin{array}{cc|cc|c|cc} 0 & \lambda_1 & 0 & 0 & \cdots & 0 & 0 \\ -\lambda_1 & 0 & 0 & 0 & \cdots & 0 & 0 \\ \hline 0 & 0 & 0 & \lambda_2 & \cdots & 0 & 0 \\ 0 & 0 & -\lambda_2 & 0 & \cdots & 0 & 0 \\ \hline \vdots & \vdots & \vdots & \vdots & \ddots & \vdots & \vdots \\ \hline 0 & 0 & 0 & 0 & \cdots & 0 & \lambda_l \\ 0 & 0 & 0 & 0 & \cdots & -\lambda_l & 0 \end{array}\right)_{2l \times 2l}, \tag{273}$$

with $\lambda_{q=1\cdots l}$ *real*. Using the rotation matrix $\mathbb{R}$, we can define $2l$ new combinations of Majorana fermions:

$$\check{\mathbf{d}}_q = \sum_{n=1}^{2l} \mathbb{R}_{nq} \check{\mathbf{c}}_n, \quad \text{for} \quad q = 1, \cdots, 2l. \tag{274}$$

These new Majorana combinations, which now mix different sites of the sub-chain $\{1\cdots l\}$, have a very simple correlation matrix:

$$\text{Tr}_{\{1\cdots l\}}\left(\check{\mathbf{d}}_q \check{\mathbf{d}}_{q'} \hat{\rho}_l\right) = \delta_{q,q'} + i\not{\bowtie}_{q,q'}. \tag{275}$$

Now we switch back to ordinary fermions, simply because we are much more trained and used to thinking in terms of them. We therefore define:

$$\hat{d}_q = \frac{1}{2}\left(\check{\mathbf{d}}_{2q-1} + i\check{\mathbf{d}}_{2q}\right), \quad \text{for} \quad q = 1, \cdots, l. \tag{276}$$

By construction, given the simple correlations encoded by the matrix $\not{\bowtie}$, these fermions have averages

$$\text{Tr}_{\{1\cdots l\}}\left(\hat{d}_q^\dagger \hat{d}_{q'} \hat{\rho}_l\right) = \delta_{q,q'} \frac{1+\lambda_q}{2} \equiv \delta_{q,q'} P_q, \tag{277}$$

which shows that $\lambda_q \in [-1, 1]$, in order for the average fermionic occupation to be $0 \le P_q \le 1$, and that the different fermions $\hat{d}_q$ are *uncorrelated*.

**Step 4**: We have found $l$ new uncorrelated fermionic operators. Hence, in this rather non-local basis the reduced density matrix *factorizes*, each $2 \times 2$ block having eigenvalues

$$P_q = \frac{1+\lambda_q}{2}, \quad \text{and} \quad 1-P_q = \frac{1-\lambda_q}{2}, \tag{278}$$

and contributing an entropy $S = -P_q \log P_q - (1-P_q)\log(1-P_q)$. Hence, we finally arrive at the entanglement entropy:

$$S_l = -\sum_{q=1}^{l}\left(P_q \log P_q + (1-P_q)\log(1-P_q)\right). \tag{279}$$

The evaluation of the correlation matrix and the entanglement entropy can be implemented numerically with rather standard techniques.

> **Problem 13. Ground state entanglement for a uniform Ising chain.**
>
> Calculate the ground-state entanglement entropy for a uniform Ising chain, with $l = L/2$, in the three cases: A) $h/J = 1/2$, b) $h/J = 2$, and c) $h/J = 1$. Show that the half-chain entanglement entropy tends to a constant, for increasing $L$, in cases a) and b), while it grows logarithmically in the critical case c).[49]

## 11 Thermal averages

Thermal properties of the random transverse-field Ising chain have been studied by P. Young in Ref. [76]. The case of open boundary conditions (OBC) poses indeed no particular difficulties in calculating thermal averages: we are, after all, dealing with a free Fermi gas of Bogoliubov-de Gennes quasiparticles, whose spectrum can be numerically determined. The calculations become more tricky in the ring geometry (PBC) case, where two different fermionic Hamiltonians should be used to determine the spectrum in the two different even and odd fermion parity sub-sectors of the Hilbert space. This constraint on the parity of the total number of fermions makes the calculation more difficult, in a way that is conceptually similar to that of a free Fermi gas in the *canonical ensemble*. Such a complication is dealt with in the present section.

Let us recall a few basic facts about the general structure of the Ising model Hamiltonian. The full Hamiltonian, when PBC are imposed to the spins reads:

$$\widehat{H}_{\text{PBC}} = \left( \begin{array}{c|c} \widehat{\text{H}}_0 & 0 \\ \hline 0 & \widehat{\text{H}}_1 \end{array} \right). \tag{280}$$

The two blocks of even and odd parity can be written as:

$$\widehat{\text{H}}_0 = \widehat{\text{P}}_0 \widehat{\mathbb{H}}_0 \widehat{\text{P}}_0 = \widehat{\mathbb{H}}_0 \widehat{\text{P}}_0 \,, \qquad \text{and} \qquad \widehat{\text{H}}_1 = \widehat{\text{P}}_1 \widehat{\mathbb{H}}_1 \widehat{\text{P}}_1 = \widehat{\mathbb{H}}_1 \widehat{\text{P}}_1 \,, \tag{281}$$

where $\widehat{\mathbb{H}}_{\text{p}=0,1}$ both conserve the fermionic parity, hence they commute with $\widehat{\text{P}}_{0,1}$ and are given by:

$$\widehat{\mathbb{H}}_{\text{p}=0,1} = -\sum_{j=1}^{L} \left( J_j^+ \hat{c}_j^\dagger \hat{c}_{j+1} + J_j^- \hat{c}_j^\dagger \hat{c}_{j+1}^\dagger + \text{H.c.} \right) + \sum_{j=1}^{L} h_j (2\hat{n}_j - 1), \tag{282}$$

with the boundary condition set by the requirement $\hat{c}_{L+1} \equiv (-1)^{\text{p}+1} \hat{c}_1$.

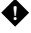

> **Warning:** Neither $\widehat{\mathbb{H}}_{\text{p}=0}$ nor $\widehat{\mathbb{H}}_{\text{p}=1}$, alone, expresses the correct fermionic form of the PBC-spin Hamiltonian. Indeed, after the BdG diagonalization, we can write:
>
> $$\widehat{\mathbb{H}}_{\text{p}} = \sum_{\mu=1}^{L} 2\epsilon_{\text{p},\mu} \left( \hat{\gamma}_{\text{p},\mu}^\dagger \hat{\gamma}_{\text{p},\mu} - \frac{1}{2} \right), \tag{283}$$
>
> hence we have, in general, two different vacuum states $|\emptyset_{\text{p}}\rangle$ such that $\hat{\gamma}_{\text{p},\mu}|\emptyset_{\text{p}}\rangle = 0$, and $2^L$ corresponding Fock states, with different eigenvalues, unless the spins have OBC, in which case $J_L = 0$ and therefore $\widehat{\mathbb{H}}_{\text{p}=1} = \widehat{\mathbb{H}}_{\text{p}=0}$. But we should keep only $2^{L-1}$ even and $2^{L-1}$ odd eigenvalues! How to do correctly the sums involved in the thermal averages is the next issue we are going to consider.

---

[49]The logarithmic divergence in the critical case is consistent with the conformal field theory analysis of Ref. [107, 136]. Away from criticality, where the correlation length $\xi$ is large but finite, the results of Ref. [107] apply.

Consider calculating the thermal average of an operator $\hat{O}$. We should calculate:

$$
\begin{aligned}
\mathrm{Tr}\left(\hat{O}\,e^{-\beta\widehat{H}}\right) &= \sum_{p=0,1}\mathrm{Tr}\left(\widehat{P}_p\hat{O}\,e^{-\beta\widehat{H}}\right) = \sum_{p=0,1}\mathrm{Tr}\left(\widehat{P}_p\hat{O}\,e^{-\beta\widehat{H}}\widehat{P}_p\right) \\
&= \sum_{p,p'}\mathrm{Tr}\left(\widehat{P}_p\hat{O}\widehat{P}_{p'}e^{-\beta\widehat{H}}\widehat{P}_p\right) = \sum_{p}\mathrm{Tr}\left(\widehat{P}_p\hat{O}\widehat{P}_p e^{-\beta\widehat{\mathbb{H}}_p}\widehat{P}_p\right) \\
&= \sum_{p}\mathrm{Tr}\left(\widehat{P}_p\hat{O}\widehat{P}_p e^{-\beta\widehat{\mathbb{H}}_p}\widehat{P}_p\right) = \sum_{p}\mathrm{Tr}\left(\widehat{P}_p\hat{O}\widehat{P}_p e^{-\beta\widehat{\mathbb{H}}_p}\right) \\
&\overset{[\hat{O},\widehat{P}_p]=0}{=} \sum_{p}\mathrm{Tr}\left(\hat{O}\widehat{P}_p e^{-\beta\widehat{\mathbb{H}}_p}\right).
\end{aligned}
\tag{284}
$$

This derivation uses standard properties of projectors and the trace, and, in the final step, the assumption that $\hat{O}$ commutes with the parity. The thing to remark is that now the fermionic Hamiltonians $\widehat{\mathbb{H}}_p$ appear, accompanied by a single projector $\widehat{P}_p$. Next, we recall that

$$
\widehat{P}_p = \frac{1}{2}(\hat{1} + (-1)^p e^{i\pi\widehat{N}}).
\tag{285}
$$

Hence we arrive at:

$$
\mathrm{Tr}\left(\hat{O}\,e^{-\beta\widehat{H}}\right) = \frac{1}{2}\sum_{p=0,1}\left(\mathrm{Tr}\left(\hat{O}\,e^{-\beta\widehat{\mathbb{H}}_p}\right) + (-1)^p\,\mathrm{Tr}\left(\hat{O}\,e^{i\pi\widehat{N}}e^{-\beta\widehat{\mathbb{H}}_p}\right)\right).
\tag{286}
$$

The next thing to consider is how to deal with the term $e^{i\pi\widehat{N}}$, which we can always re-express as:

$$
e^{i\pi\widehat{N}} = \langle\emptyset_p|e^{i\pi\widehat{N}}|\emptyset_p\rangle\,e^{i\pi\sum_{\mu=1}^{L}\hat{\gamma}^{\dagger}_{p,\mu}\hat{\gamma}_{p,\mu}}.
\tag{287}
$$

> **ⓘ Info:** The meaning of such expression should be reasonably transparent. Parity is a good quantum number. Once you determine it on the Bogoliubov vacuum, calculating $\langle\emptyset_p|e^{i\pi\widehat{N}}|\emptyset_p\rangle = \pm 1$, then the parity of each Fock state simply amounts to counting the number of $\hat{\gamma}^{\dagger}$ operators applied to the Bogoliubov vacuum. There is a slight ambiguity in the meaning of $|\emptyset_p\rangle$ that is good to clarify here. We have *not* defined $|\emptyset_p\rangle$ to be the ground state in the sub-sector with parity p — in which case you would directly anticipate that $\langle\emptyset_p|e^{i\pi\widehat{N}}|\emptyset_p\rangle = (-1)^p$, but rather the Bogoliubov vacuum state associated with $\widehat{\mathbb{H}}_p$. So, depending on the couplings and boundary conditions, the parity of $|\emptyset_p\rangle$ might differ from $(-1)^p$. There are cases where, for instance, there is a single $\widehat{\mathbb{H}}$ with a single associated $|\emptyset\rangle$, but also cases where such a single $\widehat{\mathbb{H}}$ can have two degenerate vacuum states $|\emptyset_p\rangle$, as discussed in Sec. 7.2.

The explicit evaluation of $\langle\emptyset_p|e^{i\pi\widehat{N}}|\emptyset_p\rangle$ can be carried out with the techniques explained in Sec. 9. With the same notation used there, you can show that:

$$
\begin{aligned}
\langle\emptyset_p|e^{i\pi\widehat{N}}|\emptyset_p\rangle &= \langle\emptyset_p|\widehat{A}_1\widehat{B}_1\widehat{A}_2\widehat{B}_2\cdots\widehat{A}_L\widehat{B}_L|\emptyset_p\rangle \\
&= (-1)^L\langle\emptyset_p|\widehat{B}_1\widehat{A}_1\widehat{B}_2\widehat{A}_2\cdots\widehat{B}_L\widehat{A}_L|\emptyset_p\rangle \\
&= (-1)^L\det\begin{pmatrix} \mathbf{M}_{1,1} & \mathbf{M}_{1,2} & \cdots & \mathbf{M}_{1,L} \\ \mathbf{M}_{2,1} & \mathbf{M}_{2,2} & \cdots & \mathbf{M}_{2,L} \\ \vdots & \vdots & \ddots & \vdots \\ \mathbf{M}_{L,1} & \mathbf{M}_{L,2} & \cdots & \mathbf{M}_{L,L} \end{pmatrix}_{L\times L}.
\end{aligned}
\tag{288}
$$

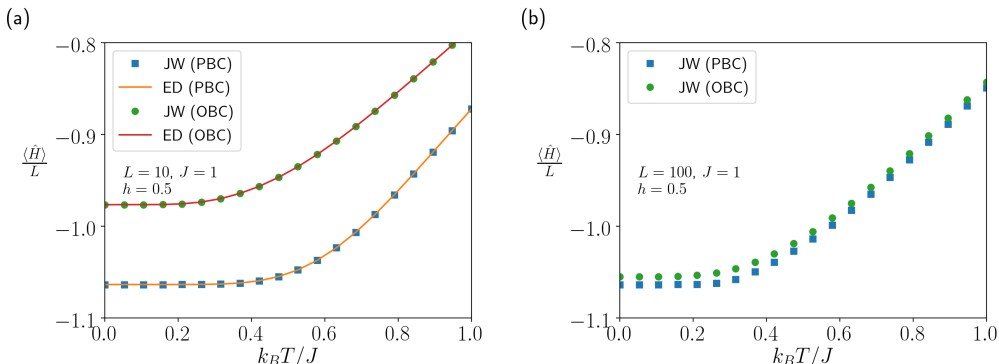

Figure 11: (a) Comparison between Jordan-Wigner (JW) and numerical Exact Diagonalization (ED) results for the average energy density, on a small ordered Ising chain ($L = 10$). (Right panel) Open boundary conditions (OBC) and periodic boundary conditions (PBC) are compared for a larger chain ($L = 100$), where ED cannot be performed. The system considered is always a Ising chain with $J_j = J = 1$ and $h_j = h = 0.5J$, with $\kappa = 1$. For OBC we set $J_L = 0$.

Hence, defining:

$$\eta_{\mathrm{p}} = (-1)^{\mathrm{p}} \langle \emptyset_{\mathrm{p}} | e^{i\pi\widehat{N}} | \emptyset_{\mathrm{p}} \rangle, \tag{289}$$

we finally get:

$$\mathrm{Tr}\left(\hat{O}\,e^{-\beta\widehat{H}}\right) = \frac{1}{2}\sum_{\mathrm{p}=0,1}\left(\mathrm{Tr}\left(\hat{O}\,e^{-\beta\widehat{\mathbb{H}}_{\mathrm{p}}}\right) + \eta_{\mathrm{p}}\,\mathrm{Tr}\left(\hat{O}\,e^{-\beta\widehat{\mathbb{H}}_{\mathrm{p}}+i\pi\sum_{\mu=1}^{L}\hat{\gamma}_{\mathrm{p},\mu}^{\dagger}\hat{\gamma}_{\mathrm{p},\mu}}\right)\right). \tag{290}$$

In particular, the partition function can be expressed as:

$$Z = \mathrm{Tr}\left(e^{-\beta\widehat{H}}\right) = \frac{1}{2}\sum_{\mathrm{p}=0,1}\left(\mathrm{Tr}\left(e^{-\beta\widehat{\mathbb{H}}_{\mathrm{p}}}\right) + \eta_{\mathrm{p}}\,\mathrm{Tr}\left(e^{-\beta\widehat{\mathbb{H}}_{\mathrm{p}}-i\pi\sum_{\mu=1}^{L}\hat{\gamma}_{\mathrm{p},\mu}^{\dagger}\hat{\gamma}_{\mathrm{p},\mu}}\right)\right)$$

$$= \frac{1}{2}\sum_{\mathrm{p}=0,1}e^{\beta\sum_{\mu=1}^{L}\epsilon_{\mathrm{p},\mu}}\left(\prod_{\mu=1}^{L}(1+e^{-2\beta\epsilon_{\mathrm{p},\mu}}) + \eta_{\mathrm{p}}\prod_{\mu=1}^{L}(1-e^{-2\beta\epsilon_{\mathrm{p},\mu}})\right). \tag{291}$$

The relevant single-particle thermal averages needed are then:[50]

$$\langle\hat{\gamma}_{\mathrm{p},\mu}^{\dagger}\hat{\gamma}_{\mathrm{p},\mu'}\widehat{P}_{\mathrm{p}}\rangle = \frac{1}{Z}\langle\hat{\gamma}_{\mathrm{p},\mu}^{\dagger}\hat{\gamma}_{\mathrm{p},\mu'}\widehat{P}_{\mathrm{p}}e^{-\beta\widehat{H}}\rangle$$

$$= \frac{1}{2Z}\left(\mathrm{Tr}\left(\hat{\gamma}_{\mathrm{p},\mu}^{\dagger}\hat{\gamma}_{\mathrm{p},\mu'}\,e^{-\beta\widehat{\mathbb{H}}_{\mathrm{p}}}\right) + \eta_{\mathrm{p}}\,\mathrm{Tr}\left(\hat{\gamma}_{\mathrm{p},\mu}^{\dagger}\hat{\gamma}_{\mathrm{p},\mu'}\,e^{-\beta\widehat{\mathbb{H}}_{\mathrm{p}}+i\pi\sum_{\mu=1}^{L}\hat{\gamma}_{\mathrm{p},\mu}^{\dagger}\hat{\gamma}_{\mathrm{p},\mu}}\right)\right)$$

$$= \delta_{\mu,\mu'}\frac{e^{-2\beta\epsilon_{\mathrm{p},\mu}}}{2Z}\left(\prod_{\substack{l=1\\l\neq\mu}}^{L}\left(1+e^{-2\beta\epsilon_{\mathrm{p},l}}\right) - \eta_{\mathrm{p}}\prod_{\substack{l=1\\l\neq\mu}}^{L}\left(1-e^{-2\beta\epsilon_{\mathrm{p},l}}\right)\right), \tag{294}$$

---

[50]Observe that to calculate the fermionic single-particle Green's functions we need the separate ingredients for the two sub-sectors p:

$$\mathbb{G} = \left\langle\widehat{\boldsymbol{\Psi}}\,\widehat{\boldsymbol{\Psi}}^{\dagger}\right\rangle = \sum_{\mathrm{p}=0,1}\left\langle\widehat{\boldsymbol{\Psi}}\,\widehat{\boldsymbol{\Psi}}^{\dagger}\widehat{P}_{\mathrm{p}}\right\rangle = \sum_{\mathrm{p}=0,1}\mathbb{U}_{\mathrm{p}}\left\langle\widehat{\boldsymbol{\Phi}}_{\mathrm{p}}\,\widehat{\boldsymbol{\Phi}}_{\mathrm{p}}^{\dagger}\widehat{P}_{\mathrm{p}}\right\rangle\mathbb{U}_{\mathrm{p}}^{\dagger}, \tag{292}$$

where for every p-sector we defined (see Sec. 7)

$$\widehat{\boldsymbol{\Phi}}_{\mathrm{p}} = \left(\begin{array}{c}\hat{\boldsymbol{\gamma}}_{\mathrm{p}} \\ \hat{\boldsymbol{\gamma}}_{\mathrm{p}}^{\dagger}\end{array}\right) = \mathbb{U}_{\mathrm{p}}^{\dagger}\widehat{\boldsymbol{\Psi}}. \tag{293}$$

Since, in (292) $\mathbb{U}_{\mathrm{p}}$ depends on p, we have a weighted sum: It is not enough to calculate directly $\sum_{\mathrm{p}=0,1}\left\langle\widehat{\boldsymbol{\Phi}}_{\mathrm{p}}\,\widehat{\boldsymbol{\Phi}}_{\mathrm{p}}^{\dagger}\widehat{P}_{\mathrm{p}}\right\rangle$.

and:

$$
\begin{aligned}
\langle \hat{\gamma}_{\mathrm{p},\mu} \hat{\gamma}^{\dagger}_{\mathrm{p},\mu'} \widehat{\mathrm{P}}_{\mathrm{p}} \rangle &= \frac{1}{Z} \langle \hat{\gamma}_{\mathrm{p},\mu} \hat{\gamma}^{\dagger}_{\mathrm{p},\mu'} \widehat{\mathrm{P}}_{\mathrm{p}} \mathrm{e}^{-\beta \widehat{H}} \rangle \\
&= \frac{1}{2Z} \left( \mathrm{Tr}\left( \hat{\gamma}_{\mathrm{p},\mu} \hat{\gamma}^{\dagger}_{\mathrm{p},\mu'} \, \mathrm{e}^{-\beta \widehat{\mathbb{H}}_{\mathrm{p}}} \right) + \eta_{\mathrm{p}} \, \mathrm{Tr}\left( \hat{\gamma}_{\mathrm{p},\mu} \hat{\gamma}^{\dagger}_{\mathrm{p},\mu'} \, \mathrm{e}^{-\beta \widehat{\mathbb{H}}_{\mathrm{p}} + i\pi \sum_{\mu=1}^{L} \hat{\gamma}^{\dagger}_{\mathrm{p},\mu} \hat{\gamma}_{\mathrm{p},\mu}} \right) \right) \\
&= \delta_{\mu,\mu'} \frac{1}{2Z} \left( \prod_{\substack{l=1 \\ l \neq \mu}}^{L} \left( 1 + \mathrm{e}^{-2\beta \epsilon_{\mathrm{p},l}} \right) + \eta_{\mathrm{p}} \prod_{\substack{l=1 \\ l \neq \mu}}^{L} \left( 1 - \mathrm{e}^{-2\beta \epsilon_{\mathrm{p},l}} \right) \right).
\end{aligned} \tag{295}
$$

Clearly, the averages where you destroy or create two $\hat{\gamma}$ fermions vanish:

$$
\langle \hat{\gamma}_{\mathrm{p},\mu} \hat{\gamma}_{\mathrm{p},\mu'} \widehat{\mathrm{P}}_{\mathrm{p}} \rangle = 0, \qquad \text{and} \qquad \langle \hat{\gamma}^{\dagger}_{\mathrm{p},\mu} \hat{\gamma}^{\dagger}_{\mathrm{p},\mu'} \widehat{\mathrm{P}}_{\mathrm{p}} \rangle = 0. \tag{296}
$$

> ℹ️ **Info:** From the averages of the Bogoliubov operators, it is a simple matter to reconstruct all elements of the ordinary and anomalous thermal Green's functions for the original fermions, as defined in Sec. 8.2. From these, using Wick's theorem, other thermal averages and correlation functions can be calculated, see Sec. 9.

We have tested these formulas on a chain of length $L = 10$. In the left panel of Fig. 11 we show the average internal energy $E = \langle \widehat{H} \rangle$ for $L = 10$, comparing the Jordan-Wigner results to those obtained by Exact Diagonalisation (ED) of the problem with both open and periodic boundary conditions. In the right panel of Fig. 11 we compare, for $L = 100$, the thermal averages for $E = \langle \widehat{H} \rangle$ obtained with the Jordan-Wigner approach for OBC and PBC.

---

**Problem 14. Free-energy and entropy.**

Consider a uniform quantum Ising chain with PBC. Using the techniques explained, calculate (numerically) the free energy per site, $F/L = -\frac{1}{\beta L} \log Z$, and the entropy per site, $S/L = (E-F)/(LT)$, as a function of the temperature $T$, for three representative values of the (uniform) transverse field: $h = 0.5J$, $h = J$ and $h = 2J$.

---

# 12 Conclusion

We have presented a pedagogical description of the numerical and analytical methods for studying the statics and the dynamics of the quantum Ising model in a transverse field and, in general, of quadratic fermionic models. These methods are useful because these models are at the center of many recent studies in quantum many-body non-equilibrium physics, due to their rich phenomenology and the fact that the numerical analysis is feasible up to very large system sizes.

We have started with the Jordan-Wigner transformation, needed to write the Ising chain in the form of a quadratic fermionic model (known as the Kitaev model) and we have diagonalized it in the case of a uniform chain. We have given special attention to the second-order phase transition of this model and the connection with the classical phase transition of the Ising model in two dimensions.

Then we moved to the diagonalization of the generic disordered model. We have introduced the Nambu formalism, which has allowed us to show that the ground state has a BCS (or Gaussian) form, and to get the Bogoliubov-de Gennes equations for the diagonalization. To find the quasiparticle excitations, and then find eigenvalues and eigenstates, one reduces to diagonalize a matrix with size quadratic in the system size, which allows to scale up to large chains. We have applied these methods to the open Kitaev chain, to show the existence of the zero-energy Majorana modes in the topological phase of this model.

Then we have considered the case of the dynamics, where a very similar formalism is valid and the state has still a Gaussian form for which Wick's theorem is valid. Here the dynamical Bogoliubov-de Gennes equations are valid, that are similar to a set of Schrödinger equations for a single quasi-particle Hamiltonian. Their solution provides all the properties of the Gaussian state and applying Wick's theorem one can get the expectations of all the observables.

Using this formalism, valid both for the ground state and the dynamics, we have then studied how to get from the Gaussian state some important properties of the system, such as the spin correlators and the entanglement entropy. The formulae we get are valid not only for the Ising model but also for generic quadratic fermionic systems.

Finally, we moved to consider thermal averages. We have shown how to compute the expectation of the observables at finite temperature by paying attention to fermion parity effects, that are important at finite size.

# Acknowledgments

GES acknowledges that his research has been conducted within the framework of the Trieste Institute for Theoretical Quantum Technologies (TQT).

**Funding information**  The research was partly supported by EU Horizon 2020 under ERC-ULTRADISS, Grant Agreement No. 834402. A. R. acknowledges financial support from PNRR MUR Project PE0000023-NQSTI and thanks SISSA for the warm hospitality received during the preparation of this work. G.E.S. acknowledges financial support from the PNRR MUR project PE0000023-NQSTI, and from the project QuantERA II Programme STAQS project that has received funding from the European Union's Horizon 2020 research and innovation programme under Grant Agreement No 101017733.

# A  Overlap between BCS states

Sometimes, for instance, in the context of quantum quenches, where the Hamiltonian is abruptly changed, it is important to know how to calculate the overlap between BCS states belonging to two different XY-Ising Hamiltonians $\widehat{H}_0$ and $\widehat{H}_1$. This appendix gives details on this.

Let us start considering the two BCS ground states of $\widehat{H}_0$ and $\widehat{H}_1$. These two states are Bogoliubov vacua for the fermionic operators $\hat{\gamma}_{0\mu}$ and $\hat{\gamma}_{1\mu}$, and we denote them, for a more compact notation, as $\left|\emptyset_{\gamma_0}\right\rangle = |\emptyset_0\rangle$ and $\left|\emptyset_{\gamma_1}\right\rangle = |\emptyset_1\rangle$. We will first compute $|\langle\emptyset_1|\emptyset_0\rangle|^2$, and then we will extend the result to the overlap of general excited states. The two sets of fermions can be written in terms of the original Jordan-Wigner fermions as:

$$\begin{pmatrix} \hat{\boldsymbol{\gamma}}_\alpha \\ \hat{\boldsymbol{\gamma}}_\alpha^\dagger \end{pmatrix} = \mathbb{U}_\alpha^\dagger \widehat{\boldsymbol{\Psi}} = \begin{pmatrix} \mathbf{U}_\alpha^\dagger & \mathbf{V}_\alpha^\dagger \\ \mathbf{V}_\alpha^\mathsf{T} & \mathbf{U}_\alpha^\mathsf{T} \end{pmatrix} \begin{pmatrix} \hat{\mathbf{c}} \\ \hat{\mathbf{c}}^\dagger \end{pmatrix}, \tag{A.1}$$

where $\alpha = 0, 1$. We can write the direct unitary transformation from one set to the other as follows:

$$\begin{pmatrix} \hat{\gamma}_1 \\ \hat{\gamma}_1^\dagger \end{pmatrix} = \mathbb{U}_1^\dagger \mathbb{U}_0 \begin{pmatrix} \hat{\gamma}_0 \\ \hat{\gamma}_0^\dagger \end{pmatrix} = \mathbb{U}^\dagger \begin{pmatrix} \hat{\gamma}_0 \\ \hat{\gamma}_0^\dagger \end{pmatrix} = \begin{pmatrix} \mathbf{U}^\dagger & \mathbf{V}^\dagger \\ \mathbf{V}^\mathrm{T} & \mathbf{U}^\mathrm{T} \end{pmatrix} \begin{pmatrix} \hat{\gamma}_0 \\ \hat{\gamma}_0^\dagger \end{pmatrix}, \tag{A.2}$$

where:

$$\mathbb{U} \equiv \begin{pmatrix} \mathbf{U} & \mathbf{V}^* \\ \mathbf{V} & \mathbf{U}^* \end{pmatrix}, \tag{A.3}$$

with:

$$\mathbf{U} = \mathbf{U}_0^\dagger \mathbf{U}_1 + \mathbf{V}_0^\dagger \mathbf{V}_1, \qquad \mathbf{V} = \mathbf{V}_0^\mathrm{T} \mathbf{U}_1 + \mathbf{U}_0^\mathrm{T} \mathbf{V}_1. \tag{A.4}$$

We will prove that, if $|\emptyset_0\rangle$ and $|\emptyset_1\rangle$ are not orthogonal, then:

$$|\langle \emptyset_1 | \emptyset_0 \rangle|^2 = |\det(\mathbf{U})|, \tag{A.5}$$

a relationship which is known as *Onishi formula*. Indeed, we have already given proof of this relationship in Sec. 7.3, for the special case in which one of the two sets of fermions were the original Jordan-Wigner fermions $\hat{c}_j$ with associated vacuum state $|0\rangle$. There we showed that, with the present notation:

$$|\emptyset_\alpha\rangle = \mathcal{N}_\alpha \exp\Big( \frac{1}{2} \sum_{j_1 j_2} (\mathbf{Z}_\alpha)_{j_1 j_2} \hat{c}_{j_1}^\dagger \hat{c}_{j_2}^\dagger \Big) |0\rangle, \tag{A.6}$$

with:

$$\mathbf{Z}_\alpha = -(\mathbf{U}_\alpha^\dagger)^{-1} \mathbf{V}_\alpha^\dagger, \tag{A.7}$$

and $|\langle 0 | \emptyset_\alpha \rangle|^2 = |\mathcal{N}_\alpha|^2 = |\det(\mathbf{U}_\alpha)|$. With the same algebra, we could establish, for instance, that:

$$|\emptyset_1\rangle = \mathcal{N} \, e^{\mathcal{Z}} |\emptyset_0\rangle = \mathcal{N} \exp\Big( \frac{1}{2} \sum_{j_1 j_2} \mathbf{Z}_{j_1 j_2} \hat{\gamma}_{0,j_1}^\dagger \hat{\gamma}_{0,j_2}^\dagger \Big) |\emptyset_0\rangle, \tag{A.8}$$

with:

$$\mathbf{Z} = -(\mathbf{U}^\dagger)^{-1} \mathbf{V}^\dagger, \tag{A.9}$$

and $|\langle \emptyset_0 | \emptyset_1 \rangle|^2 = |\mathcal{N}|^2 = |\det(\mathbf{U})|$. Several points in the previous derivation would call for appropriate specifications: for instance, we assumed that $\mathbf{U}$ is invertible. Also, the case of possible orthogonality of the two ground states was not discussed. Finally, the case of pure Slater determinant *without BCS-pairing*, relevant for the isotropic XY model, was not explicitly addressed.

We will now give an alternative proof that makes use of an interesting theorem due to Bloch and Messiah [83, 137], and which clarifies all these issues. We will perform an intermediate canonical transformation which first allows us to write an explicit equation for $|\emptyset_1\rangle$ in terms of $|\emptyset_0\rangle$, and then to compute easily $\langle \emptyset_1 | \emptyset_0 \rangle$. The theorem that Bloch and Messiah proved [137] shows that matrices with the structure of $\mathbb{U}$ above can be decomposed into a product of three unitary transformations as follows:

$$\mathbb{U} = \begin{pmatrix} \mathbf{D} & \mathbf{0} \\ \mathbf{0} & \mathbf{D}^* \end{pmatrix} \begin{pmatrix} \overline{\mathbf{U}} & \overline{\mathbf{V}} \\ \overline{\mathbf{V}} & \overline{\mathbf{U}} \end{pmatrix} \begin{pmatrix} \mathbf{C} & \mathbf{0} \\ \mathbf{0} & \mathbf{C}^* \end{pmatrix}, \tag{A.10}$$

SciPost Phys. Lect. Notes 82 (2024)

where $\mathbf{D}$, $\mathbf{C}$ are $L \times L$ unitary matrices and $\overline{\mathbf{U}}$, $\overline{\mathbf{V}}$ are $L \times L$ real matrices of the form:

$$
\overline{\mathbf{U}} = \begin{pmatrix}
0 & & & & & & & & & \\
& \ddots & & & & & & & & \\
& & 0 & & & & & & & \\
& & & u_1 & 0 & & & & & \\
& & & 0 & u_1 & & & & & \\
& & & & & \ddots & & & & \\
& & & & & & u_n & 0 & & \\
& & & & & & 0 & u_n & & \\
& & & & & & & & 1 & \\
& & & & & & & & & \ddots \\
& & & & & & & & & & 1
\end{pmatrix}, \tag{A.11}
$$

$$
\overline{\mathbf{V}} = \begin{pmatrix}
1 & & & & & & & & & \\
& \ddots & & & & & & & & \\
& & 1 & & & & & & & \\
& & & 0 & v_1 & & & & & \\
& & & -v_1 & 0 & & & & & \\
& & & & & \ddots & & & & \\
& & & & & & 0 & v_n & & \\
& & & & & & -v_n & 0 & & \\
& & & & & & & & 0 & \\
& & & & & & & & & \ddots \\
& & & & & & & & & & 0
\end{pmatrix}, \tag{A.12}
$$

in which $u_p > 0$, $v_p > 0$ and $u_p^2 + v_p^2 = 1$. From these relations we have:

$$
\mathbf{U} = \mathbf{D}\overline{\mathbf{U}}\mathbf{C}, \qquad \mathbf{V} = \mathbf{D}^*\overline{\mathbf{V}}\mathbf{C}. \tag{A.13}
$$

To proceed, we now notice that since:

$$
\begin{pmatrix} \hat{\boldsymbol{\gamma}}_0 \\ \hat{\boldsymbol{\gamma}}_0^\dagger \end{pmatrix} = \mathbb{U} \begin{pmatrix} \hat{\boldsymbol{\gamma}}_1 \\ \hat{\boldsymbol{\gamma}}_1^\dagger \end{pmatrix} = \begin{pmatrix} \mathbf{D} & \mathbf{0} \\ \mathbf{0} & \mathbf{D}^* \end{pmatrix} \begin{pmatrix} \overline{\mathbf{U}} & \overline{\mathbf{V}} \\ \overline{\mathbf{V}} & \overline{\mathbf{U}} \end{pmatrix} \begin{pmatrix} \mathbf{C} & \mathbf{0} \\ \mathbf{0} & \mathbf{C}^* \end{pmatrix} \begin{pmatrix} \hat{\boldsymbol{\gamma}}_1 \\ \hat{\boldsymbol{\gamma}}_1^\dagger \end{pmatrix}, \tag{A.14}
$$

we can think of the transformation as the product of **1)** a first unitary transformation $\mathbf{C}$ which does not mix particles and holes for fermions $\hat{\boldsymbol{\gamma}}_1$, defined by

$$
\begin{pmatrix} \hat{\boldsymbol{\alpha}}_1 \\ \hat{\boldsymbol{\alpha}}_1^\dagger \end{pmatrix} = \begin{pmatrix} \mathbf{C} & \mathbf{0} \\ \mathbf{0} & \mathbf{C}^* \end{pmatrix} \begin{pmatrix} \hat{\boldsymbol{\gamma}}_1 \\ \hat{\boldsymbol{\gamma}}_1^\dagger \end{pmatrix}, \tag{A.15}
$$

followed by **2)** a simple "canonical form" of a transformation leading to new fermions:

$$
\begin{pmatrix} \hat{\boldsymbol{\alpha}}_0 \\ \hat{\boldsymbol{\alpha}}_0^\dagger \end{pmatrix} = \begin{pmatrix} \overline{\mathbf{U}} & \overline{\mathbf{V}} \\ \overline{\mathbf{V}} & \overline{\mathbf{U}} \end{pmatrix} \begin{pmatrix} \hat{\boldsymbol{\alpha}}_1 \\ \hat{\boldsymbol{\alpha}}_1^\dagger \end{pmatrix}, \tag{A.16}
$$

and a final transformation **3)** leading to the fermions $\hat{\boldsymbol{\gamma}}_0$, through a unitary $\mathbf{D}$ which does not mix particles and holes:

$$
\begin{pmatrix} \hat{\boldsymbol{\gamma}}_0 \\ \hat{\boldsymbol{\gamma}}_0^\dagger \end{pmatrix} = \begin{pmatrix} \mathbf{D} & \mathbf{0} \\ \mathbf{0} & \mathbf{D}^* \end{pmatrix} \begin{pmatrix} \hat{\boldsymbol{\alpha}}_0 \\ \hat{\boldsymbol{\alpha}}_0^\dagger \end{pmatrix}. \tag{A.17}
$$

In essence, what the Bloch-Messiah theorem guarantees is that one can always find a basis such that the transformed fermions, $\hat{\boldsymbol{\alpha}}_0$ and $\hat{\boldsymbol{\alpha}}_1$, are coupled by a particularly simple matrix in which there are only three possibilities: *i*) for some indices, which we denote by $l$, there is no transformation at all (the 1s in the diagonal of $\overline{\mathbf{U}}$), i.e., $\hat{\alpha}_{1l} = \hat{\alpha}_{0l}$; *ii*) for some other indices, which we denote by $k$, the transformation is a pure particle-hole $\hat{\alpha}_{1k}^{\dagger} = \hat{\alpha}_{0k}$: These indices correspond to the 0s in the diagonal of $\overline{\mathbf{U}}$, and the 1s in the diagonal of $\overline{\mathbf{V}}$; *iii*) all other indices, denoted by $(p, \overline{p})$, are BCS-paired in a simple way, and they form $2 \times 2$ blocks in the matrices $\overline{\mathbf{U}}$ and $\overline{V}$ with coefficients $u_p$ and $v_p$, such that:

$$
\begin{aligned}
\hat{\alpha}_{1p}^{\dagger} &= u_p \hat{\alpha}_{0p}^{\dagger} - v_p \hat{\alpha}_{0\overline{p}}, \\
\hat{\alpha}_{1\overline{p}}^{\dagger} &= u_p \hat{\alpha}_{0\overline{p}}^{\dagger} + v_p \hat{\alpha}_{0p}.
\end{aligned}
\tag{A.18}
$$

We must stress that the theorem does not tell us *how many* indices belong to the three categories above: In some cases, all the indices might be $2 \times 2$-paired, but it is also possible that the transformation is a pure particle-hole transformation without any pairing at all.

The construction of the relationship between $|\emptyset_0\rangle$ and $|\emptyset_1\rangle$ becomes particularly simple in terms for the fermions $\hat{\boldsymbol{\alpha}}_{0(1)}$. The key idea is the $\hat{\boldsymbol{\alpha}}_{0(1)}$ is related to $\hat{\boldsymbol{\gamma}}_{0(1)}$ by a transformation which does not mix particles and holes, and therefore it is still true that $\hat{\alpha}_{0n} |\emptyset_0\rangle = 0$ and $\hat{\alpha}_{1n} |\emptyset_1\rangle = 0$. Since $|\emptyset_1\rangle$ is the state which is annihilated by any $\hat{\alpha}_{1n}$ we can write it as:

$$
|\emptyset_1\rangle = \mathcal{N} \prod_n \hat{\alpha}_{1n} |\emptyset_0\rangle = \prod_k \hat{\alpha}_{0k}^{\dagger} \prod_p \left( u_p + v_p \hat{\alpha}_{0p}^{\dagger} \hat{\alpha}_{0\overline{p}}^{\dagger} \right) |\emptyset_0\rangle,
\tag{A.19}
$$

where $\mathcal{N}$ is a normalization constant. Notice that we included only BCS-paired indices and particle-hole transformed $k$-indices but *not* $l$-indices, since $\hat{\alpha}_{1l} = \hat{\alpha}_{0l}$ and the inclusion of such terms would give zero, since $\hat{\alpha}_{0l} |\emptyset_0\rangle = 0$. Since, by hypothesis, the two states $|\emptyset_0\rangle$ and $|\emptyset_1\rangle$ are not orthogonal there should not be pure particles-holes $k$-indices either, and therefore:

$$
\langle \emptyset_0 | \emptyset_1 \rangle = \langle \emptyset_0 | \prod_p \left( u_p + v_p \hat{\alpha}_{0p}^{\dagger} \hat{\alpha}_{0\overline{p}}^{\dagger} \right) |\emptyset_0\rangle = \prod_p u_p = \sqrt{\prod_p u_p^2} = \sqrt{\det(\overline{\mathbf{U}})}.
\tag{A.20}
$$

Finally, since $\overline{\mathbf{U}} = \mathbf{D}^{\dagger} \mathbf{U} \mathbf{C}^{\dagger}$, and $\mathbf{D}$, and $\mathbf{C}$ are unitary transformations:

$$
|\langle \emptyset_0 | \emptyset_1 \rangle|^2 = |\det(\mathbf{D}^{\dagger} \mathbf{U} \mathbf{C}^{\dagger})| = |\det(\mathbf{U})|,
\tag{A.21}
$$

which is what we wanted to show.

The extension to the calculation of the overlap between $|\emptyset_0\rangle$ and any eigenstate $\left| \{n_{1\mu}\} \right\rangle = \prod_{\mu \in I} \hat{\gamma}_{1\mu}^{\dagger} |\emptyset_1\rangle$, where $I$ is the set of occupied states ($n_{1\mu} = 1$), is in principle straightforward. Here is a possible way to tackle the problem.

This state can be thought of as the vacuum of the following new set of fermions:

$$
\hat{\beta}_{\mu}^{\dagger} = \hat{\gamma}_{1\mu}^{\dagger}, \quad \text{if } \mu \notin I, \qquad \hat{\beta}_{\mu}^{\dagger} = \hat{\gamma}_{1\mu}, \quad \text{if } \mu \in I,
\tag{A.22}
$$

in which we have performed a particle-hole transformation for the occupied modes. Now we can use the equation obtained for the scalar product between empty states, i.e.,

$$
\left| \langle \emptyset_0 | \{n_{1\mu}\} \rangle \right|^2 = \left| \det(\mathbf{U}') \right|,
\tag{A.23}
$$

where the matrix $\mathbf{U}'$ is:

$$
\mathbf{U}' = \mathbf{U}_0^{\dagger} \mathbf{U}_1' + \mathbf{V}_0^{\dagger} \mathbf{V}_1',
\tag{A.24}
$$

in which:

$$
\begin{aligned}
(\mathbf{U}_1')_{j\mu} &= (\mathbf{U}_1)_{j\mu}, \quad \text{if } \mu \notin, & I(\mathbf{U}_1')_{j\mu} &= (\mathbf{V}_1^*)_{j\mu}, \quad \text{if } \mu \in I, \\
(\mathbf{V}_1')_{j\mu} &= (\mathbf{V}_1)_{j\mu}, \quad \text{if } \mu \notin, & I(\mathbf{V}_1')_{j\mu} &= (\mathbf{U}_1^*)_{j\mu}, \quad \text{if } \mu \in I.
\end{aligned}
\tag{A.25}
$$

A second approach to calculating these overlaps with excited states makes explicit use of the Gaussian nature of the states. The relevant algebra follows directly from that of Sec. 7.3. Let us start by considering the overlap between $\hat{\gamma}_{0\mu_1}^\dagger \hat{\gamma}_{0\mu_2}^\dagger |\emptyset_0\rangle$ and $|\emptyset_1\rangle = \mathcal{N} e^{\mathcal{Z}} |\emptyset_0\rangle$. This is given by:

$$
\begin{aligned}
\langle \emptyset_0 | \hat{\gamma}_{0\mu_2} \hat{\gamma}_{0\mu_1} | \emptyset_1 \rangle &= \mathcal{N} \langle \emptyset_0 | \hat{\gamma}_{0\mu_2} \hat{\gamma}_{0\mu_1} e^{\mathcal{Z}} | \emptyset_0 \rangle \\
&= \mathcal{N} \langle \emptyset_0 | e^{\mathcal{Z}} \Big( \hat{\gamma}_{0\mu_2} + \sum_{\mu_2'} \mathbf{Z}_{\mu_2 \mu_2'} \hat{\gamma}_{0\mu_2'}^\dagger \Big) \Big( \hat{\gamma}_{0\mu_1} + \sum_{\mu_1'} \mathbf{Z}_{\mu_1 \mu_1'} \hat{\gamma}_{0\mu_1'}^\dagger \Big) | \emptyset_0 \rangle \\
&= \mathcal{N} \langle \emptyset_0 | e^{\mathcal{Z}} \hat{\gamma}_{0\mu_2} \Big( \sum_{\mu_1'} \mathbf{Z}_{\mu_1 \mu_1'} \hat{\gamma}_{0\mu_1'}^\dagger \Big) | \emptyset_0 \rangle = \langle \emptyset_0 | \emptyset_1 \rangle \, \mathbf{Z}_{\mu_1 \mu_2},
\end{aligned}
$$

wherein the second step we have made use of the commutation property:

$$
\hat{\gamma}_{0\mu} e^{\mathcal{Z}} = e^{\mathcal{Z}} \left( \hat{\gamma}_{0\mu} + [\hat{\gamma}_{0\mu}, \mathcal{Z}] \right) = e^{\mathcal{Z}} \left( \hat{\gamma}_{0\mu} + \sum_{\mu'} \mathbf{Z}_{\mu\mu'} \hat{\gamma}_{0\mu'}^\dagger \right).
\tag{A.26}
$$

Notice that, for the overlap to be non-vanishing, we were forced to contract $\hat{\gamma}_{0\mu_2}$ against $\hat{\gamma}_{0\mu_1'}^\dagger$ in the final step. A similar calculation shows that, if we have an *even* number $2n$ of operators, the result is highly reminiscent of Wick's theorem sum-of-products of contractions:

$$
\begin{aligned}
\langle \emptyset_0 | \hat{\gamma}_{0\mu_{2n}} \cdots \hat{\gamma}_{0\mu_1} | \emptyset_1 \rangle &= \mathcal{N} \langle \emptyset_0 | e^{\mathcal{Z}} \Big( \hat{\gamma}_{0\mu_{2n}} + \sum_{\mu_{2n}'} \mathbf{Z}_{\mu_{2n} \mu_{2n}'} \hat{\gamma}_{0\mu_{2n}'}^\dagger \Big) \cdots \Big( \hat{\gamma}_{0\mu_1} + \sum_{\mu_1'} \mathbf{Z}_{\mu_1 \mu_1'} \hat{\gamma}_{0\mu_1'}^\dagger \Big) | \emptyset_0 \rangle \\
&= \langle \emptyset_0 | \emptyset_1 \rangle \sum_P (-1)^P \mathbf{Z}_{\mu_{P_1} \mu_{P_2}} \mathbf{Z}_{\mu_{P_3} \mu_{P_4}} \cdots \mathbf{Z}_{\mu_{P_{2n-1}} \mu_{P_{2n}}} \\
&= \langle \emptyset_0 | \emptyset_1 \rangle \, \mathrm{Pf}(\mathbf{Z})_{2n \times 2n},
\end{aligned}
\tag{A.27}
$$

while the overlap vanishes for an odd number of $\hat{\gamma}_{0\mu_i}$. In the last expression, the Wick's sum is rewritten in terms of the so-called *Pfaffian* of the anti-symmetric matrix $\mathbf{Z}$ (or more properly, of the $2n \times 2n$ elements of $\mathbf{Z}$ required by the indices $\mu_1 \cdots \mu_{2n}$):

$$
\begin{aligned}
\mathrm{Pf}(\mathbf{Z})_{2n \times 2n} &= \mathrm{Pf} \begin{pmatrix}
0 & \mathbf{Z}_{\mu_1 \mu_2} & \mathbf{Z}_{\mu_1 \mu_3} & \cdots & \mathbf{Z}_{\mu_1 \mu_{2n}} \\
\mathbf{Z}_{\mu_2 \mu_1} & 0 & \mathbf{Z}_{\mu_2 \mu_3} & \cdots & \mathbf{Z}_{\mu_2 \mu_{2n}} \\
\vdots & \vdots & \vdots & \vdots & \vdots \\
\mathbf{Z}_{\mu_{2n} \mu_1} & \mathbf{Z}_{\mu_{2n} \mu_2} & \mathbf{Z}_{\mu_{2n} \mu_3} & \cdots & 0
\end{pmatrix} \\
&\stackrel{\text{def}}{=} \sum_P (-1)^P \underbrace{\mathbf{Z}_{\mu_{P_1} \mu_{P_2}} \mathbf{Z}_{\mu_{P_3} \mu_{P_4}} \cdots \mathbf{Z}_{\mu_{P_{2n-1}} \mu_{P_{2n}}}}_{n \text{ factors}}.
\end{aligned}
\tag{A.28}
$$

Notice that the Pfaffian is defined by a Wick's sum which contains $n$ products of $\mathbf{Z}$-matrix elements, and not $2n$, as the familiar $\det(\mathbf{Z})_{2n \times 2n}$. However, a remarkable identity exists [138] which links the two objects:

$$
\begin{aligned}
\det(\mathbf{Z})_{2n \times 2n} &= \sum_P (-1)^P \underbrace{\mathbf{Z}_{\mu_1 \mu_{P_1}} \mathbf{Z}_{\mu_2 \mu_{P_2}} \cdots \mathbf{Z}_{\mu_{2n} \mu_{P_{2n}}}}_{2n \text{ factors}} \\
&= (\mathrm{Pf}(\mathbf{Z})_{2n \times 2n})^2.
\end{aligned}
\tag{A.29}
$$

Notice, however, that the link exists only if the dimension of the antisymmetric matrix we are considering is *even*: The determinant of an odd-dimension anti-symmetric matrix is simply zero, while the Pfaffian is not defined. To summarise, we have obtained the generalization of the Onishi formula in the form:

$$\langle \emptyset_0 | \hat{\gamma}_{0\mu_{2n}} \cdots \hat{\gamma}_{0\mu_1} | \emptyset_1 \rangle = \langle \emptyset_0 | \emptyset_1 \rangle \, \mathrm{Pf}(\mathbf{Z})_{2n \times 2n} = \langle \emptyset_0 | \emptyset_1 \rangle \, (\det(\mathbf{Z})_{2n \times 2n})^{1/2} \,. \tag{A.30}$$

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
