# Peer review of "The quantum Ising chain for beginners"

_SciPost Physics Lecture Notes, doi:SciPost Phys. Lect. Notes 82 (2024)_

## Round 1 · Referee Report · Anonymous (Referee 1) · 2020-12-22

Strengths

see report

Weaknesses

see report

Report

The lecture notes "The quantum Ising chain for beginners" by G. B. Mbeng, A. Russomanno, G. E. Santoro,
constitute an introduction to various techniques allowing one to study one dimensional quantum Ising chains.
The lectures are comprehensible starting from a master level student and are written in a almost uniformly pedagogical way.
Most of the presented results come along with detailed derivation, what makes the notes essentially self contained. I can only regret that
the authors did not provide more sections relative to the explicit computation of correlation functions in the models they consider.
Still, it is more of a personal regret than a criticism on my part. To summarise, I believe that it is an interesting piece of work.

However, the only thing which does not allow me to recommend, so far, these for publication is the following sentence that appears in the
introduction: "We, unfortunately, do no justice to the immense literature where concepts and techniques were first introduced or derived, and even less so to the
many papers where physical applications are presented: this would require too much effort for our limited goals."
While such a statement would, in principle, be OK for a master lecture given on the blackboard, I do not believe that it is possible to do so
in a set of lecture notes to be published in a journal. It is not just the question of making justice to the various predecessors
that worked and pioneered the analysis related to this models. It is also the issue of giving the students the right impression about the
amount of work that was carried on these models so as to eventually get to such a nice and neat description. Finally, the references may also be of quite some
use to the researched that would need to remind themselves of this or that point of history of the model.

Therefore, for the time being, I do not recommend the lecture notes for publication, and kindly ask the authors to complement these with a detailed set of
references. If this is done, then I would not see any more obstruction for publication.

Below, I provide a short list of the missprints I have spotted:

\begin{itemize}

\item Page 9 and several instances that follow. The authors speak of the number of Eigenvalues as a way to count dimensionality. I would suggest to rather speak of the
number of Eigenvalues counted with multiplicities.

\item Eq (33). I would suggest to use a slightly different notation for the Fourier transforms of the fermion operators. For instance, to change the letter c typography.
That should lead to more clarity.

\item Discussion below (69). Taken the overall level of detail given in the lecture notes, I would suggest to provide more explicit details on the content of the paragraph.

\item I think it is rather Hermitian than Hermitean

\item "Unfortunately, the computer will produce eigenvectors associated with the degenerate zero-energy eigenvalues which do
not have the structure alluded at in Eq" I think that this statement strongly depends on the routine adopted by the computer. Maybe some more
explanation of the origin of this problem would be interesting.

\item Discussion around (114). Taken the overall level of detail given in the lecture notes, I would suggest to provide more explicit details on the derivation of (114). Here, the authors
could recall the central limit theorem and the convergence of averages and could explain a bit more the main steps.

\item I could not identify the definition of $\tau_L^{*}$ in the footnote of exercise 5 on page 40

\item "using" on page 49 is probably a typo

\item page 53, Section 10, paragraph 2 $\rho_{L} \hookrightarrow \rho_{l}$

\item $(t)$ is missing in the definition of $\mathbb{\Lambda}$, in the elements $\lambda_k$ as well as in the line below of (270).

\end{itemize}

Requested changes

see report

---

## Round 1 · Referee Report · Anonymous (Referee 2) · 2021-1-5

Strengths

1- Very detailed lectures notes. 2- Reasonably self-contained.

Weaknesses

1- Tendency to focus too much on technicalities. 2- The section on entanglement could be improved.

Report

The lecture notes 'The quantum Ising chain for beginners' are an introduction to the various set of free fermionic techniques which can be used to study the one-dimensional quantum Ising model. The focus is on explicit computations (spectrum, eigenstates, overlaps, time-dependence, entanglement), of specific but nevertheless useful quantities, which are explained in considerable detail.

Overall I find these lectures to be pedagogical, already accessible to a master-level student. The authors often 'go the extra mile', and explain all the various hidden subtleties related to boundary conditions, fermionic parity, coherent states, etc, which are sometimes neglected in other lectures.

There are also downsides to the way things are presented. Reading these notes sometimes feels like browsing through a list of specific calculations. Those are carried out without discussing motivations, why they are possible, what is important or not. Such explanations would be useful to the targeted audience, especially if they need at some point to be able to compute something slightly different from what is presented.

Here is a list of general comments, that elaborate further on this criticism. Provided those are addressed, I would be happy to recommend publication in Scipost lecture notes.

a) The introduction is too short. It would be nice to explain some of the logic of these lectures here, together with some broader 'take home messages', which would be useful to the targeted audience. For example, a common feature shared by all sections is that to compute essentially anything in the quantum Ising chain with L sites, the only thing you need to be able to do is diagonalise a matrix of size $2L$, resulting in a massive simplification.

b) I also concur with the first referee regarding the first paragraph, which, even though it was probably not the author's intentions, sounds like trying to make excuses for not providing a proper bibliography.

c) There are too many footnotes, not even taking into account footnotes in problems and info boxes. While those can be a nice complement to the text, they should be used sparingly. There are several places where those could be put in the main text, or simply removed. For example, the fact that there are 5 long footnotes on page 19 is a sign that something went very wrong with the presentation in this particular place.

d) Specific notations are used to distinguish between the various types of operators and matrices. This is a good idea, but explaining those to the nonexpert reader would be even better. For example bold capital letters are used for square matrices of size $L$, while the hat is reserved for fermionic operators, Hamiltonians, of size $2^L$. I do not fully understand the mathbb notation, however. In some case it is just a $2L\times 2L$ matrix with blocks in bold capital letters, but the hat mathbb are used to emphasize different parity sectors, for example in section 3.

e) Section 9 and especially 10 are subpar compared to the previous ones, see comments below. While computations say of the energy spectrum does not really require discussing motivations, it is clearly necessary for the entanglement entropy.

Below is a list of more specific suggestions:

1) section 1, page 3. It would be useful to explicitly state somewhere that for a collection of L spins, the dimension of the Hilbert space is 2^L. In footnote 1, remove the 'one the same site', since these are just 2 by 2 Pauli matrices, there is no notion of 'same site' yet. 'typical of fermionic commutation rules' reads awkwardly, since there is the anti-commutator.

2) Page 5. 'Notice that $K_j=K_j^\dagger=K_j^{-1}$ and $K_j^2=1$'. It was already mentioned two lines above that $K_j$ is a sign, so the reader should be able to figure this out by themselves.

3) Footnote 4 does not fit well with the spirit of these lectures, which deals almost exclusively with finite chains. It can be safely removed.

4) In section 2 and 3, there is some ambiguity in what is meant by transverse field Ising model. The text is reasonably clear in stating that J^y=0 in (18) in this case, but then discussions later on do not always make this assumption, even though section 2 and 3 are named 'Ising model'. Perhaps give a different name to (18).

5) Page 10. It seems to me the most important point is that ${c_k,c_{k'}^\dagger}=\delta_{kk'}$, which is implicitly used later on. The notation in (36), (37) are somewhat nonstandard, and advantageously replaced by $\displaystyle{\sum_{k\in \mathcal{K}_p}}$ as is done on the next page. Remove footnote 7, or put back in the main text. In footnote 9, replace the first and by a comma.

6) Page 11, before (42). $\hat{H}_k$ 'lives in the 4-dimensional space' because all terms with different |k| in the Hamiltonian commute, which is related to the previous point. Also, technically $H_k$ are still matrices of size $2^L \times 2^L$, they just happen to act nontrivially only on this four-dimensional subspace.

7) Page 13. Figure 2 is more stylish, but figure 3 provides the same information and is much more easy to visualize.

8) Page 15. First line in section 3.1. 'in Eq. (59)'. Together with (50), which is nonnegative. Just before (69), the discussion of the energy spliting is not very clear. The reader 'naively expects' (69) to hold, and they would be perfectly right, since this equation deals only with the extensive part of the energy. Subleading corrections, exponential or not, are irrelevant in (69).

9) At the beginning of section 3.2. 'In this section, we will focus on the Ising case'. But section 3 is named 'Uniform Ising model', and the same for section 3.1.

10) Page 20. Footnote 20 can be safely put back in the main text.

11) While discussing the 'Nambu formalism', it would be nice to mention how one can recover the results of the previous sections, simply because the matrices $\mathbf{A}$ and $\mathbf{B}$ have a simple translationally invariant structure.

12) Page 21. Similar abuse of notation as before, with concatenation of two vectors ${\bf u}$ and ${\bf v}$ to make a larger vector.

13) Page 22, footnote 25. 'Similarly, one must have [...]'. This does not bring additional information, since a left inverse is also a right inverse for square matrices.

14) Page 23, before (108). Replace 'The $2^L$ Fock states' by 'The $2^L$ eigenstates'.

15) Page 24. $2\epsilon_\mu \equiv \epsilon_k$ looks like a clash of notations.

16) Page 25. 'More in detail' reads awkwardly.

17) Page 25, 26. Make explicitly the point that in the absence of disorder, eigenstates are extended.

18) The discussion in section 5.3 is pretty good I think. 'a theorem of linear algebra' becomes a consequence of the Schur decomposition in a later section.

19) Page 41. The notation $1(0)$ makes the formulas harder to read. It would be better to replace by $\mathbf{Z}_{\alpha}$ with $\alpha=0,1$, or something similar.

20) Discussion after (200). I understand the point, but the assumption of invertibility is actually not a problem since everything is finite-dimensional. Invertible matrices are dense in the space of all matrices, so the Onishi formula holds as a limit when $U$ is not invertible. A mathematician would even be very happy with this. Same comment in the absence of pairings.

21) The discussion in section 9 ends quite abruptly. It would be perhaps nicer to mention that these exact formulas are the starting point of many subsequent large distance results --also related to the 2d classical Ising model-- and give a few extra references. Also, footnote a in problem 6 can be put in the text.

22) Page 50. The notation ${l}$ for subsystem is non standard and confusing, since ${l}$ is usually understood precisely as the set containing only $l$. And it gets worse with 'for instance, ${l}=1,\ldots,l$', where a standard notation for the rhs would be ${1,\ldots,l}$. Also, it is not clear whether the implication is that ${l}$ necessarily contains $l$ elements, consecutive or not.

23) Page 50, 'For a system described by a quantum state $|\psi(t)\rangle$'. It is strange to use notations implying time-dependence without ever mentioning it, since the calculation of the entanglement entropy has in principle nothing to do with time. You just need a state where Wick's theorem holds. Of course, this is also the case when time-evolving any such state with a quadratic fermions Hamiltonian. So time could in principle be removed from this whole section, resulting in lighter notations. Perhaps the authors want to make some kind of point that any state where Wick's theorem holds can be seen in this way, or just consider time-evolved BCS states, in which case they need to explain what they want to discuss, and what assumptions they make.

24) Page 52, 'its state $|\psi(t)\rangle$ is a gaussian state'. I do not know what is the state of the quantum Ising chain.

25) Page 53. There is a section 10.1, but no section 10.2. In step 2, 'to a sub-chain ${l}=1,\ldots,l$, say'. This assumption is not innocent at all. For a chain with PBC, it is in fact very important that the 'subchain' be in one connected piece. Otherwise the calculation explained in this section breaks down due to issues with Jordan-Wigner strings. That point should be made clearly, otherwise the 'beginner' reader might think this holds as well for subsystem $1,3,5,8$, which it does not. Also, subsystem is possibly clearer than subchain.

26) Page 53. Footnote 53 is not very clear, since much more than normalization is lost for unitary matrices, and the Nambu matrix is not unitary.

27) The derivation presented here is fine, but other methods should at least be mentioned as well, also in the slightly broader context of entanglement computations in free bosonic or fermionic systems, dating back to Bombelli et al, Srednicki, etc. This is perhaps a matter of taste, but I find the (closely related) derivation of Peschel to be easier to understand, especially for non experts. It just explains why the reduced density matrix (RDM) is the exponential of a quadratic fermion Hamiltonian, similar to Step 1. In fact, the explicit expression for the RDM follows from (277) in the text.

28) Problem 7. Give at least a few references for the logarithmic divergence, including the paper by Holzhey, Larsen, Wilczek.

Requested changes

See report

---

## Round 2 · Referee Report · Anonymous (Referee 2) · 2024-2-20

Strengths

See previous report.

Weaknesses

See previous report.

Report

The authors have made significant changes to the lectures notes, I believe the manuscript has improved.

I recommend publication, provided the minor issues listed below are addressed.

Requested changes

1) In page 3, say that H.c means Hermitian conjugate.

2) Info box on page 10: Replace 'by Hamiltonian' by 'by the Hamiltonian'.

3) Page 19. I find the discussion after equation (74) to be a little bit confusing. There is nothing 'cavalier' about how the limit is taken, the energy per site as defined is independent on boundary conditions in the thermodynamic limit. Asking for the behavior of the energy splitting is a different question.

4) In section 4, page 24: remove one of the two 'celebrated' on the second and third lines. At the bottom of the page 'denotes the so-called transfer matrix'. This is not the transfer matrix, this is a matrix element. $T$ is the transfer matrix.

5) It should perhaps be stated somewhere that the transfer matrix of the Ising model can be written as the exponential of a quadratic form in the fermionic operators, so can in principle be solved using the techniques presented later in the notes.

6) In at least two exercises (Problem 1 and Problem 10) the questions are formulated as 'find the value of' or 'evaluate the integral' without giving the answer one should find. However, this very answer is then put in a very visible footnote right below. It would be better to either remove the footnote, or put its content in the formulation of the question.

---

## Round 2 · Author Response

Warnings issued while processing user-supplied markup:

  • Inconsistency: plain/Markdown and reStructuredText syntaxes are mixed. Markdown will be used.
    Add "#coerce:reST" or "#coerce:plain" as the first line of your text to force reStructuredText or no markup.
    You may also contact the helpdesk if the formatting is incorrect and you are unable to edit your text.

Dear Editor, first of all, we would like to beg your pardon for the quite long time we have taken to resubmit our manuscript: needless to say, this being not our daily research work, and the additions made being quite extensive, it took us almost three years to respond. We hope you will understand this. Next, we would like to thank the Referees for their time and effort in assessing our work — truly impressive and detailed reports were provided — which led to a greatly improved manuscript. Almost all the modifications introduced in the text are marked in red. As a major addition, we have now an entirely new section 4 on the connection between the classical Ising model in two dimensions and the quantum Ising chain. Secs. 3.1.1 and 3.1.2 are also new. Please find below a detailed reply to all the Referees’ queries.

Yours faithfully, Glen Bigan Mbeng (on behalf of the authors)

Report 1

--- Query The lecture notes ”The quantum Ising chain for beginners” by G. B. Mbeng, A.Russomanno, G. E. Santoro, constitute an introduction to various techniques allowing one to study one-dimensional quantum Ising chains. The lectures are comprehensible starting from a master level student and are written in a almost uniformly pedagogical way. Most of the presented results come along with detailed derivation, what makes the notes essentially self contained. I can only regret that the authors did not provide more sections relative to the explicit computation of correlation functions in the models they consider. Still, it is more of a personal regret than a criticism on my part. To summarise, I believe that it is an interesting piece of work. --- Answer We thank the Referee for her/his positive assessment towards our work.

--- Query However, the only thing which does not allow me to recommend, so far, these for publication is the following sentence that appears in the introduction: ”We, unfortunately, do no justice to the immense literature where concepts and techniques were first introduced or derived, and even less so to the many papers where physical applications are presented: this would require too much effort for our limited goals.” While such a statement would, in principle, be OK for a master lecture given on the blackboard, I do not believe that it is possible to do so in a set of lecture notes to be published in a journal. It is not just the question of making justice to the various predecessors that worked and pioneered the analysis related to this models. It is also the issue of giving the students the right impression about the amount of work that was carried on these models so as to eventually get to such a nice and neat description. Finally, the references may also be of quite some use to the researched that would need to remind themselves of this or that point of history of the model. Therefore, for the time being, I do not recommend the lecture notes for publication, and kindly ask the authors to complement these with a detailed set of references. If this is done, then I would not see any more obstruction for publication. --- Answer We completely agree. We have significantly enriched the bibliography. From one side we have added many references to the recent literature where the methods we describe are applied to topics interesting for the nowadays scientific research, to show the reader the usefulness of our discussion. From the other side, we have added references about the methods themselves, addressing the reader to alternative methods (like in the evaluation of the entanglement entropy) or to deeper discussions (as in the case of correlations where we have added citations to works about exact formulas, large distance results, and relation to the 2d classical Ising model). In so doing the number of references has increased from 21 to 138. Answer We completely agree. We have significantly enriched the bibliography. From one side, we have added many references to the recent literature where the methods we describe are applied to topics interesting for the nowadays scientific research to show the reader the usefulness of our discussion. From the other side, we have added references about the methods themselves, addressing the reader to alternative methods (like in the evaluation of the entanglement entropy) or to deeper discussions (as in the case of correlations where we have added citations to works about exact formulas, large distance results, and relation to the 2d classical Ising model). In so doing the number of references has increased from 21 to 138.

--- Query Page 9 and several instances that follow. The authors speak of the number of Eigenvalues as a way to count dimensionality. I would suggest to rather speak of the number of Eigenvalues counted with multiplicities. --- Answer We have added the specification “(including multiplicity)” in several places.

--- Query I would suggest to use a slightly different notation for the Fourier transforms of the fermion operators. For instance, to change the letter c typography. That should lead to more clarity. --- Answer We have applied the suggested correction denoting as $\mathrm{c}_k$ the destruction operator in momentum space, differently from $c_j$ for the operator in real space. --- Query Discussion below (69). Taken the overall level of detail given in the lecture notes, I would suggest to provide more explicit details on the content of the paragraph. --- Answer In the new version, we have better clarified that we focus on the fact that the energy difference between the ground states in the two manifolds decreases exponentially in $L$, thereby leading to symmetry breaking in the thermodynamic limit. More details are also given in Sec. 3.2.

--- Query I think it is rather Hermitian than Hermitean. --- Answer Yes. We have applied the suggested correction.

--- Query ”Unfortunately, the computer will produce eigenvectors associated with the degenerate zero-energy eigenvalues which do not have the structure alluded at in Eq” I think that this statement strongly depends on the routine adopted by the computer. Maybe some more explanation of the origin of this problem would be interesting. --- Answer We better clarify now that to enforce such a structure, you can exploit the swap matrix $\mathbb{S}$ and simultaneously diagonalize numerically this operator and the Hamiltonian. --- Query Discussion around (114). Taken the overall level of detail given in the lecture notes, I would suggest to provide more explicit details on the derivation of (114). Here, the authors could recall the central limit theorem and the convergence of averages and could explain a bit more the main steps. --- Answer We have removed this part.

--- Query I could not identify the definition of $\tau_L^*$ in the footnote of exercise 5 on page 40. --- Answer We have now discussed that $\tau_L^*$ is a characteristic time where the finite-size critical gap, scaling as $1/L$, starts to be visible. For $\tau\gg\tau_L^*$ the system becomes adiabatic. For details, we refer to Ref. [35].

--- Query ”using” on page 49 is probably a typo. --- Answer Yes, indeed. Thanks. We have removed that typo.

--- Query page 53, Section 10, paragraph 2 $\rho_L\to \rho_l$. --- Answer We have changed symbols [see new Eq. (249) and surroundings] in order to make them clearer.

--- Query $(t)$ is missing in the definition of $\Lambda$, in the elements $\lambda_k$ as well as in the line below of (270). --- Answer We have removed $t$ everywhere. Thanks to a comment from the other Referee we have recognized that the notion of time is inessential here: one needs only a Gaussian state (that does not necessarily come from time dynamics).  

Report 2

--- Query The lecture notes ’The quantum Ising chain for beginners’ are an introduction to the various set of free fermionic techniques which can be used to study the one-dimensional quantum Ising model. The focus is on explicit computations (spectrum, eigenstates, overlaps, time-dependence, entanglement), of specific but nevertheless useful quantities, which are explained in considerable detail. Overall I find these lectures to be pedagogical, already accessible to a master-level student. The authors often ’go the extra mile’, and explain all the various hidden subtleties related to boundary conditions, fermionic parity, coherent states, etc, which are sometimes neglected in other lectures. --- Answer We thank the Referee for her/his positive assessment of our work.

--- Query
There are also downsides to the way things are presented. Reading these notes sometimes feels like browsing through a list of specific calculations. Those are carried out without discussing motivations, why they are possible, what is important or not. Such explanations would be useful to the targeted audience, especially if they need at some point to be able to compute something slightly different from what is presented. --- Answer We thank the Referee for her/his clear and specific criticism that helped us improve our manuscript. We have added motivations and perspective paragraphs whenever possible. We describe the modifications below when answering the detailed points raised by the Referee.

--- Query Here is a list of general comments, that elaborate further on this criticism. Provided those are addressed, I would be happy to recommend publication in Scipost lecture notes. a) The introduction is too short. It would be nice to explain some of the logic of these lectures here, together with some broader ’take home messages’, which would be useful to the targeted audience. For example, a common feature shared by all sections is that to compute essentially anything in the quantum Ising chain with L sites, the only thing you need to be able to do is diagonalise a matrix of size $2L$, resulting in a massive simplification. --- Answer We have extended the Introduction. From one side, we have emphasized the importance of the recent scientific literature on the models, to which the methods described can be applied, significantly extending the bibliography. From the other side, as the Referee suggests, we have highlighted that essentially any static and dynamic property of the model can be determined, diagonalizing a $2L\times 2L$ matrix.

--- Query b) I also concur with the first referee regarding the first paragraph, which, even though it was probably not the author’s intentions, sounds like trying to make excuses for not providing a proper bibliography. --- Answer We have removed this paragraph and significantly extended the bibliography, adding new references to the recent research literature where the methods described are applied and the (usually older) literature where these methods were developed. The number of references has now increased from 21 to 138.

--- Query c) There are too many footnotes, not even taking into account footnotes in problems and info boxes. While those can be a nice complement to the text, they should be used sparingly. There are several places where those could be put in the main text, or simply removed. For example, the fact that there are 5 long footnotes on page 19 is a sign that something went very wrong with the presentation in this particular place. --- Answer We have pruned the footnotes. We have put some of them in the main text. We have merged and extended some others to provide a new section (the new Section 4 where we discuss the connection with the Onsager solution of the classical 2-dimensional Ising model).

--- Query d) Specific notations are used to distinguish between the various types of operators and matrices. This is a good idea, but explaining those to the nonexpert reader would be even better. For example bold capital letters are used for square matrices of size $L$, while the hat is reserved for fermionic operators, Hamiltonians, of size $2L$. I do not fully understand the mathbb notation, however. In some case it is just a $2L\times 2L$ matrix with blocks in bold capital letters, but the hat mathbb are used to emphasize different parity sectors, for example in section 3. --- Answer We have better clarified, at the end of the Introduction, the notation used, including a table of the main symbols used, with their meaning.

--- Query e) Section 9 and especially 10 are subpar compared to the previous ones, see comments below. While computations say of the energy spectrum does not really require discussing motivations, it is clearly necessary for the entanglement entropy. --- Answer We have extended both sections (now Sections 8 and 9). In the section on the correlators, we have added the correlators $C^{yy}$, $C^{zz}$, added some analytical formulas (presented as solutions of problems), and added references to the very wide literature related to this topic. In the section on the entanglement entropy, we have added a one-page long introduction describing the importance of the entanglement entropy in recent developments of many-body physics, in the contexts of quantum quenches, many-body localization and entanglement transitions, and we have highlighted the central role played in these studies by the methods explained in our manuscript.

--- Query Below is a list of more specific suggestions: 1) section 1, page 3. It would be useful to explicitly state somewhere that for a collection of $L$ spins, the dimension of the Hilbert space is $2^L$. In footnote 1, remove the 'one the same site', since these are just 2 by 2 Pauli matrices, there is no notion of 'same site' yet. 'typical of fermionic commutation rules' reads awkwardly, since there is the anti-commutator. --- Answer We have applied the suggested corrections.

--- Query 2) Page 5. 'Notice that $K_j=K_j^\dagger=K_j^{-1}$ and $K_j^2=1$'. It was already mentioned two lines above that $K_j$ is a sign, so the reader should be able to figure this out by themselves. --- Answer We have removed this sentence.

--- Query 3) Footnote 4 does not fit well with the spirit of these lectures, which deals almost exclusively with finite chains. It can be safely removed. --- Answer We have removed this footnote.

--- Query 4) In section 2 and 3, there is some ambiguity in what is meant by transverse field Ising model. The text is reasonably clear in stating that $J^y=0$ in (18) in this case, but then discussions later on do not always make this assumption, even though section 2 and 3 are named ’Ising model’. Perhaps give a different name to (18). --- Answer Now we introduce the Hamiltonian Eq. (3) of the Introduction and call it XY-Ising Hamiltonian, to avoid confusion. An info box on page 9, including Eq.(23), further clarifies the concept of an anisotropic XY model in a transverse field.

--- Query 5) Page 10. It seems to me the most important point is that ${c_k,\,c_{k'}^\dagger}=\delta_{kk'}$, which is implicitly used later on. The notation in (36), (37) are somewhat nonstandard, and advantageously replaced by $\sum_{k\in\mathcal{K}_p}$ as is done on the next page. Remove footnote 7, or put back in the main text. In footnote 9, replace the first and by a comma. --- Answer Done, thanks.

--- Query 6) Page 11, before (42). $\hat{H}_k$ 'lives in the 4-dimensional space' because all terms with different $|k|$ in the Hamiltonian commute, which is related to the previous point. Also, technically $\hat{H}_k$ are still matrices of size $2L\times 2L$, they just happen to act nontrivially only on this four-dimensional subspace. --- Answer Indeed. We have modified the phrasing and replaced “lives” with “acts nontrivially”.

--- Query 7) Page 13. Figure 2 is more stylish, but figure 3 provides the same information and is much more easy to visualize. --- Answer We fully agree. They are now united as different panels of the new Fig.2

--- Query 8) Page 15. First line in section 3.1. ’in Eq. (59)’. Together with (50), which is nonnegative. Just before (69), the discussion of the energy splitting is not very clear. The reader ’naively expects’ (69) to hold, and they would be perfectly right, since this equation deals only with the extensive part of the energy. Subleading corrections, exponential or not, are irrelevant in (69). --- Answer Done. We have corrected the text before Eq. (69) (now (74)) and we specify that this equation correctly describes the energy density in the thermodynamic limit. For the energy itself, there are also nontrivial sub-leading corrections. Due to these corrections, the ground states in the two symmetry sectors have an energy difference that exponentially decreases with $L$, so the two sectors provide the required ground-state double degeneracy of the ferromagnetic phase.

--- Query 9) At the beginning of section 3.2. ’In this section, we will focus on the Ising case’. But section 3 is named ’Uniform Ising model’, and the same for section 3.1. --- Answer We have modified the titles of the relevant sections. To be more consistent we have rephrased the mentioned sentence as “Because we are focusing on the Ising case, we fix the anisotropy parameter to $\kappa=1$.”

--- Query 10) Page 20. Footnote 20 can be safely put back in the main text. --- Answer Done, thanks.

--- Query 11) While discussing the ’Nambu formalism’, it would be nice to mention how one can recover the results of the previous sections, simply because the matrices A and B have a simple translationally invariant structure. --- Answer We have now added, at the beginning of Sec. 6, the sentence: “We remark that one can recover the results of Sec. (3) when the couplings are uniform and the matrices ${\bf A}$ and ${\bf B}$ have a simple translationally invariant structure.”

--- Query 12) Page 21. Similar abuse of notation as before, with concatenation of two vectors u and v to make a larger vector. --- Answer We have now better clarified the notation used.

--- Query 13) Page 22, footnote 25. ’Similarly, one must have [...]’. This does not bring additional information, since a left inverse is also a right inverse for square matrices. --- Answer Correct. We have updated the note.

--- Query 14) Page 23, before (108). Replace 'The $2^L$ Fock states' by 'The $2^L$ eigenstates'. --- Answer Done, thanks.

--- Query 15) Page 24. $2\epsilon_\mu=\epsilon_k$ looks like a clash of notations. --- Answer We have now rewritten this as $2\epsilon_\mu \rightarrow \epsilon_k$.

--- Query 16) Page 25. ’More in detail’ reads awkwardly. --- Answer Removed.

--- Query 17) Page 25, 26. Make explicitly the point that in the absence of disorder, eigenstates are extended. --- Answer Done, thanks, just before the box for Problem 5.

--- Query 18) The discussion in section 5.3 is pretty good I think. ’a theorem of linear algebra’ becomes a consequence of the Schur decomposition in a later section. --- Answer We have removed the mention of the Schur decomposition and cited the new Ref. [84] in both the points mentioned by the Referee.

--- Query 19) Page 41. The notation 1(0) makes the formulas harder to read. It would be better to replace by $Z_\alpha$ with $\alpha = 0, 1$, or something similar. --- Answer We have applied the suggested correction.

--- Query 20) Discussion after (200). I understand the point, but the assumption of invertibility is actually not a problem since everything is finite-dimensional. Invertible matrices are dense in the space of all matrices, so the Onishi formula holds as a limit when U is not invertible. A mathematician would even be very happy with this. Same comment in the absence of pairings. --- Answer The referee is right. We have now moved this rather technical part to an Appendix, alluding to some specifications to be made (without however entering into details) and moving to the Bloch-Messiah theorem.

--- Query 21) The discussion in section 9 ends quite abruptly. It would be perhaps nicer to mention that these exact formulas are the starting point of many subsequent large distance results –also related to the 2d classical Ising model– and give a few extra references. Also, footnote a in problem 6 can be put in the text. --- Answer We have added a concluding paragraph with citations of works about exact formulas, large distance results, and relation to the 2d classical Ising model (to which the new Section 4 is entirely devoted). The footnote is now in the main text of the problem.

--- Query 22) Page 50. The notation ${l}$ for subsystem is non standard and confusing, since ${l}$ is usually understood precisely as the set containing only $l$. And it gets worse with 'for instance, ${l} = 1, \ldots , l'$, where a standard notation for the rhs would be ${1, \ldots , l}$. Also, it is not clear whether the implication is that ${l}$ necessarily contains l elements, consecutive or not. --- Answer Yes, indeed, thanks. In the new version of the manuscript we use a more standard notation (see Eq. (249) and below) of the new version).

--- Query 23) Page 50, ’For a system described by a quantum state $|\psi(t)\rangle$’. It is strange to use notations implying time-dependence without ever mentioning it, since the calculation of the entanglement entropy has in principle nothing to do with time. You just need a state where Wick’s theorem holds. Of course, this is also the case when time-evolving any such state with a quadratic fermions Hamiltonian. So time could in principle be removed from this whole section, resulting in lighter notations. Perhaps the authors want to make some kind of point that any state where Wick’s theorem holds can be seen in this way, or just consider time-evolved BCS states, in which case they need to explain what they want to discuss, and what assumptions they make. --- Answer We have removed the unnecessary time dependence.

--- Query 24) Page 52, ’its state $|\psi(t)\rangle$ is a gaussian state’. I do not know what is the state of the quantum Ising chain. --- Answer We have now better explained — in several places in the new version — the ideas behind having a Gaussian state for which Wick’s theorem allows calculating any expectation value of fermionic operators in terms of elementary one-particle Green’s functions.”.

--- Query 25) Page 53. There is a section 10.1, but no section 10.2. In step 2, ’to a sub-chain ${l} = 1, \ldots , l$ say’. This assumption is not innocent at all. For a chain with PBC, it is in fact very important that the ’subchain’ be in one connected piece. Otherwise the calculation explained in this section breaks down due to issues with Jordan-Wigner strings. That point should be made clearly, otherwise the ’beginner’ reader might think this holds as well for subsystem 1, 3, 5, 8, which it does not. Also, subsystem is possibly clearer than subchain. --- Answer We have stressed that the subchain ${ 1, \cdots, l}$ is connected. We have removed the separation in subsections of the Section about entropy (the present Section 9).

--- Query 26) Page 53. Footnote 53 is not very clear, since much more than normalization is lost for unitary matrices, and the Nambu matrix is not unitary. --- Answer We have eliminated this unnecessary footnote.

--- Query 27) The derivation presented here is fine, but other methods should at least be mentioned as well, also in the slightly broader context of entanglement computations in free bosonic or fermionic systems, dating back to Bombelli et al, Srednicki, etc. This is perhaps a matter of taste, but I find the (closely related) derivation of Peschel to be easier to understand, especially for non experts. It just explains why the reduced density matrix (RDM) is the exponential of a quadratic fermion Hamiltonian, similar to Step 1. In fact, the explicit expression for the RDM follows from (277) in the text. --- Answer We cite the mentioned references in the new version (see new Refs. [132-135]).

--- Query 28) Problem 7. Give at least a few references for the logarithmic divergence, including the paper by Holzhey, Larsen, Wilczek. --- Answer We cite the mentioned reference in the new version (see new Ref. [136]).

---

## Round 2 · List of Changes

We have added references about the methods we describe and their applications. The number of references has increased from 21 to 138.

We have reduced the number of footnotes by integrating some of them in the main text.

On pages 2-4, we have extended the Introduction, emphasizing the scientific literature of the models to which the methods described can be applied.

On page 2, in the Introduction, we have introduced the XY-Ising Hamiltonian in Eq. (3).

On page 4, we have stated that, for a chain of L spins, the Hilbert space dimension is $2^L$ and have highlighted that essentially any static and dynamic property of the model can be determined, diagonalizing a $2L \times 2L$ matrix.

On page 4, in the Introduction, we have better clarified the notation used, including a table of the main symbols used, with their meaning.

On page 5, we have improved the wording by replacing ‘fermionic commutation rules’ with ‘canonical anti-commutation rules for fermions’.

On page 7, We deleted the redundant statement ’Notice that $K_j=K_j^\dagger=K_j^{-1}$ and $K_j^2=1$’.

On pages 4-9, we added several intermediated steps in the derivation of the Jordan-Wigner transformation.

On page 9, we have added an info box that explains the anisotropic XY model in a transverse field.

On page 11, we change Section 3’s title to “Uniform XY-Ising model”.

On page 11, we mentioned Ref. [65], where the uniform case was originally solved.

On page 12, we have changed the notation, denoting as $\mathrm{c}_k$ the destruction operator in momentum space, differently from $c_j$ for the operator in real space.

On page 13, we have specified that $[\hat{H}_k$ commute for different $k$ and act non-trivially only the subspace generated by the states in Eq. (47).

On page 14, we have added the specification “(including multiplicity)” when counting Eigenvalues.

On page 15, in footnote 11, we explicitly computed the anticommutator $\left\{\hat{\gamma}}_{k}, {\hat{\gamma}^{\dagger}}_{k}\right\}$.

On page 16, we joined to figures as different panels of the new Fig. 2.

On page 17, we renamed Section 3.1 as ‘Ground state and excited states of the uniform XY-Ising model’.

On page 19, we clarified that Eq. (74) describes the energy density in the thermodynamic limit and commented on the role of nontrivial sub-leading corrections.

On page 19, we have clarified that the paragraph below Eq. (74) focuses on the fact that the energy difference between the ground states in the two manifolds decreases exponentially in L, thereby leading to symmetry breaking in the thermodynamic limit.

On page 20, we explicitly state “Because we are focusing on the Ising case, we fix the anisotropy parameter to κ = 1.”

On pages 21-22, we added two entirely new sections, Section 3.1.1, titled ‘The spectral gap’ and Section 3.1.2, titled ‘The spectral gap The Green’s functions’.

On page 24, we clarified the meaning of the expression “macroscopically ordered”.

On pages 24-31, we added Section 4, a new section where we discuss the connection with the Onsager solution of the classical 2-dimensional Ising model.

On page 31, we have mentioned Refs. [65,73,74].

On page 32, we added Eq. (123) and commented on it.

On page 33, we have mentioned that one can recover the results of Sec. 3 when the couplings are uniform.

On page 33, have clarified that the notation $\left( \begin{array}{c} {\textbf u}_{\mu}\\ {\textbf v}_{\mu} \end{array} \right)$ indicates a $2L$-dimensional column vector.

On page 35, we have simplified footnote 24 using the fact that a left inverse is also a right inverse for square matrices.

On page 35, before Eq. (114). We replace ’The $2^L$ Fock states’ with’The $2^L$ eigenstates’.

On page 36, we clarified that to enforce the structure alluded to in Eq. (134), we can exploit the swap matrix S.

On page 37, we have replaced ‘$2\epsilon_\mu=\epsilon_k$’ by ‘$2\epsilon_\mu \rightarrow \epsilon_k$’.

On page 37, we have explicitly stated that in the absence of disorder, eigenstates are extended.

On pages 37-38, we simplified the discussion on Anderson localization and added references that discuss the topic in depth.

On page 44, after writing ‘According to a theorem of linear algebra’, we have cited Ref. [84].

On page 45, we renamed Section 7 ‘Schrödinger dynamics in the time-dependent case’ and added multiple references on the topic.

On page 50, in Problem 7, we defined $\tau_L^*$ to be a characteristic time where the finite-size critical gap, scaling as 1/L, starts to be visible.

On page 51, we mentioned the field of Floquet engineering and added references on the topic.

On page 54, we added Problem 8.

On pages 53-58, we have extended Section 8. We have added the correlators $C^{yy}$, $C^{zz}$, added some analytical formulas (presented as solutions to problems), and added references to the very wide literature related to this topic.

On page 58, we mentioned that the exact formulas are the starting point of many subsequent large-distance results and added multiple references on the topic.

On page 58, in Problem 11, we integrated the footnote into the main text.

On pages 58-59, We have extended Section 9 by including an introduction describing the importance of entanglement entropy in recent developments of many-body physics and the central role played in these studies by the methods explained in our manuscript.

On page 59, we have changed symbols ($\rho_L\to \rho_l$) to make the notation clearer.

On page 59, we dropped the ambiguous notation $\{l\}$ and $\{L\}$ in favor of the more standard notation $\{1, \ldots, l\}$ and $\{l+1, \ldots, L\}$ to denote the subsystems considered.

On page 59, we removed the unnecessary time dependence of the state $|\psi\rangle$ and changed the following equations accordingly.

On page 59, we have stressed the assumption that the considered subsystem $\{1,\ldots, l\}$ is connected.

On page 61, we have clarified that the ground state of the quantum Ising chain has a Gaussian form and that the relevant expectation values can be computed with Wick’s theorem.

On page 63, starting from Eq. (269), we have removed $t$ from various equations because the notion of time is irrelevant here.

On page 64, we removed the mention of the Schur decomposition and cited Ref. [84] instead.

On pages 58-65, we removed the separation in subsections of Section 9.

On page 65, in Section 10, we added a paragraph introducing the section and mentioning Ref. [76]

On page 67, we derived Eq. (287).

On page 69, we added Problem 13.

On page 69, we added Section 10, containing the conclusions of the lecture notes.

On page 70, we have moved the discussion on ‘overlap between BCS states’ to Appendix A.

On page 70, we replaced the notation $Z_{1(0)}$ by $Z_\alpha$ with $\alpha = 0, 1$.

On page 71, after Eq. (304), we explicitly stated that the given derivation would call for appropriate specifications and mentioned some issues that were not addressed in it.

---

## Editorial Decision

published